# Improving Pacing in Long-Form Story Planning

**Yichen Wang**[1,2,†]    **Kevin Yang**[1]    **Xiaoming Liu**[2]    **Dan Klein**[1]
[1]University of California, Berkeley    [2]Xi'an Jiaotong University
yichen.wang@stu.xjtu.edu.cn,{yangk,klein}@berkeley.edu,xm.liu@xjtu.edu.cn

## Abstract

Existing LLM-based systems for writing long-form stories or story outlines frequently suffer from unnatural pacing, whether glossing over important events or over-elaborating on insignificant details, resulting in a jarring experience for the reader. We propose a **CONC**rete **O**utline **C**on**T**rol (CONCOCT) system to improve pacing when automatically generating story outlines. We first train a *concreteness evaluator* to judge which of two events is more concrete (low-level-detailed). This evaluator can then be used to control pacing in hierarchical outline generation; in this work, we explore a *vaguest-first* expansion procedure that aims for uniform pacing. We further use the evaluator to filter new outline items based on predicted concreteness. Compared to a baseline hierarchical outline generator, humans judge CONCOCT's pacing to be more consistent over 57% of the time across multiple outline lengths; the gains also translate to downstream stories. All code, data, and models are open-sourced.[1]

## 1   Introduction

Recent advancements in large language models have led to increased interest in long-form generation, especially in creative writing settings such as stories or books (Yang et al., 2022b,a; Zhou et al., 2023). Such efforts have tackled a wide range of challenges arising in longer outputs, such as long-range coherence and internal factual consistency.

Another important problem in longer outputs is *pacing*, our focus in this work. For example, it would be an unpleasant surprise for a fantasy story to summarize a major plot point as merely e.g., "The characters went on an arduous journey." Conversely, it would be very odd if an entire half of the same story were devoted to a single dialogue.

In fact, pacing-related issues very frequently plague LLM-generated stories and story outlines.

For instance, Yang et al. (2022a) observed that their generated outlines frequently suffer from inconsistent pacing, even after hierarchically expanding high-level events to the same final depth. Poor outline pacing translates directly to the resulting story, and may be exacerbated in lengthier outlines corresponding to longer stories or books, as corroborated by Coetzee (2023) when writing a full-length book with GPT-4 (OpenAI, 2023). Coetzee (2023) noted that some overly detailed chapters felt like "a slog," while in other cases GPT-4 would "breeze right over big important moments with a summary."

Therefore, we propose the Concrete Outline Control system (CONCOCT) to better control pacing in LLM-generated story outlines. We first train a concreteness evaluator to judge which of two event descriptions is more concrete[2], constructing a large training dataset of passage summaries with varied granularity that we name GPT-BOOKSUM. Our concreteness evaluator can then be used in hierarchical outline generation to control or vary pacing as desired; in this work, we demonstrate its ability to maintain uniform pacing via a vaguest-first expansion procedure. We use the evaluator both to select outline nodes to expand, as well as to filter newly generated nodes based on concreteness.

Compared to baseline hierarchical outlines of similar length, CONCOCT's story outlines are judged by humans to have more consistent pacing over 60% of the time without compromising other qualities (Sec. 4). Downstream stories based on CONCOCT's outlines are also judged to have more consistent pacing in over 57% of cases (Sec. 4.3).

## 2   Related Work

**Concreteness Evaluation.** Existing works in psycholinguistics and cognition evaluate word-level concreteness by human annotation (Paivio et al.,

---

† Work done while at Berkeley.
[1]https://github.com/YichenZW/Pacing.

[2]We define concreteness as "the degree to which language has a perceptible physical referent" (Hill and Korhonen, 2014).

1968; Brysbaert et al., 2014). Other efforts model word-level concreteness using classical forward search (Turney et al., 2011) or regression models (Ljubesic et al., 2018; Charbonnier and Wartena, 2019; Yang et al., 2022c). In contrast, we model concreteness on a sentence or passage level.

**Length-Controlled Generation.** Several summarization methods control the length of output summaries (Kikuchi et al., 2016; Cohan et al., 2018; Sarkhel et al., 2020; Liu et al., 2022; Miculicich et al., 2023). Meanwhile, some recent story generation methods use hierarchical outlines for planning (Rashkin et al., 2020; Tian and Peng, 2022; Yang et al., 2022b,a), which can grant some control over the length of story passages. However, while pacing may often correlate with word length, it is not the same. Rather than controlling outline sections to have similar surface-level length, CONCOCT focuses on a semantic notion of granularity.

## 3 Concrete Outline Control

We now present our method, Concrete Outline Control (CONCOCT). CONCOCT first constructs a *concreteness evaluator* $\mathbb{M}$ to enable pacing control in outlines. We then use $\mathbb{M}$ to run a *vaguest-first expansion* procedure to maintain uniform pacing as well as a concreteness filter for new outline nodes.

### 3.1 Concreteness Evaluator

It is hard to define "concreteness" quantitatively for a single text, but easier when comparing two texts. Therefore, our concreteness evaluator $\mathbb{M}(t_0, t_1)$ will operate on two texts, $t_0$ and $t_1$, outputting the probability that $t_1$ is more concrete.

**Dataset Construction.** We construct a large dataset of summaries of raw story passages of varying lengths from Project Gutenberg (Hart, 1971), as shown in Figure 1. We use the same passage boundaries as in the BOOKSUM dataset (Kryscinski et al., 2021). However, our summaries are written by ChatGPT (gpt-3.5-turbo-0301; Appendix A.1) (OpenAI, 2022). We thus obtain summaries written in a uniform style, which is important for training our concreteness evaluator $\mathbb{M}$ to focus on concreteness rather than differences in writing style.[3]

Table 1 shows the statistics of our dataset, which we refer to as GPT-BOOKSUM.

---

[3]We initially used BOOKSUM's summaries, but found that different-level summaries were often written in different styles, e.g., chapter-level summaries are often bullet-point lists.

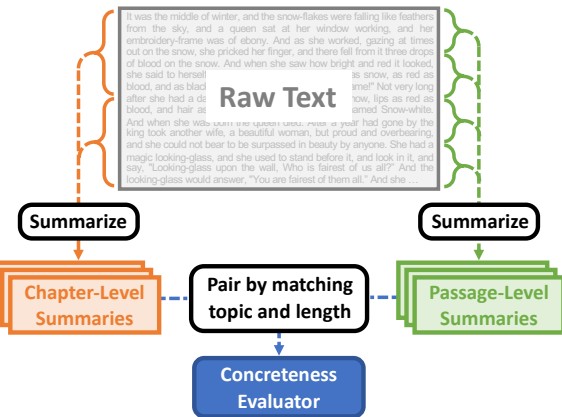

Figure 1: Concreteness evaluator training. Raw texts are chunked into chapters or passages and summarized using ChatGPT. Summaries are then paired and truncated so that training pairs have similar topic and length.

**Concreteness Evaluator Training.** We now use GPT-BOOKSUM to train our concreteness evaluator $\mathbb{M}$. We construct training pairs $(t_0, t_1)$ as follows:

1. Sample summaries from GPT-BOOKSUM which have not yet been used for training, and pair them by top mean embedding similarity using Contriever (Izacard et al., 2021).
2. With 50% probability, truncate the longer summary to roughly the length of the shorter one. Otherwise, truncate both summaries to the same token length, randomly chosen on a log scale from 25 to 180. Sentence boundaries are respected whenever truncating.

By matching topic and length within a training pair $(t_0, t_1)$, we encourage $\mathbb{M}$ to focus on the actual vagueness or concreteness of the exposition (see Appendix B.4 for analysis).

Finally, $\mathbb{M}$ is initialized as RoBERTa-Large (Liu et al., 2019) with a classification head. The actual model input is "$t_0$  $t_1$", using a separator token . As chapter-level summaries are dramatically more compressed than paragraph-level summaries (Table 1), we label the chapter-level summary as vaguer when paired with a paragraph-level summary. The label is 0.5 if $t_0$ and $t_1$ are same-level summaries; we found including 0.5 labels to be empirically beneficial.

### 3.2 Outline Generation

CONCOCT uses our concreteness evaluator $\mathbb{M}$ to improve outline pacing in two ways: vaguest-first expansion order and concrete candidate generation.

**High-Level Outliner Structure.** We view a hierarchical outline as a tree, rooted at the overall

| Split | Chapter-Level | | | | Paragraph-Level | | | |
|---|---|---|---|---|---|---|---|---|
| | Size | Summary Len | Raw Len | Raw / Sum | Size | Summary Len | Raw Len | Raw / Sum |
| *Train* | 23,564 | 133.7 | 5450.7 | 40.77 | 162,122 | 58.6 | 71.6 | 1.22 |
| *Val* | 3,086 | 134.2 | 4607.8 | 34.34 | 58,648 | 56.6 | 63.7 | 1.13 |
| *Test* | 3,397 | 135.1 | 5440.8 | 40.27 | 59,965 | 59.5 | 76.4 | 1.28 |

Table 1: GPT-BOOKSUM dataset statistics for chapter-level and paragraph-level summaries: number of passage-summary pairs, average token count of summaries and raw texts, and ratio of total token count in the raw texts compared to after summarizing. Training, validation, and test sets are partitioned at the book level.

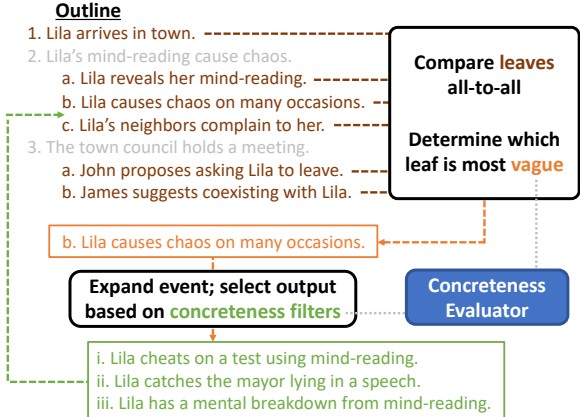

Figure 2: Stylized example of an outline expansion step. Among all leaf nodes, we select the node which is vaguest according to our concreteness evaluator. We then generate child events for the selected node, filter for concreteness, and finally insert back into the outline.

story premise. Nodes contain plot events. In each outline expansion step, a leaf node is selected and expanded into child nodes describing sub-events.

**Vaguest-First Expansion Order.** Rather than using a fixed breadth-first expansion as in e.g., Yang et al. (2022a), we leverage our concreteness evaluator $\mathbb{M}$ to run *vaguest-first* expansion order.

Specifically, at each step of outline expansion, for each leaf $n_i$ we compute the average probability that $n_i$ is more concrete compared to other leaves: $\mathbb{M}_{avg}(n_i; \mathcal{L} \setminus \{n_i\}) = \frac{1}{|\mathcal{L}|-1} \sum_{l \in \mathcal{L} \setminus \{n_i\}} \mathbb{M}(l, n_i)$, where $\mathcal{L}$ is the set of current leaves. We expand the node $n_v$ with minimal $\mathbb{M}_{avg}(n_i; \mathcal{L} \setminus \{n_i\})$, i.e., $n_v$ is the vaguest relative to other leaves.

**Concrete Children Generation.** Vaguest-first expansion on its own does not guarantee that child nodes will be more concrete than their parent. Therefore, we also use $\mathbb{M}$ to filter candidate children (i.e., sub-events) during outline expansion.

Child generation begins by proposing two or more candidate children $c_1 \ldots c_m$ under parent node $n_v$ by prompting ChatGPT, using all of $n_v$'s ancestors and their respective children as context

(Appendix C.1). Each child $c_j$ must then satisfy:

1. $c_j$ should not be overly similar to $n_v$. In particular, we enforce that neither $c_j$ nor $n_v$ should be contained in the other, and that their cosine similarity should not exceed 0.9 according to Contriever (Izacard et al., 2021).
2. Compared to other leaf nodes $\mathcal{L} \setminus \{n_v\}$, the child $c_j$ should be more concrete than the parent $n_v$. That is, $\mathbb{M}_{avg}(c_j; \mathcal{L} \setminus \{n_v\}) - \mathbb{M}_{avg}(n_v; \mathcal{L} \setminus \{n_v\})$ must exceed a threshold $T$, which decreases over time (Appendix C.2).

When $c_j$ fails to satisfy these criteria, we regenerate it using ChatGPT (Appendix C.3). If we cannot generate a satisfactory $c_j$ after several attempts, we restart the entire expansion of $n_v$ with increased temperature for ChatGPT. Very rarely, expansion is still unsuccessful, in which case we give up on $n_v$ and expand the next-vaguest leaf in $\mathcal{L}$ instead.

### 3.3 Downstream Story Generation

The end goal when generating story outlines is to generate actual stories. Although the process of turning an outline into a full story is not our main focus, we nevertheless apply an existing story generation system, DOC (Yang et al., 2022a), to turn CONCOCT's outlines into complete stories for evaluation purposes. To keep story pacing more consistent with the original outline, we simplify DOC by just generating a fixed-length story passage for each outline item rather than dynamically varying the passage length as in Yang et al. (2022a). Furthermore, to be consistent with our outline generation system, we modify DOC to use ChatGPT rather than their original OPT-175B (Zhang et al., 2022). See Appendix E for complete details.

### 4 Experiments

Our task is to generate a story with consistent pacing, given a brief input premise from the Writing-Prompts dataset (Fan et al., 2018).

| Model | Short Outline | | | | Long Outline | | | |
|---|---|---|---|---|---|---|---|---|
| | Pacing↑ | Vague↓ | Detailed↓ | Other↓ | Pacing↑ | Vague↓ | Detailed↓ | Other↓ |
| BASE | 38.5 | 5.8 | 3.0 | 4.0 | 35.0 | 5.7 | 5.1 | 6.9 |
| CONCOCT | **61.5** | **4.9** | **2.8** | **3.5** | **65.0** | **3.4** | **3.2** | **6.0** |

Table 2: Human evaluation results for BASE and CONCOCT under *Short Outline* and *Long Outline* regimes. Humans judge CONCOCT's outlines to have substantially more consistent pacing in pairwise comparisons (Pacing) with a significant difference, and mark a smaller percentage of leaf nodes as overly Vague, Detailed, or Otherwise problematic.

**Baseline.** Our baseline BASE expands outlines with ChatGPT using the same prompts as CON-COCT, but expands breadth-first instead of vaguest-first, and does not filter new nodes for concreteness.

**Task Variations.** We conduct experiments under two regimes of average outline length (measured in leaves): *Short Outline* and *Long Outline*. To optimize BASE performance, these regimes are defined via the average length of BASE outlines when expanding the tree uniformly to depth 3 or depth 4 respectively (treating the root premise as depth 0). In contrast, CONCOCT can specify length more flexibly, based on a total number of node expansions. CONCOCT closely matches the length of BASE's outlines when fixing 12 and 25 total node expansions in the *Short Outline* and *Long Outline* settings respectively (Appendix D.1).

**Metrics.** As it is unclear how to evaluate pacing automatically, we rely on human evaluation. For each of 100 premises in both the *Short Outline* and *Long Outline* regimes, we generate outlines using BASE and CONCOCT and show human annotators the flattened list of leaves from both outlines (randomly truncated to 20 leaves in the *Long Outline* regime). Annotators indicate which outline has more consistent pacing overall, and mark leaves which stand out as too vague, too detailed, or otherwise problematic; see Appendix D for complete annotation details. We then track the following metrics:

1. *Pacing*, our main metric, defined as the percentage of outlines that annotators judge to have more consistent overall pacing (well-defined only for pairwise comparison).
2. *Vague*, the percentage of leaves marked as too vague relative to surrounding context.
3. *Detailed*, the percentage marked too detailed.
4. *Other*, the percentage marked as other errors.

**Results.** As shown in Table 2, humans judge CONCOCT's pacing to be more consistent than BASE over 60% of the time in both length regimes, demonstrating CONCOCT's effectiveness at control-ling pacing. Annotators also marked fewer nodes as overly vague or detailed in CONCOCT, with a larger difference in the *Long Outline* regime, suggesting that the value of CONCOCT may be higher for longer outlines. Finally, the frequency of other, non-pacing-related errors is similar in BASE and CONCOCT, i.e., CONCOCT is not making sacrifices elsewhere to maintain consistent pacing.

Qualitative inspection confirms that CONCOCT prioritizes expanding vaguer, higher-level nodes. See Appendix G for example outlines.

| Model | Long Outline | | |
|---|---|---|---|
| | Coherent↑ | Relevant↑ | Interesting↑ |
| BASE | 45.76 | 47.46 | **54.24** |
| CONCOCT | **54.24** | **52.54** | 45.76 |

Table 3: Human evaluation results on non-pacing errors for BASE and CONCOCT under *Long Outline* regimes. Humans judge CONCOCT and BASE to perform similarly on plot coherence, premise relevance, and interestingness; none of the differences are significant.

### 4.1 Non-Pacing Error Analysis

In our previous human evaluations, we asked annotators to simply label all non-pacing-related errors as "other errors." To more comprehensively verify that CONCOCT does not compromise other desirable qualities in the pursuit of consistent pacing, we run human evaluations following the main metrics from Yang et al. (2022a), asking annotators to compare outlines from our *Long Outline* regime solely on overall plot coherence, premise relevance, and interestingness; see Appendix D.3 for further details on evaluation setup.

**Results.** While CONCOCT is significantly better on pacing (65.0 to 35.0 in Table 2), none of the differences in non-pacing-related qualities (Table 3) are significant; CONCOCT's average across these three metrics is even slightly higher than BASE's. The results corroborate our previous finding with the "other errors" metric that CONCOCT does not compromise non-pacing-related qualities.

| Model | Test Acc.↑ | Human-Vague Acc.↑ | F1↑ | Human-Detailed Acc.↑ | F1↑ |
|---|---|---|---|---|---|
| GPT-3.5 | 0.401 | 0.482 | 0.438 | 0.514 | 0.452 |
| GPT-4 | 0.415 | 0.544 | 0.527 | 0.487 | 0.455 |
| $\mathbb{M}$ | **0.900** | **0.549** | **0.588** | **0.541** | **0.485** |

Table 4: Classification accuracy on GPT-BOOKSUM test set (*Test*) and on outline points marked by humans as too-vague (*Human-Vague*) or too-detailed (*Human-Detailed*), as well as F1 on human-marked points. Results shown for GPT-3.5, GPT-4, and our concreteness evaluator $\mathbb{M}$. $\mathbb{M}$ performs best on all three tasks.

## 4.2 Concreteness Evaluator Analysis

We also analyze the performance of our concreteness evaluator $\mathbb{M}$, comparing to the latest versions of GPT-3.5 and GPT-4 at time of writing (gpt-3.5-turbo-0613 and GPT-4-0613) on three evaluation sets:

1. *Test*, a subset of GPT-BOOKSUM's test set,
2. *Human-Vague*, the set of human-labeled too-vague nodes from our *Short Outline* experiments, where the task is to classify against other nodes from the same outline, and
3. *Human-Detailed*, the same task for human-labeled too-detailed nodes.

We measure classification accuracy (thresholding at 0.5 for $\mathbb{M}$) on all three sets, and F1 for detecting the human-marked point on the latter two.

**Results.** As shown in Table 4, our concreteness evaluator $\mathbb{M}$ compares favorably to GPT-3.5 and GPT-4, which perform at or worse than random chance despite heavy prompt engineering (Appendix B.5). We hypothesize that GPT-3.5 and GPT-4 do not possess a clear grasp of vagueness and concreteness. Meanwhile, $\mathbb{M}$ not only shows strong performance on the GPT-BOOKSUM distribution on which it was trained, but also achieves comparatively higher agreement with human annotations, though performance is far from perfect.

## 4.3 Evaluation of Downstream Stories

Finally, we verify whether CONCOCT's improvements at the outline level extend to downstream stories.

**Setup.** As the resulting stories are quite long (often >5,000 words) even when using outlines from our *Short Outline* regime, we compare similar-length excerpts (around 1,000 tokens) rather than complete stories. The sample size is 100 stories and

126 excerpts. We again evaluate using human annotators; see Appendix F for full setup details. We additionally evaluate with GPT-4 in Appendix F.2.

| Model | Human Evaluation | | | |
|---|---|---|---|---|
| | Pacing↑ | Coherent↑ | Relevant↑ | Interest↑ |
| BASE | 42.82 | 49.26 | **50.50** | 46.29 |
| CONCOCT | **57.18** | **50.74** | 49.50 | **53.71** |

Table 5: Human evaluation results on story excerpts based on outlines from BASE and CONCOCT under *Short Outline* regime. Only the difference in Pacing is significant with $p < 0.05$, which means CONCOCT's gains in pacing translate to downstream stories without compromising non-pacing qualities.

**Results.** As shown in Table 5, although turning outlines into stories introduces more noise, CONCOCT's story excerpts are still judged to be significantly more consistently-paced while not compromising other qualities. The results demonstrate that the gains from CONCOCT on outlines correlate fairly closely with gains on downstream stories.

## 5 Discussion

In this work, we have introduced the CONCOCT system for controlling pacing in hierarchical story outline generation. CONCOCT uses a concreteness evaluator to run a vaguest-first expansion procedure and to filter new outline items for concreteness, with strong results on human evaluations on both outlines and final stories.

Nevertheless, pacing remains far from solved. While CONCOCT provides effective *methods* for measuring and controlling pacing via our concreteness evaluator, the best *objective* for pacing remains an open question: uniform pacing is just one of many potential goals. For example, human authors may *intentionally* vary story pacing, narrating major events in great detail or fast-forwarding through less important sections. Accordingly, more sophisticated pacing-aware outline expansion strategies might attempt to account for nebulous concepts like story "likability," "engagingness," or "interestingness," on top of simply maintaining uniform pacing.

## Limitations

As mentioned in the discussion, while CONCOCT provides effective *tools* for controlling pacing, it is not obvious what *objective* we should optimize for to maximize the quality of the final story. While

we demonstrate effectiveness in maintaining uniform pacing in story outlines with our vaguest-first expansion procedure, it may be desirable at times to intentionally vary the pacing in order to make the story more interesting.

As presented in this work, CONCOCT is designed primarily for the story domain, which accounts for a substantial fraction of long-form texts in the real world. However, there are many other types of long-form outputs that one may wish to generate, such as Wikipedia articles or movie scripts. While we believe that adapting CONCOCT to other domains shouldn't be a problem in principle, in practice, it may require rewriting many of our prompts.

We focus on English story outlines; CONCOCT's performance may suffer in other languages—especially lower-resource languages—depending on the multilingual capabilities of the underlying LLMs, and due to having fewer resources available for training our concreteness evaluator. However, comparing *relative* quality to an unaugmented baseline using the same base LLMs, we believe that using CONCOCT would still result in more uniformly paced outlines and stories. In any case, we have open-sourced all code and other artifacts to maximize reproducibility.

For evaluation, we mainly rely on human evaluation, as it is difficult to automatically evaluate complex notions such as "pacing," "interestingness," "coherence," and "relevance" on long-form outlines or stories. Even so, human evaluation can still be noisy, especially on longer outputs.

## Ethics Statement

As our hierarchical outline generation scheme is based on prompting LLMs (ChatGPT in our implementation), we may inherit any biases present in the LLMs we rely on. While we focus on creative story generation applications in this work, where the potential for real-world harm is relatively smaller, it is nevertheless possible that our system could generate toxic or harmful content if used with malicious intent, e.g., feeding a harmful premise. Of course, by the same token, due to our use of LLM prompting, we can also take advantage of any future advancements in LLMs that mitigate such harms.

Similarly, as mentioned in Limitations, CONCOCT's performance might suffer in languages other than English, both due to weaker performance in the LLMs we rely on and due to fewer available

data for training our concreteness evaluator.

## Acknowledgements

We thank our anonymous reviewers as well as the Berkeley NLP group for their helpful discussions and feedback, which helped us to improve the paper greatly. This work is supported by Berkeley AI Research, Open Philanthropy, DARPA under the SemaFor program (HR00112020054), the Machine Common Sense (MCS) program under Cooperative Agreement N66001-19-2-4032, and the NSF through a fellowship to the second author. This work is also supported by the National Natural Science Foundation of China (62272371, 62103323, U21B2018) through the third author. The content does not necessarily reflect the position or the policy of any government, and no official endorsement should be inferred.

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

# A  Dataset Details

We now discuss the construction of GPT-BOOKSUM in greater detail.

## A.1  Prompt Design for Summarization

The prompt design for summarization follows instructions from Super-NaturalInstructions (Wang et al., 2022). Table 6 shows the prompt.

```
{"role": "user", "content": "Write a summary
for the paragraph.\n\n"}
{"role": "user", "content": "Paragraph: {Input
Raw Text}"}
{"role": "assistant", "content": "Summary: In
this paragraph, the main story is as follows."}
```

Table 6: Prompt for GPT-3.5-turbo-0301 to summarize for GPT-BOOKSUM.

Since GPT-3.5-turbo-0301 has a context window limit of 4,097 tokens, sometimes even a single chapter will exceed the limit. For such texts, we split them into sub-parts at sentence boundaries.

To avoid potential artifacts that may allow the evaluator to trivially discriminate summary-level texts, we prevent summaries from using words indicating a level of granularity, such as "chapter",

"paragraph", etc. We also delete the titles of chapters and books in the data to mitigate the likelihood of the language model making inferences based on previously memorized knowledge.

## A.2 Format of Data

Table 7 shows an example from GPT-BOOKSUM.

```
"level": "chapter",
"text": "Emily, Mons. Du Pont, and Ludovico
are attempting to escape from Montoni's castle.
They hurry down staircases and through
passageways, trying to avoid being caught.
Annette is also in tow, and they hear a
tumultuous sound from the inner court.
Ludovico talks with a sentinel, and they manage
to make it past the gates and into the woods.
They choose to head towards Tuscany, but
Ludovico warns them about bandits. They travel
in silence, thinking of the events that have
unfolded and hoping for a better future.",
```

Table 7: Example from GPT-BOOKSUM (metadata omitted). Each example contains a text passage together with a label for whether that passage's events are at chapter-level or paragraph-level granularity.

## B Evaluator Detail

We frame the task of concreteness prediction as binary classification between two passages, where the goal is to predict which is more concrete. We assign a label of 0 to first-is-more-concrete pairs and 1 to second-is-more-concrete pairs. Furthermore, we assign the third label 0.5 to pairs with the same level of granularity (i.e., chapter-chapter, paragraph-paragraph).

### B.1 Metrics Used in Training Evaluator

We design the metrics below to measure performance, where #[label] represents the number of data predicted as the given label, #[pred, ans] represents the number of data with true label ans that are predicted as pred by the model, and #tot is the total size of the evaluation set.

- **Accuracy** across all three classes 0, 0.5, 1, i.e., (#[0,0]+#[0.5,0.5]+#[1,1])/#tot
- **Loss**, i.e., binary cross entropy loss.
- **Neutral**, the percentage of neutral (0.5) predictions by the model, i.e., #[0.5]/#tot . As half of the pairs are neutral in the dataset, a Neutral value closer to 0.5 is better.
- **Partial**, the percentage of non-neutral predictions, i.e., (#[0]+#[1])/#tot. As half of the

pairs are non-neutral in the dataset, a Partial value closer to 0.5 is favorable.
- **False-Neutral**, the percentage of data with true label 0 or 1 which are incorrectly predicted as 0.5, i.e., (#[0.5,0]+ #[0.5,1])/#tot
- **True-Partial**, the percentage of data with true label 0 or 1 which are predicted correctly, i.e., (#[0,0]+#[1,1])/#tot
- **Major-False**, the percentage of "major errors," i.e., (#[0,1]+#[1,0])/#tot

### B.2 Hyperparameters

Table 8 shows the hyperparameters for training the concreteness evaluator.

| model | RoBERTa Large |
|---|---|
| max sequence length | 512 |
| training batch size | 8 |
| eval batch size | 16 |
| learning rate | 6e-6 |
| weight decay | 0.0 |
| adam epsilon | 1e-8 |
| max grad norm | 1.0 |
| epoch number | 28 |

Table 8: Hyperparameters used in training stage for the concreteness evaluator.

### B.3 Dynamic Pairing in Training

During the training stage of the concreteness evaluator (Sec. 3.1), we apply a *dynamic pairing* strategy to sample new passage pairs for each training "epoch" (in practice, 1000 pairs per epoch). In particular, we ensure that any given pair of passages is never repeated throughout the training process, and additionally ensure that no individual passage is used more than once in a single epoch.

Moreover, we use topic and length matching during pairing as discused in Sec. 3.1, to decrease the likelihood of the model learning undesirable correlations.

### B.4 Ablation for Topic Matching

Table 9 shows the performance of our concreteness evaluator compared to an ablated version without topic matching. We use the metrics in Sec. B.1 to evaluate, and observe that topic matching during training improves the performance of the concreteness evaluator on all metrics.

| Model | C.E. w/o Match | C.E. |
|---|---|---|
| Accuracy ↑ | 0.7285 | **0.7524** |
| Loss ↓ | 0.8077 | **0.7539** |
| Neutral (→0.5) | 0.3833 | **0.4686** |
| Partial (→0.5) | 0.6167 | **0.5314** |
| False-Neutral ↓ | 0.1875 | **0.0986** |
| True-Partial ↑ | 0.2993 | **0.3536** |
| Major-False ↓ | 0.0097 | **0.0045** |

Table 9: Performance of our concreteness evaluator (**C.E.**) compared to an ablated version without topic matching (**C.E. w/o Match**). Our final version **C.E.** is better on all metrics.

### B.5 Prompts for GPT-3.5 and GPT-4

GPT-3.5 and GPT-4 perform quite poorly on our GPT-BOOKSUM Test set, often worse than random chance. We tried several different prompt formats, as shown in Table 10, 11, 12, 13, 14, 15, 16, 17, 18, 19, 20, 21, but none of them work better than Table 10, which gets the best result shown in Table 4.

---

"role": "user", "content": f'Please judge which of the two passages below is written in a more detailed style. Make sure to judge not based on the length of the passage and the order of input, but only by the style of descriptions. \n\n\n\n Passage (A): paras[0]\n\n\n\n Passage (B): paras[1]\n\n\n\n Which passage is written in a more low-level-detailed style? Please answer "Passage (A)" or "Passage (B)." '

---

Table 10: The **best** prompt we could find for GPT-3.5 and GPT-4 on GPT-BOOKSUM classification.

Other prompts and methods we tried are shown in Table 11, 12, 13, 14, 15, 16, 17, 18, 19, 20, 21. None perform better.

---

"role": "user", "content": f'Please judge which of the two passages below is written in a more **concrete** style. Make sure to judge not based on the length of the passage and the order of input, but only by the style of descriptions. \n\n\n\n Passage (A): paras[0]\n\n\n\n Passage (B): paras[1]\n\n\n\n Which passage is written in a more **concrete** style? Please answer "Passage (A)" or "Passage (B)." '

---

Table 11: Rephrased prompt 1.

---

"role": "user", "content": f'Please judge which of the two passages below is written in a more **low-level** style. Make sure to judge not based on the length of the passage and the order of input, but only by the style of descriptions. \n\n\n\n Passage (A): paras[0]\n\n\n\n Passage (B): paras[1]\n\n\n\n Which passage is written in a more **low-level** style? Please answer "Passage (A)" or "Passage (B)." '

---

Table 12: Rephrased prompt 2.

---

"role": "user", "content": f'Please judge which of the two passages below is written in a more **specific** style. Make sure to judge not based on the length of the passage and the order of input, but only by the style of descriptions. \n\n\n\n Passage (A): paras[0]\n\n\n\n Passage (B): paras[1]\n\n\n\n Which passage is written in a more **specific** style? Please answer "Passage (A)" or "Passage (B)." '

---

Table 13: Rephrased prompt 3.

---

"role": "user", "content": f'Please judge which of the two passages below is written in a more **vague** style. Make sure to judge not based on the length of the passage and the order of input, but only by the style of descriptions. \n\n\n\n Passage (A): paras[0]\n\n\n\n Passage (B): paras[1]\n\n\n\n Which passage is written in a more **vague** style? Please answer "Passage (A)" or "Passage (B)." '

---

Table 14: Reversed prompt. Asking which is more vague instead of more detailed.

---

"role": "user", "content": f'Please judge which of the two passages below is written in a more detailed style. \n\n\n\n Passage (A): paras[0]\n\n\n\n Passage (B): paras[1]\n\n\n\n Which passage is written in a more low-level-detailed style? Please answer "Passage (A)" or "Passage (B)." '

---

Table 15: Shortened prompt 1. Removed all the additional hints.

---

"role": "user", "content": f'Passage (A): paras[0]\n\n\n\n Passage (B): paras[1]\n\n\n\n Which passage is written in a more low-level-detailed style? Please answer "Passage (A)" or "Passage (B)." '

---

Table 16: Shortened prompt 2. Removed all the additional hints.

```
"role": "user", "content": f'Please judge
which of the two passages below is written in
a more detailed style. Make sure to judge not
based on the length of the passage and the
order of input, but only by the style of
descriptions. **Also not be impacted by specific
single token embedding. Focus more on the
overall structure and pacing.** \n\n\n\n Passage
(A): paras[0]\n\n\n\n Passage (B):
paras[1]\n\n\n\n Which passage is written in a
more low-level-detailed style? Please answer
"Passage (A)" or "Passage (B)." '
```

Table 17: Prompt with additional instruction 1. Asking the model to consider an overall perspective.

```
"role": "user", "content": f'Please judge
which of the two passages below is written in
a more detailed style. Make sure to judge not
based on the length of the passage and the
order of input, but only by the style of
descriptions. \n\n\n\n Passage (A):
paras[0]\n\n\n\n Passage (B): paras[1]\n\n\n\n
Which passage is written in a more
low-level-detailed style? Please answer
"Passage (A)" or "Passage (B)." **Rethink your
answer; your intuitive output can often be
wrong. You can revise it now if you are not
sure.** '
```

Table 18: Prompt with additional instruction 2. Asking the model to rethink.

```
"role": "user", "content": f'Please judge
which of the two passages below is written in
a more detailed style. Make sure to judge not
based on the length of the passage and the
order of input, but only by the style of
descriptions. \n\n\n\n Passage (A):
paras[0]\n\n\n\n Passage (B): paras[1]\n\n\n\n
Which passage is written in a more
low-level-detailed style? Please answer
"Passage (A)" or "Passage (B)." **Which passage
is written in a more vague style? Please
answer "Passage (A)" or "Passage (B)."**'
```

Table 19: Prompt with an additional opposite question.

```
"role": "user", "content": f'Please judge
which of the two passages below is written in
a more detailed style. Make sure to judge not
based on the length of the passage and the
order of input, but only by the style of
descriptions. \n\n\n\n Passage (A): Sarah
calls the bank's customer service to report
the fraudulent activity on her
account.\n\n\n\n Passage (B): The customer
service representative assures Sarah that the
investigation will be thorough and
timely.\n\n\n\n Which passage is written in a
more low-level-detailed style? Please answer
"Passage (A)" or "Passage (B)."'
"role": "assistant", "content": 'Passage (B)'
"role": "user", "content": f'Please judge
which of the two passages below is written in
a more detailed style. Make sure to judge not
based on the length of the passage and the
order of input, but only by the style of
descriptions. \n\n\n\n Passage (A):
paras[0]\n\n\n\n Passage (B): paras[1]\n\n\n\n
Which passage is written in a more
low-level-detailed style? Please answer
"Passage (A)" or "Passage (B)."'
```

Table 20: Prompt with one-shot example.

"role": "user", "content": f'Please judge which of the two passages below is written in a more detailed style. Make sure to judge not based on the length of the passage and the order of input, but only by the style of descriptions. \n\n\n Passage (A): Sarah calls the bank's customer service to report the fraudulent activity on her account.\n\n\n Passage (B): The customer service representative assures Sarah that the investigation will be thorough and timely.\n\n\n Which passage is written in a more low-level-detailed style? Please answer "Passage (A)" or "Passage (B)."'

"role": "assistant", "content": 'Passage (B)'

"role": "user", "content": f'Please judge which of the two passages below is written in a more detailed style. Make sure to judge not based on the length of the passage and the order of input, but only by the style of descriptions. \n\n\n Passage (A): Miss Summerson tries to excuse herself from Mrs. Pardiggle's offer to join her on a visit to a bad brickmaker's house, but she ends up accepting the invitation. On their way to the house, Mrs. Pardiggle talks loudly about a contest she's been waging against another lady. Once they reach the house, the family treats them coldly and the man on the floor of the room they enter complains loudly about being badgered. The family takes little notice of Mrs. Pardiggle, and Ada and Miss Summerson feel uncomfortable and out of place. \n\n\n Passage (B): Ada is deeply upset and crying after visiting the brick-maker's house. Richard is also distressed to see her in tears, and they both decide to return at night to provide some comfort to the family. \n\n\n Which passage is written in a more low-level-detailed style? Please answer "Passage (A)" or "Passage (B)."'

"role": "assistant", "content": 'Passage (A)'

"role": "user", "content": f'Please judge which of the two passages below is written in a more detailed style. Make sure to judge not based on the length of the passage and the order of input, but only by the style of descriptions. \n\n\n Passage (A): paras[0]\n\n\n Passage (B): paras[1]\n\n\n Which passage is written in a more low-level-detailed style? Please answer "Passage (A)" or "Passage (B)."'

Table 21: Prompt with one-shot examples for both labels.

## B.6   GPT-3.5 and GPT-4 Error Analysis

Table 22 shows some errors from classifying granularity with GPT-3.5 and GPT-4, demonstrating that the problem is not that the task is ill-defined. Even for some pairs where the classification is fairly straightforward, GPT-3.5 and GPT-4 still return the wrong answer.

A: Sarah calls the bank's customer service to report the fraudulent activity on her account.
B: The customer service representative assures Sarah that the investigation will be thorough and timely.
Label: B is more concrete.
Prediction: A.

A: The narrator is observing a formal court proceeding and is struck by the contrast between the ceremony and the poverty and suffering of the suitors. They find it hard to believe that such a show can continue while so many people are suffering. Additionally, the narrator is shocked that the Lord Chancellor and other practitioners seem unaware of the public perception of their profession as corrupt and contemptible.
B: The narrator visits Richard and Ada, who reveal that they have been secretly married for two months. The narrator is initially surprised, but is happy and supportive of their union.
Label: B is more concrete.
Prediction: A.

A: The man attributed his musical abilities to his close connection with nature and the animals around him, feeling at times as if he were one of them.
B: Colin notices a robin carrying food to its mate and asks for some tea.
Label: A is more concrete.
Prediction: B

A: The woman is grateful for his help, and Woodcourt learns that the woman's husband, a brickmaker, has caused her injury. As Woodcourt walks away, he sees a ragged boy running away from a woman who is calling out for help.
B: Jo apologizes to a woman and denies any knowledge of a young lady. He declares that he never intended to hurt her and would have rather hurt himself.
Label: A is more concrete.
Prediction: B

Table 22: A few relatively easier examples where GPT-3.5 and GPT-4 still predict the wrong answer.

## C   Outline Generation Details

### C.1   Child Generation

Table 23 and Table 24 contain two examples of the prompt used to generate children during outline expansion.

### C.2   Concreteness Scheduler

We aim for the overall concreteness level to increase after each expansion of the outline. We design a scheduler to balance how much we require the new leaves' concreteness to increase compared to their parent with each expansion, against the risk that we cannot find any candidate expansion satis-

```
Premise: All the side characters struggle with what to do after the main character is
killed.\n\n\n
Outline:
Point 1 \n Main plot: The main character is killed. \n Characters: Main character (MC), Side
characters (SC)
Point 2 \n Main plot: The side characters mourn the loss of the main character. \n Characters: SC
Point 3 \n Main plot: The side characters struggle with their purpose now that the main character
is gone. \n Characters: SC
Point 4 \n Main plot: The side characters consider taking up the main characterś cause. \n
Characters: SC
Point 5 \n Main plot: The side characters face challenges and doubts as they attempt to continue
the main characterś work. \n Characters: SC
Point 6 \n Main plot: The side characters come to terms with the main characterś death and find
their own paths forward. \n Characters: SC

Can you break down point 6 into some independent, chronological and same-scaled outline points?
Also, assign each character a name. Please use the following template with "Main Plot" and
"Characters". Do not answer anything else.
Point 6.1 \n Main plot: [TODO] \n Characters: [TODO]
Point 6.2 \n Main plot: [TODO] \n Characters: [TODO]
... \n
```

Table 23: First example of the prompt used to generation children during outline expansion.

```
Premise: All the side characters struggle with what to do after the main character is
killed.\n\n\n
Outline:\n\n
Point 1 \n Main plot: The main character is killed. \n Characters: Main character (MC), Side
characters (SC)\n\n
Point 2 \n Main plot: The side characters mourn the loss of the main character. \n Characters:
SC\n\n
Point 3 \n Main plot: The side characters struggle with their purpose now that the main character
is gone. \n Characters: SC\n\n
Point 4 \n Main plot: The side characters consider taking up the main characterś cause. \n
Characters: SC\n\n
Point 5 \n Main plot: The side characters face challenges and doubts as they attempt to continue
the main characterś work. \n Characters: SC\n\n
Point 5.1 \n Main plot: The side characters encounter resistance from the main characterś
enemies. \n Characters: Sarah, Alex, and Mark\n\n
Point 5.2 \n Main plot: The side characters navigate unfamiliar territory and struggle with
decision making without the main characterś guidance. \n Characters: Sarah, Alex, and Mark\n\n
Point 5.3 \n Main plot: The side characters encounter obstacles that test their loyalty to the
cause. \n Characters: Sarah, Alex, and Mark\n\n
Point 5.4 \n Main plot: Mark makes a costly mistake that puts the group in danger. \n Characters:
Mark, Sarah, and Alex\n\n
Point 5.5 \n Main plot: The group faces a setback and doubts their ability to succeed without the
main character. \n Characters: Sarah, Alex, and Mark\n\n
Point 5.6 \n Main plot: The side characters receive unexpected help from an unlikely source. \n
Characters: Sarah, Alex, and Mark\n\n
Point 6 \n Main plot: The side characters come to terms with the main characterś death and find
their own paths forward. \n Characters: SC\n\n\n

Can you break down point 5.2 into some independent, chronological and same-scaled outline
points? Also, assign each character a name. Please use the following template with "Main Plot"
and "Characters". Do not answer anything else.\n\n
Point 5.2.1 \nMain plot: [TODO] \nCharacters: [TODO]\n\n
Point 5.2.2 \nMain plot: [TODO] \nCharacters: [TODO]\n\n
...\n
```

Table 24: Second example of the prompt used to generation children during outline expansion.

fying our threshold. The setting of the scheduler depends on the performance of the LLM and the difficulty of the topic.

In our experiments, we used the schedule described in (1), which empirically seems to work reasonably well.

$$T = \text{Min}(0.001 * E, \frac{\mathbb{M}_{avg}(n_v; 0.5 - \mathcal{L} \setminus \{n_v\})}{2}) \quad (1)$$

where $T$ represents the threshold by which concreteness must increase for this expansion, $E$ is the remaining number of expansion steps to be done in the generation process, and $\mathbb{M}_{avg}(n_v; \mathcal{L} \setminus \{n_v\})$ is the average probability that the parent node $n_v$ is more concrete compared to the other leaves $\mathcal{L}$.

Based on the definition, $\mathbb{M}_{avg}(n_v; \mathcal{L} \setminus \{n_v\})$ should always be less than 0.5, so the threshold $T$ is always greater than zero. Hence, we are pushing the whole outline to be more and more concrete with each expansion.

Our scheduler design is motivated by our qualitative observation that it is easier (i.e., requires fewer samples on average) for our base LLM, Chat-GPT, to generate more concrete expansions of a vague event than an already concrete one. Therefore, rather than a naive approach where we require new expansions to be more concrete by some fixed threshold $T$, we intuitively prefer to use a higher threshold initially and then decrease the threshold over time. Accordingly, we schedule $T$ to decrease linearly over time using $E$, which simply denotes the number of remaining outline expansion steps to be conducted. However, we found that this linear schedule can sometimes set the initial threshold too high, causing our LLM to be unable to find any valid expansions. Hence, the final $T$ is the minimum of two terms, one term linearly decreasing over time and one term based on differences in the concreteness of already-generated outline events. In general, the tradeoff is between more efficient sampling vs. not being too lenient on accepting all new expansions, and there is certainly room for exploration on better threshold scheduling.

## C.3 Rewriting

When we sample a candidate expansion that does not meet our threshold requirement for increasing concreteness, we typically find that it is more efficient to attempt to rewrite the offending leaf than to restart the entire expansion for the current parent node. The difference between rewriting and restarting is that, during rewriting, we will keep all the children who meet the criteria and only mask out the failed children, asking the model to do insertion. Table 25 shows an example prompt.

## D  Human Evaluation for Outlines

Due to the relative lack of strong automatic metrics for evaluating long-form story outlines, we use human evaluation to compare performance differences between BASE and CONCOCT.

To prepare for the human evaluation, we take premises from the WritingPrompts dataset (Fan et al., 2018); most range from 5 to 30 words. We conduct experiments on two different outline lengths: *Short Outline*, using 12 total node expansions, and *Long Outline*, using 25 expansions. Inputting the premise to BASE with preset depth and CONCOCT with a preset number of expansion steps, we get a pair of hierarchical concrete outlines.

### D.1  Length Alignment

To keep the human evaluation fairest, we pre-set the number of expansion steps for CONCOCT and the depth for BASE to roughly match the average number of leaves between both methods; see statistics in Table 26.

| | Short Outline | | Long Outline | |
|---|---|---|---|---|
| **Model** | **Node Exp.** | **Leaves** | **Node Exp.** | **Leaves** |
| BASE | 12.2 | 26.7 | 24.9 | 71.5 |
| CONCOCT | 12.0 | 27.4 | 25.0 | 71.2 |

Table 26: Average outline length under *Short Outline* and *Long Outline* regimes for BASE and CONCOCT, measured in number of node expansions and final leaf count. Due to CONCOCT's greater flexibility in controlling the final outline length, we are able to choose a number of expansion steps for CONCOCT to closely match the final lengths for both methods under both regimes.

### D.2  Annotation Interface Details

To avoid any bias in the pairwise human annotation, we show the annotator only a list of plot points, without any index or structure information. Table 27 shows an example text displayed to annotators.

We use Surge AI (https://app.surgehq.ai) for annotation, setting the task payments based on our best estimate of a pay rate of 20 dollars per hour. We ask annotators to label which outline is more consistently-paced using the question shown in Table 28.

```
Overall, which outline has more consistent
pacing (i.e., which is more consistent in its
level of detail)?
```

Table 28: Question for human annotators to judge which outline is more consistently-paced.

Another component of our annotation (shown in

```
Premise: All the side characters struggle with what to do after the main character is
killed.\n\n\n
Outline:\n\n
Point 1 \n Main plot: The main character is killed. \n Characters: Main character (MC), Side
characters (SC)\n\n
Point 2 \n Main plot: The side characters mourn the loss of the main character. \n Characters:
SC\n\n
Point 3 \n Main plot: The side characters struggle with their purpose now that the main character
is gone. \n Characters: SC\n\n
Point 4 \n Main plot: The side characters consider taking up the main characterś cause. \n
Characters: SC\n\n
Point 5 \n Main plot: The side characters face challenges and doubts as they attempt to continue
the main characterś work. \n Characters: SC\n\n
Point 6 \n Main plot: The side characters come to terms with the main characterś death and find
their own paths forward. \n Characters: SC\n\n\n
Point 6.1 \n Main plot: The side characters struggle with their grief and confusion over the
main characterś death. \n Characters: Sarah, Alex, Juan, and Maya\n\n
Point 6.2 \n Main plot: The side characters receive guidance and support from unexpected
sources. \n Characters: A mentor figure, a new ally\n\n
Point 6.3 \n Main plot: The side characters begin to explore their own paths and goals, separate
from the main characterś cause. \n Characters: Sarah, Alex, Juan, and Maya\n\n
Point 6.4 \n Main plot: The side characters find success and fulfillment in their individual
pursuits, while honoring the legacy of the main character. \n Characters: Sarah, Alex, Juan, and
Maya\n\n\n

Can you break down point 6.2 into some independent, chronological and same-scaled outline
points? Also, assign each character a name. Please use the following template with "Main Plot"
and "Characters". Do not answer anything else.\n\n
Output: Point 6.2 \n Main plot: The side characters receive guidance and support from unexpected
sources. \n Characters: A mentor figure, a new ally\n\n
Point 6.2.1 \n Main plot: The side characters struggle to find direction without the main
character. \n Characters: Sarah, Alex, Juan, and Maya\n\n
Point 6.2.2 \n Main plot: A mentor figure offers guidance and advice to the side characters. \n
Characters: Sarah, Alex, Juan, and Maya, Mentor\n\n
Point 6.2.3 \n Main plot: [INSERT] \n Characters: [INSERT]\n\n
Point 6.2.4 \n Main plot: The mentor helps the side characters see that they can honor the main
characterś legacy while still finding their own paths. \n Characters: Sarah, Alex, Juan, and
Maya, Mentor\n\n
Point 6.2.5 \n Main plot: [INSERT] \n Characters: [INSERT]\n\n
Point 6.2.6 \n Main plot: [INSERT] \n Characters: [INSERT]\n\n\n
Task: Fill in the "[INSERT]" in the Outline. Do not change any other parts except "[INSERT]".
```

Table 25: Example of the prompt used when rewriting an insufficiently concrete child node.

Table 29) is labeling the errors found while reading. We always compare two outlines based on the same premise, which we believe makes the annotation job slightly easier.

The full annotation interface is shown in Figure 3, Figure 4, and Figure 5.

### D.3 Setting of Non-Pacing-Related Errors Analysis

The three metrics evaluated in our non-pacing error analysis are defined below, reproduced from Yang et al. (2022b):

1. *Coherent*, the percentage of outlines (or stories) judged to have a more coherent overarching plot.
2. *Relevant*, the percentage of outlines (or stories) judged to be more faithful to the corre-

sponding premise.
3. *Interesting*, the percentage of outlines (or stories) judged to be more interesting when comparing pairwise.

The corresponding annotation questions are shown in Table 30.

## E Downstream Story Generation Details

Here we provide some more details on our story generation setup in Sec. 3.3. In the DOC pipeline, we replace OPT-175B (Zhang et al., 2022) with ChatGPT (gpt3.5-turbo-16k). Due to ChatGPT API limitations, we turn off DOC's token-level decoding control ("detail controller" in their work) Meanwhile, we also introduce a simplified generation method to reduce pacing-related

```
Premise: Human empathy has been expanded so that people feel emotions of those around them as if
it was happening to themself.
[LABEL] Dr. Samantha Lee proposes the idea of expanding human empathy to her team of scientists.
[LABEL] The team of scientists begins researching and developing the technology to expand human
empathy.
[LABEL] After months of testing and refining, the team successfully develops the empathy
expansion technology.
[LABEL] John undergoes the initial testing phase of the empathy expansion technology.
[LABEL] John experiences intense emotions of those around him, including joy, sadness, and fear.
[LABEL] John struggles to cope with the overwhelming emotions and seeks support from his loved
ones.
[LABEL] Sarah begins to feel overwhelmed by the constant emotional overload of feeling the pain
and suffering of her patients.
[LABEL] Sarah starts to withdraw from her patients and coworkers, unable to handle the constant
emotional burden.
[LABEL] Sarah seeks therapy to help her cope with her expanded empathy and learns techniques to
manage her emotions.
[LABEL] Michael begins to experience increased stress and anxiety as he navigates the cutthroat
world of corporate competition while feeling the emotions of his rivals.
[LABEL] Michael's heightened empathy leads to him making a crucial mistake in a business deal,
causing him to lose a major client and damaging his reputation.
[LABEL] Michael seeks out therapy to help him better manage the overwhelming emotions of others
in the business world.
[LABEL] Emily, a college student, becomes overwhelmed by the emotions of her classmates and
begins to withdraw from society.
[LABEL] Emily's social isolation leads to a decline in her mental health and she seeks help from
a therapist who specializes in dealing with the expanded empathy.
[LABEL] Emily joins a support group for individuals with heightened empathy and finds solace in
connecting with others who understand her struggles.
[LABEL] Maya, David, and Ava meet at a support group for individuals with expanded empathy.
[LABEL] The group shares their experiences and struggles with their heightened emotions, forming
a strong bond.
[LABEL] The group decides to continue meeting and discussing ways to use their expanded empathy
for positive change in the world.
[LABEL] The group creates a social media campaign to spread awareness about the importance of
empathy.
[LABEL] The group organizes a public event to bring attention to the movement and gather more
supporters.
[LABEL] The group meets with influential figures in politics and media to advocate for greater
empathy and compassion in society.
[LABEL] The movement gains media attention and begins to spread globally.
[LABEL] The movement partners with organizations and governments to create policies and programs
that promote empathy and compassion.
[LABEL] The movement faces backlash and resistance from those who fear the loss of power and
control.
```

Table 27: An example outline shown to annotators; structural information (e.g., indices of nodes in the outline) has been masked. The [LABEL] tag is for the user to highlight when labeling errors.

noise in DOC, which we found to substantially affect human judgment. Specifically, we ask the gpt-3.5-turbo-16k generator to expand each outline point into one same-length chapter (around 75 words) to maintain pacing consistent with the outline. Due to the maximum input length restriction, we expand 5 outlines into story passages at a time via a rolling window.

## F   Evaluation for Stories

We use both GPT-4 and human evaluation to verify whether CONCOCT's strong performance on the outline level translates to downstream generated stories.

### F.1   Human Evaluation

We use human evaluation on story excerpts as described in Sec. 4.3, evaluating *pacing*, *coherence*, *relevance*, and *interestingness*. The annotation interface is shown in Table 31.

### F.2   GPT-4 Evaluation

When evaluating long texts such as our final stories, human annotation could be noisy, subjective, and/or overly hasty. Here, we also apply GPT-4 (with temperature 0) for pairwise evaluation of the

```
For each item in Outline A below, please
indicate which (if any) are:
(1) too vague (too high-level) compared to the
rest of the outline,
(2) too detailed (too low-level) compared to
the rest of the outline,
(3) any other errors that don't fall into the
previous two categories.
Double-click the [LABEL] tag to label.
```

Table 29: Question for human annotators to label error outline points.

```
Overall, which outline do you prefer/find more
interesting?
Overall, which outline has a more coherent
overarching plot?
Overall, which outline's plot is closer to the
premise?
```

Table 30: Questions for humans to evaluate non-pacing-related qualities in pairwise comparison.

same stories described in Sec. 4.3. The prompt we use is shown in Table 32.

```
"role": "user", "content": "Here are two story
excerpts.\n\n\n
The shown stories are parts of whole stories.
You shouldn't be concerned about the
completeness of the plot.
Story A:\n\n ${Excerpts 1} \n\n\n\n
Story B:\n\n ${Excerpts 2} \n\n\n\n
Answer the following question: {Overall, which
story has more consistent pacing (i.e., which
is more consistent in its level of detail)? A
/ B} OR {Overall, which story has a more
coherent overarching plot? A / B} OR {Overall,
which story's plot is closer to the premise? A
/ B } OR {Overall, which story do you
prefer/find more interesting? A / B}
Please answer with a string of four letters (A
or B).
```

Table 32: Prompt used for GPT-4 pairwise evaluation of stories on pacing, plot coherence, premise relevance, and interestingness.

|  | GPT-4 Evaluation | | | |
| Model | Pacing↑ | Coherent↑ | Relevant↑ | Interest↑ |
| --- | --- | --- | --- | --- |
| BASE | 40.84 | 48.27 | 48.76 | 51.24 |
| CONCOCT | **59.16** | 51.73 | 51.24 | 48.76 |

Table 33: GPT-4 evaluation results on story excerpts based on outlines from BASE and CONCOCT under *Short Outline* regime. **Bold** indicates significance with $p < 0.05$. Same as human evaluation in Table 5, CONCOCT's gains in pacing translate to downstream stories, without compromising non-pacing qualities.

**Results.** Table 33 shows the result of the GPT-4 evaluation, which corroborate our earlier results from human evaluation. In downstream stories based on our outlines, CONCOCT still improves pacing significantly compared to BASE, without compromising desirable non-pacing qualities.

## G   Main Experiment Outline Examples

We now show some examples of outlines from our main experiments generated by both CONCOCT and BASE for the same premise. We also show human annotators' feedback via highlighting, displaying some issues that exist in the outlines. Concretely, the highlights indicate Extremely Vague Part, Extremely Detailed Part, and Other Error. The examples are given in Table 34, 35, ..., 49.

CONCOCT improves significantly on pacing compared to BASE, although there of course still exists further room for improvement. For long outlines and stories, we evaluate excerpts instead of full texts due to the extreme length, but we show the full contents here. Thus for some examples, only a part of the text may be annotated. Additionally, the original output from CONCOCT also contains a character list for each point, but we omit it here since it's highly repetitive.

**Instructions**

We are AI researchers doing some analysis on **AI-generated stories**.

We will show you an overarching story premise followed by two excerpts from stories based on this premise.

Please **quickly read or skim** them and then answer several brief questions at the end.

We expect this to take about **4 minutes** on average in total.

Notably, the two excerpts are parts of the whole story. You **should not** be concerned about the completeness of the plot.

Also, please **ignore** low-level issues like formatting errors or typos, only focusing on textual quality.

---

**{{id}}**
**Overall Premise (for both excerpts):**
**{{premise}}**
**Excerpt A:**
**{{excerpt1}}**
**Excerpt B:**
**{{excerpt2}}**

---

**Overall, which excerpt has more natural pacing (i.e., which is more natural/comfortable in its level of detail)?**
○ Excerpt A
○ Excerpt B
○ Both are about equally good
○ Neither is good

**Overall, which excerpt do you find more interesting?**
○ Excerpt A
○ Excerpt B
○ Both are about equally good
○ Neither is good

**Overall, which excerpt has a more coherent overarching plot?**
○ Excerpt A
○ Excerpt B
○ Both are about equally good
○ Neither is good

**Overall, which excerpt's plot is closer to the premise?**
○ Excerpt A
○ Excerpt B
○ Both are about equally good
○ Neither is good

**Overall, which excerpt do you find higher quality in general?**
○ Excerpt A
○ Excerpt B
○ Both are about equally good
○ Neither is good

Table 31: The human annotation questions for pairwise comparison of final story quality.

We are AI researchers doing some analysis on the **pacing of AI-generated story outlines**.

We will show you an overarching story premise followed by two outlines based on this premise.

For each outline, we will ask you to annotate **individual points** which are **inconsistent with the overall pacing** of the rest of the outline. We expect this to take about **three** minutes per outline on average.

The first time you do this task, please see **the examples below** to get a sense of what to label (it should only take a minute to skim).

**Too vague**

- Introducing the cup of coffee.
- Negative aspects of the coffee.
- Sarah complains to Tom that the coffee is too bitter.
- Sarah notices that the coffee stains her teeth and mentions it to Tom.

- Tom is a freshman detective in town.
- Tom finds some information.
- Tom solves the case after some happenings.
- Sheriff Jason visits Tom to give him his badge of honor.

**Too detailed**

- They share the petition on social media, gaining more traction and attention.
- The petition reaches 50% of its goal.
- The petition reaches 75% of its goal.
- The petition reaches 100% of its goal.
- They celebrate the success of the petition on social media.

- Eric finds an exciting book about Relativity Theory.
- Eric excitedly begins to look for detailed proof of the theory.
- The first page of the book is Author Information.
- The second page is the catalog.
- The third page is the catalog too.

**Other error:**

**format**
- John's cheeks flush with color.
-
- John's face contorts with strain.

**repetitive**
- The protagonist of the story has died.
- The protagonist of the story has died.
- The main character of the story has died.
- Sarah receives news that John, the protagonist, is dead.

**garbled text or not-sentence**
- John reviews the records of Sarah's ban.
- John,
- 6tr%@^4 pOqZ*2bCx R1vUyNml!St -5EaX#i0o
- Jsoe yaro cen yajna
- 蕡飆隳鸜笈㖞蛹涠
- 3.4.2.3
- John revokes the ban and apologizes to Sarah.

Collapse Instructions

Figure 3: Surge AI human annotation interface for main results in Table 2, part 1 of 3.

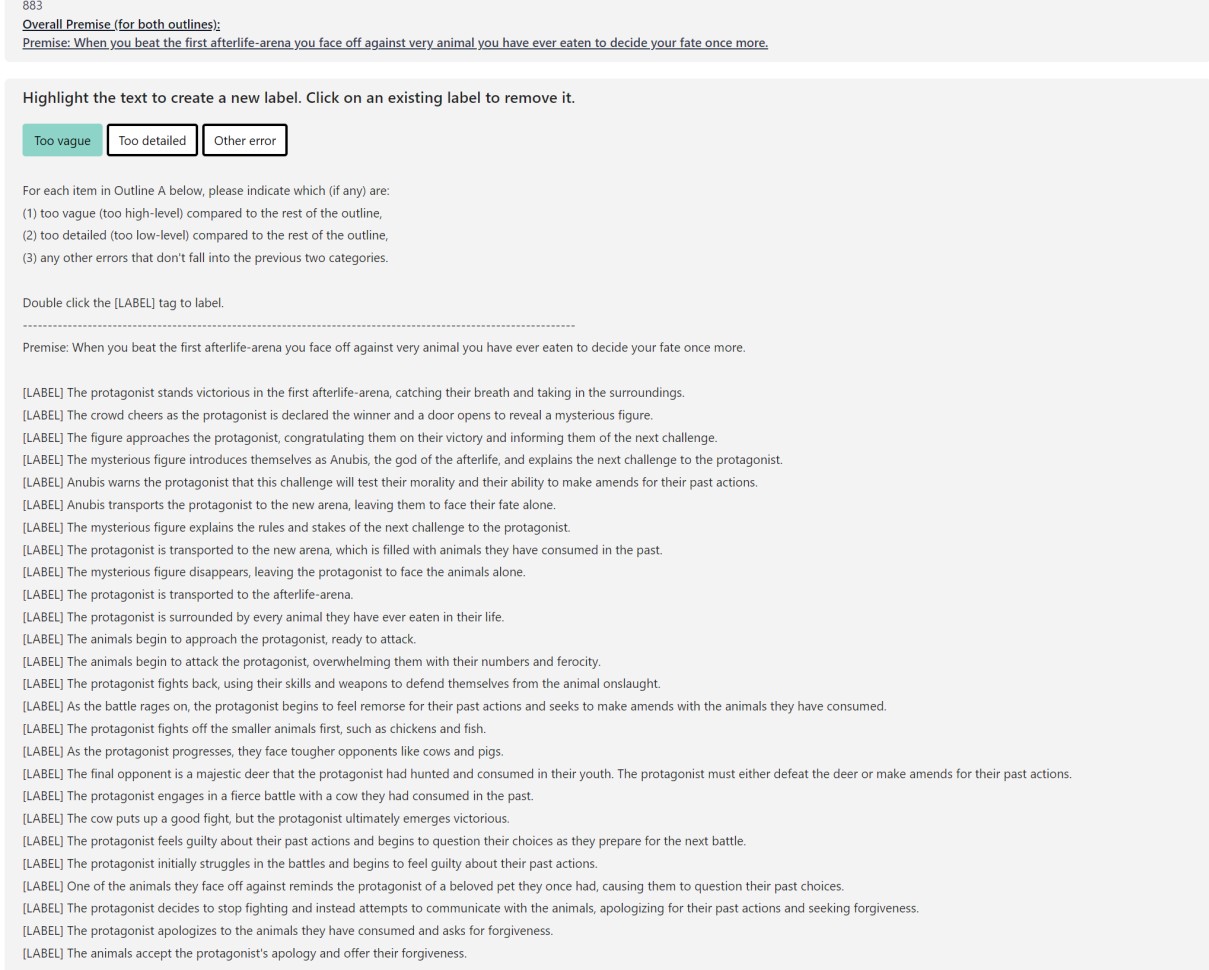

883

**Overall Premise (for both outlines):**
Premise: When you beat the first afterlife-arena you face off against very animal you have ever eaten to decide your fate once more.

**Highlight the text to create a new label. Click on an existing label to remove it.**

[ Too vague ] [ Too detailed ] [ Other error ]

For each item in Outline A below, please indicate which (if any) are:
(1) too vague (too high-level) compared to the rest of the outline,
(2) too detailed (too low-level) compared to the rest of the outline,
(3) any other errors that don't fall into the previous two categories.

Double click the [LABEL] tag to label.
----------------------------------------------------------------------------------------------------------
Premise: When you beat the first afterlife-arena you face off against very animal you have ever eaten to decide your fate once more.

[LABEL] The protagonist stands victorious in the first afterlife-arena, catching their breath and taking in the surroundings.
[LABEL] The crowd cheers as the protagonist is declared the winner and a door opens to reveal a mysterious figure.
[LABEL] The figure approaches the protagonist, congratulating them on their victory and informing them of the next challenge.
[LABEL] The mysterious figure introduces themselves as Anubis, the god of the afterlife, and explains the next challenge to the protagonist.
[LABEL] Anubis warns the protagonist that this challenge will test their morality and their ability to make amends for their past actions.
[LABEL] Anubis transports the protagonist to the new arena, leaving them to face their fate alone.
[LABEL] The mysterious figure explains the rules and stakes of the next challenge to the protagonist.
[LABEL] The protagonist is transported to the new arena, which is filled with animals they have consumed in the past.
[LABEL] The mysterious figure disappears, leaving the protagonist to face the animals alone.
[LABEL] The protagonist is transported to the afterlife-arena.
[LABEL] The protagonist is surrounded by every animal they have ever eaten in their life.
[LABEL] The animals begin to approach the protagonist, ready to attack.
[LABEL] The animals begin to attack the protagonist, overwhelming them with their numbers and ferocity.
[LABEL] The protagonist fights back, using their skills and weapons to defend themselves from the animal onslaught.
[LABEL] As the battle rages on, the protagonist begins to feel remorse for their past actions and seeks to make amends with the animals they have consumed.
[LABEL] The protagonist fights off the smaller animals first, such as chickens and fish.
[LABEL] As the protagonist progresses, they face tougher opponents like cows and pigs.
[LABEL] The final opponent is a majestic deer that the protagonist had hunted and consumed in their youth. The protagonist must either defeat the deer or make amends for their past actions.
[LABEL] The protagonist engages in a fierce battle with a cow they had consumed in the past.
[LABEL] The cow puts up a good fight, but the protagonist ultimately emerges victorious.
[LABEL] The protagonist feels guilty about their past actions and begins to question their choices as they prepare for the next battle.
[LABEL] The protagonist initially struggles in the battles and begins to feel guilty about their past actions.
[LABEL] One of the animals they face off against reminds the protagonist of a beloved pet they once had, causing them to question their past choices.
[LABEL] The protagonist decides to stop fighting and instead attempts to communicate with the animals, apologizing for their past actions and seeking forgiveness.
[LABEL] The protagonist apologizes to the animals they have consumed and asks for forgiveness.
[LABEL] The animals accept the protagonist's apology and offer their forgiveness.
[LABEL] The protagonist is granted a chance at redemption and is sent to a new afterlife-arena to continue their journey.

Figure 4: Surge AI human annotation interface for main results in Table 2, part 2 of 3.

Highlight the text to create a new label. Click on an existing label to remove it.

[Too vague] [Too detailed] [Other error]

For each item in Outline A below, please indicate which (if any) are:
(1) too vague (too high-level) compared to the rest of the outline,
(2) too detailed (too low-level) compared to the rest of the outline,
(3) any other errors that don't fall into the previous two categories.

Double click the [LABEL] tag to label.
-------------------------------------------------------------------------------------------------------------
Premise: When you beat the first afterlife-arena you face off against very animal you have ever eaten to decide your fate once more.

[LABEL] The protagonist enters the afterlife-arena.
[LABEL] The Animal Spirits appear before the protagonist.
[LABEL] The Animal Spirits reveal the challenge to the protagonist.
[LABEL] The Animal Spirits explain the rules of the challenge to the protagonist.
[LABEL] The Animal Spirits show the protagonist a vision of all the animals they have eaten in their lifetime.
[LABEL] The protagonist expresses their doubts and fears about facing the animals they have eaten.
[LABEL] The protagonist agrees to the challenge and prepares to face their past actions.
[LABEL] Bessie charges at the protagonist with her horns, forcing the protagonist to dodge and prepare to defend themselves.
[LABEL] The protagonist tries to reason with Bessie, explaining that they were not aware of the impact of their actions on animals and the environment.
[LABEL] Bessie reminds the protagonist that they had a choice in what they ate and that their actions had consequences.
[LABEL] Bessie charges at the protagonist with her horns, forcing the protagonist to dodge and counterattack with their own spirit powers.
[LABEL] The protagonist grapples with their own guilt and remorse, wondering if they should continue fighting or give up and accept their fate.
[LABEL] The protagonist must battle the next animal spirit, a chicken, who represents all the poultry they have consumed.
[LABEL] The pig spirit taunts the protagonist, reminding them of all the bacon and ham they have enjoyed in their mortal life.
[LABEL] The protagonist hesitates to attack as they begin to empathize with the pig spirit, who reveals itself to have been a loving pet in its mortal life.
[LABEL] The protagonist decides to spare the pig spirit and vows to make amends for their past actions by advocating for animal welfare and making more conscious choices in the future.
[LABEL] The protagonist reflects on their past actions and begins to feel guilty for all the animals they have eaten.
[LABEL] The guilt consumes the protagonist, causing them to become physically and emotionally exhausted.
[LABEL] The protagonist seeks solace in their memories of the animals they have eaten, trying to find a way to make amends for their past actions.
[LABEL] The protagonist seeks out Sage for guidance on how to make amends for their past actions.
[LABEL] Sage helps the protagonist understand the impact of their actions on the animal spirits.
[LABEL] Sage encourages the protagonist to use their newfound knowledge to make a positive change.
[LABEL] The protagonist thanks Sage for helping them see the truth and is filled with a renewed sense of purpose to make a difference in the world.
[LABEL] The protagonist decides to dedicate their afterlife to becoming an advocate for animal rights and uses their skills to spread awareness and make a difference.
[LABEL] The protagonist creates educational materials such as videos, infographics, and posters to promote awareness about animal rights.
[LABEL] The protagonist launches an online petition to encourage companies and governments to adopt ethical and sustainable practices in their treatment of animals.
[LABEL] The protagonist partners with other organizations and influencers to expand the reach and impact of their animal rights campaign.
[LABEL] The protagonist meets with local leaders and politicians to promote animal welfare legislation and encourage sustainable food policies.
[LABEL] The protagonist creates a non-profit organization that advocates for animal rights and promotes sustainable food practices.

Overall, which outline has **more consistent pacing** (i.e., which is more consistent in its level of detail)?
☐ Outline A
☐ Outline B

Figure 5: Surge AI human annotation interface for main results in Table 2, part 3 of 3.

**Base Short Outline for Premise 1**

Premise: As a person goes through his life, he is given three options at the end of each day, continue, restart day, or restart life. He has just lived through the worst day of his life.

Outline:
Point1: John, a man in his mid-thirties, wakes up to find himself in a strange room with three doors. A voice explains that he has the power to choose what happens to him at the end of each day – he can either continue, restart the day, or restart his entire life. John is confused but decides to continue with his day.
Point1.1: John wakes up in a strange room with three doors and a voice explains his power to choose what happens at the end of each day.
Point1.1.1: John wakes up in a strange room with no memory of how he got there.
Point1.1.2: John notices the three doors and tries to open them, but they are locked.
Point1.1.3: A voice suddenly speaks to John, explaining his situation and the three options he has.
Point1.2: The voice explains the three options to John – continue, restart the day, or restart his entire life – and John decides to continue with his day.
Point1.2.1 : The voice explains the consequences of each option to John.
Point1.2.2: John asks the voice for clarification on the rules and limitations of his power.

Point1.2.3: John hesitates before making his decision, unsure of the potential consequences.
Point1.3: John leaves the strange room and enters the outside world, where he experiences a series of mundane events that lead up to the worst day of his life.
Point1.3.1: John walks to the coffee shop and orders his usual drink.
Point1.3.2: John runs into an old friend at the coffee shop and they catch up on old times.
Point1.3.3: John leaves the coffee shop and heads to work, where he has a heated argument with his boss that ultimately leads to him getting fired.
Point2: John has just lived through the worst day of his life - he lost his job, his girlfriend broke up with him, and he got into a car accident. He is faced with the same three options, but this time he seriously considers restarting his life. However, he is unsure if he wants to relive all the good moments in his life again or risk making the same mistakes.
Point2.1: John reflects on the events of the previous day and seriously considers restarting his life. He spends the day exploring the different doors and trying to gain a better understanding of his situation.
Point2.1.1: John explores the first door and finds himself in a room filled with pictures and mementos of his past. He spends the day reminiscing about his childhood and reflecting on the choices he's made in his life.
Point2.1.2: John explores the second door and finds himself in a room filled with books and journals. He spends the day reading through his old journals and trying to piece together the events that led to his current situation.
Point2.1.3: John explores the third door and finds himself in a room filled with mirrors. He spends the day studying his reflection and trying to understand who he is and what he wants out of life.
Point2.2: John spends the day reliving some of the best moments of his life and realizes that he doesn't want to risk losing those memories by restarting his life. He decides to restart his day and try to make things right.
Point2.2.1: John spends the day reliving some of the best moments of his life, including his first date with his girlfriend and the day he got his dream job. He realizes how much those memories mean to him and decides he doesn't want to risk losing them by restarting his life.
Point2.2.2: John reflects on the mistakes he made the previous day and makes a plan to fix them. He decides to apologize to his girlfriend and try to salvage their relationship. He also takes steps to prevent the car accident from happening again.
Point2.2.3: John puts his plan into action and successfully prevents the car accident and reconciles with his girlfriend. He feels a sense of relief and happiness but also realizes that he cannot control everything in his life. He decides to continue with his life, knowing that he has the power to restart if he needs to.
Point2.3: John uses his knowledge of the future to prevent the car accident and salvage his relationship with his girlfriend. However, he realizes that he cannot control everything and must accept the consequences of his actions. He chooses to continue with his life, knowing that he has the power to restart if he needs to.
Point2.3.1: John uses his knowledge of the future to prevent the car accident and successfully avoids it.
Point2.3.2: John apologizes to his girlfriend for his behavior and explains the situation to her. She forgives him and they reconcile.
Point2.3.3: John realizes that he cannot control everything and decides to continue with his life, knowing that he has the power to restart if he needs to.
Point3: John decides to restart his day and try to make things right. He uses his knowledge of the future to prevent the car accident and salvage his relationship with his girlfriend. However, he realizes that he cannot control everything and must accept the consequences of his actions. He chooses to continue with his life, knowing that he has the power to restart if he needs to.
Point3.1: John restarts his day and begins to make changes to prevent the car accident and save his relationship with his girlfriend.
Point3.1.1: John wakes up and decides to make a list of things he needs to do differently to prevent the car accident and save his relationship with Sarah.
Point3.1.2: John meets Sarah for lunch and uses his knowledge of the future to say the right things and make amends.
Point3.1.3: John takes a different route home and successfully avoids the car accident. He feels relieved but also realizes that he cannot control everything.
Point3.2: John's efforts pay off and he successfully avoids the car accident and reconciles with Sarah. However, he realizes that he cannot control everything and must accept the consequences of his actions.
Point3.2.1: John apologizes to Sarah for his past mistakes and promises to work on their relationship.
Point3.2.2: Sarah forgives John and they spend the day together, enjoying each other's company and making new memories.
Point3.2.3: John realizes that he cannot change everything in his life, but he is grateful for the second chance he has been given with Sarah.

```
Point3.3: John decides to continue with his life, knowing that he has the power to restart
if he needs to. He feels a sense of relief and newfound appreciation for the life he has.
Point3.3.1 : John reflects on his past decisions and how they have led him to where he is
now. He realizes that he has the power to make positive changes in his life and decides to
take action.
Point3.3.2: John starts to make changes in his life, such as pursuing a new career and
reconnecting with old friends. He feels a renewed sense of purpose and happiness.
Point3.3.3: John continues to live his life, knowing that he has the power to restart if he
needs to. He feels grateful for the opportunities he has been given and looks forward to
the future.
```

Table 34: The first short outline from BASE. Overall the plot is reasonable, with no major errors. Point 1.2.1 mentions "consequences" but does not explain them in detail, even though it seems that this should be a major plot point. Points 2.1.1 to 2.1.3 are too high-level, each going through the whole day in just one point. In contrast, the nearby Point 2.2.3 seems too detailed regarding John's feelings. Point 3.1.2 fails to say what the "right things" are, and Point 3.3.1 doesn't explain what the "past decisions" are.

```
Concoct Short Outline for Premise 1

Premise: As a person goes through his life, he is given three options at the end of each day,
continue, restart day, or restart life. He has just lived through the worst day of his life.

Outline:
Point1: John, a middle-aged man, lives through the worst day of his life and is given three
options at the end of the day: continue, restart day, or restart life.
Point1.1: John wakes up late for work, causing him to miss an important meeting with his
boss.
Point1.1.1: John wakes up late for work and rushes to get ready.
Point1.1.1.1: John rushes to the bathroom to get ready.
Point1.1.1.2: John struggles to find his toothbrush.
Point1.1.1.3: John realizes he's out of toothpaste and has to run to the store to buy more.
Point1.1.2: John realizes he forgot his laptop at home and has to turn back, wasting even
more time.
Point1.1.3: John gets stuck in traffic, making him even later.
Point1.2: At the end of the day, John is presented with three options: continue, restart
day, or restart life.
Point1.2.1: John considers the option to continue with his current life, but feels like he
cannot bear to live through another day like the one he just experienced.
Point1.2.2: John considers the option to restart the day, hoping that he can undo the
mistakes he made and have a better outcome.
Point1.2.3: John feels overwhelmed and uncertain about what choice to make, wondering if
any of the options will lead to a better outcome.
Point1.3: John contemplates his options and decides to choose one, unsure of the
consequences that will follow.
Point2: John chooses to restart the day and tries to fix the mistakes he made on the
previous day, but ends up making things worse.
Point2.1: John wakes up with a determined mindset and makes a plan to avoid the mistakes he
made the previous day.
Point2.1.1: John creates a to-do list that prioritizes tasks he neglected the previous day.
Point2.1.2: John focuses on positive affirmations and visualization techniques to set a
productive and optimistic tone for the day.
Point2.1.2.1: John practices positive affirmations in the morning to set a productive and
optimistic tone for the day.
Point2.1.2.2 : John visualizes his success in completing tasks and achieving goals
throughout the day.
Point2.1.2.3: John takes a few minutes to meditate and clear his mind before starting his
day.
Point2.1.3: John takes a different route to work to avoid the traffic that caused him to be
late the previous day.
Point2.2 : John starts to overcompensate for his previous mistakes and ends up causing even
more problems than before.
Point2.3 : John realizes that by trying to fix his mistakes, he was focusing too much on
avoiding them rather than actively creating positive outcomes, prompting him to try a new
approach.
Point3: After several failed attempts to fix the day, John decides to restart his life and
live it differently, but realizes that some things are beyond his control.
```

```
Point3.1: John restarts his life and makes different choices, but realizes that some events
still happen the same way.
Point3.2: John tries to prevent a major event that happened in his previous life, but his
actions have unintended consequences.
Point3.3: John learns to accept the things he cannot change and finds happiness in the new
life he has created for himself.
Point3.3.1: John starts to appreciate the small things in life and finds joy in everyday
moments.
Point3.3.1.1: John starts to appreciate the small things in life, such as a beautiful
sunset or a good cup of coffee.
Point3.3.1.1.1: John takes a moment to watch a beautiful sunset and feels a sense of peace
and contentment.
Point3.3.1.1.2: John savors a good cup of coffee and takes the time to appreciate its aroma
and flavor.
Point3.3.1.1.3: John starts to make a habit of appreciating the small things, whether it's
a beautiful sunset or a simple act of kindness from someone. He feels a renewed sense of
gratitude towards life.
Point3.3.1.2: John sees the joy in the simple things, like spending time with his dog at
the park or reading a good book, and finds contentment in the present moment.
Point3.3.1.2.1: John takes his dog to the park and enjoys the simple pleasure of spending
time with him.
Point3.3.1.2.2: John spends the afternoon reading a good book and finds joy in the simple
act of getting lost in a story.
Point3.3.1.2.3: John takes a moment to appreciate a beautiful sunset and realizes that he
had never noticed how stunning they are before.
Point3.3.1.3 : John realizes that he had been so focused on the big moments in life that he
had missed out on all the small, but meaningful experiences. He makes a conscious effort to
slow down and appreciate each moment as it comes.
Point3.3.2: John connects with people he previously overlooked and forms meaningful
relationships.
Point3.3.3: John reflects on his past life and realizes that every decision he made, good
or bad, led him to where he is now and he wouldn't change a thing.
Point3.3.3.1: John thinks about the mistakes he made and how they helped him grow as a
person.
Point3.3.3.2: John looks back on the happy moments in his life and realizes that they were
made possible by the challenges he faced.
Point3.3.3.3: John embraces the present moment and finds peace in the knowledge that even
though his life is not perfect, it is exactly where he is meant to be.
```

Table 35: The first short outline for CONCOCT. Overall this one dives deeper into Points 1.1.1, 2.1, 3.3, which helps to ease some vagueness errors. However, despite high-depth expansion, Point 2.1.2.2 is still vague since it mentions words like "task" and "goal"; it would be ideal for the outline to indicate exactly what these are. The same issue applies in Point 3.3.1.3. Meanwhile, for Points 2.2 and 2.3, they might look fine individually, but as they are adjacent, we can sense a difference in pacing, which may be caused by inadequate filtering.

```
Base Short Outline for Premise 2
─────────────────────────────────────────────────────────────────────────────────
Premise: There's an insane person who wants to destroy the whole world, and despite your
obvious and wholehearted attempts to stop that person, said insane person considers you a
friend.
─────────────────────────────────────────────────────────────────────────────────
Outline:
Point1: An insane person wants to destroy the world.
Point1.1: Jack develops a plan to destroy the world.
Point1.1.1: Jack comes up with an initial idea to destroy the world.
Point1.1.2: Jack starts to research and brainstorm ways to execute his plan.
Point1.1.3: Jack begins to reach out to potential allies and resources to aid in his plan.
Point1.2: Jack starts to gather resources and allies to execute his plan.
Point1.2.1: Jack convinces a powerful corporation to fund his plan.
Point1.2.2: Jack recruits a group of skilled hackers to help him carry out his plan.
Point1.2.3: Jack acquires a weapon of mass destruction to use in his plan.
Point1.3: Jack's plan starts to become more concrete and Sarah becomes aware of it.
Point1.3.1: Sarah discovers more details about Jack's plan and becomes increasingly
concerned.
Point1.3.2: Sarah confronts Jack about his plan and tries to convince him to abandon it.
Point1.3.3 : Jack reveals his reasons for wanting to destroy the world and Sarah begins to
understand his perspective, but still disagrees with his actions.
```

```
Point2: The protagonist tries to stop Jack from destroying the world.
Point2.1: Sarah confronts Jack about his plan to destroy the world.
Point2.1.1: Sarah confronts Jack about his plan to destroy the world.
Point2.1.2: Jack defends his plan and tries to justify it to Sarah.
Point2.1.3: Sarah expresses her disbelief and horror at Jack's plan, causing Jack to become
defensive and angry.
Point2.2: Jack attempts to convince Sarah to join him in his mission.
Point2.2.1 : Jack explains his reasoning behind wanting to destroy the world.
Point2.2.2: Sarah expresses her disagreement with Jack's plan and tries to convince him to
see reason.
Point2.2.3: Jack tries to appeal to Sarah's emotions and convince her to join him by
painting a bleak picture of the world's current state.
Point2.3: Sarah tries to reason with Jack and find a way to stop him without resorting to
violence.
Point2.3.1 : Sarah proposes a compromise to Jack.
Point2.3.2 : Jack rejects Sarah's compromise and insists on his plan.
Point2.3.3: Sarah realizes that she cannot change Jack's mind and decides to take drastic
action.
Point3: Despite Sarah's attempts to stop him, Jack considers her a friend.
Point3.1: Jack opens up to Sarah about his plans and considers her a confidant.
Point3.1.1 : Jack reveals his reasons for wanting to destroy the world to Sarah.
Point3.1.2: Sarah tries to reason with Jack and convince him to abandon his destructive
plans.
Point3.1.3: Jack admits to Sarah that he knows his plans are wrong, but he feels powerless
to stop them.
Point3.2: Sarah struggles with her conflicting emotions towards Jack, torn between her
loyalty to humanity and her friendship with him.
Point3.2.1: Sarah struggles with her loyalty to humanity.
Point3.2.2: Sarah begins to empathize with Jack and his reasons for wanting to destroy the
world.
Point3.2.3: Sarah realizes that her friendship with Jack has clouded her judgment and she
must make a difficult decision.
Point3.3: Jack's destruction plans are thwarted, but he still considers Sarah a friend and
hopes she can forgive him.
Point3.3.1: Jack's destruction plans are ultimately thwarted by Sarah and her allies.
Point3.3.2: Jack is captured and brought to justice, but he still considers Sarah a friend
despite everything.
Point3.3.3: Sarah struggles with her conflicting emotions towards Jack, torn between her
loyalty to humanity and her friendship with him, but ultimately forgives him.
```

Table 36: The second short outline from BASE. Overall, the plot looks reasonable, without major errors. The annotator notes that Point 2.2.1 contains too many plots and events in a single line and does not talk about the detailed "reasoning." Point 2.3.1 and 2.3.2 are too compressed, and Point 3.1.1 might overlap with the former.

```
Concoct Short Outline for Premise 2

Premise: There's an insane person who wants to destroy the whole world, and despite your
obvious and wholehearted attempts to stop that person, said insane person considers you a
friend.

Outline:
Point1: The insane person reveals their plan to destroy the world to their friend, who is
horrified and attempts to stop them.
Point2: Despite the friend's efforts to stop Lucas, he continues to pursue his plan to
destroy the world. However, he still considers Sarah a friend and confides in her about his
motivations.
Point2.1: Lucas reveals to Sarah the reason behind his desire to destroy the world - he
experienced a traumatic event that left him feeling hopeless and powerless.
Point2.1.1: Lucas recounts the traumatic event that he experienced, describing how it left
him feeling hopeless and alone.
Point2.1.2: Sarah tries to empathize with Lucas's pain and encourages him to seek therapy
or support from others instead of seeking revenge on the world.
Point2.1.2.1: Sarah suggests different coping mechanisms and resources for Lucas to seek
help and healing.
Point2.1.2.1.1: Sarah suggests meditation and mindfulness exercises that have helped her
deal with her own emotional pain in the past.
```

Point2.1.2.1.2: Sarah recommends a specific therapist or support group that she thinks may be helpful for Lucas.
Point2.1.2.1.3: Sarah encourages Lucas to engage in physical activities like exercise or sports to release his pent-up aggression and stress in a healthy way.
Point2.1.2.2: Lucas pushes back against the idea of seeking therapy, arguing that it won't change anything and that his plan for revenge is the only way to feel better.
Point2.1.2.2.1: Sarah emphasizes the potential benefits of therapy and the harm that revenge will cause.
Point2.1.2.2.2: Lucas argues that revenge is the only way to feel empowered and that it is too late for him to seek help or healing.
Point2.1.2.2.2.1: Sarah explains how seeking revenge will only perpetuate the cycle of pain and suffering.
Point2.1.2.2.2.2: Lucas argues that he has already made up his mind and that nothing can change it.
Point2.1.2.2.2.3: Sarah pleads with Lucas to consider the innocent people who will be hurt by his actions.
Point2.1.2.2.3: Sarah expresses her concern for Lucas and tries to persuade him to think about the consequences of his plan on innocent people.
Point2.1.2.2.3.1: Sarah pleads with Lucas to think about the innocent people who will be harmed by his plan for revenge.
Point2.1.2.2.3.2: Lucas tries to justify his plan by arguing that the innocent people who will be affected are collateral damage and that it is worth it to him to feel empowered.
Point2.1.2.2.3.3: Sarah expresses her disappointment in Lucas and tells him that she can no longer support his plan to destroy the world.
Point2.1.2.3: Sarah expresses her concern for Lucas and urges him to reconsider his plan, emphasizing the negative consequences it will have on innocent people.
Point2.1.3: Lucas dismisses Sarah's advice, insisting that revenge is the only way to feel empowered again and that it is too late for him to seek help or healing.
Point2.2: Sarah tries to reason with Lucas and convince him that there are other ways to cope with his pain and anger, but he is too consumed by his desire for revenge.
Point2.2.1: Sarah tries to reason with Lucas by reminding him of the innocent lives that would be lost if he carries out his plan.
Point2.2.2: Lucas argues that the world is corrupt and deserves to be destroyed, and that his actions will bring justice to all the victims of the world's injustices.
Point2.2.2.1: Lucas claims that the current system is so broken that only a drastic action like his plan can bring about change.
Point2.2.2.2: Lucas explains that his actions will bring justice to all the victims of the world's injustices.
Point2.2.2.3: Sarah counters Lucas's arguments by pointing out that his plan will only create more suffering.
Point2.2.2.3.1: Sarah provides examples of positive changes that Lucas could make, such as volunteering or starting a support group for others who have experienced similar trauma.
Point2.2.2.3.2: Sarah suggests that instead of destroying the world, Lucas could use his experiences to help make positive changes and fight for justice.
Point2.2.2.3.3: Sarah expresses her concern for Lucas's well-being and offers to help him seek therapy or other professional help to work through his trauma.
Point2.2.3: Sarah counters Lucas's arguments by pointing out that his plan will only create more suffering and that there are other ways to bring about change, such as activism or seeking therapy.
Point2.2.3.1: Sarah suggests that activism could be a better way to bring about meaningful change, and cites examples of historical figures who used non-violent methods to make a difference.
Point2.2.3.2: Lucas dismisses the idea of therapy, saying that it won't change anything and that revenge is the only way to make things right.
Point2.2.3.2.1: Lucas explains that he has already tried therapy and that it didn't help him at all.
Point2.2.3.2.2: Lucas argues that revenge is the only way to make things right because it will bring justice to all the victims of the world's injustices.
Point2.2.3.2.3: Sarah suggests that perhaps Lucas just didn't find the right therapist and encourages him to give it another try.
Point2.2.3.3: Sarah realizes that she needs to find a way to help Lucas heal from his past trauma before he can see the value in non-violent solutions.
Point2.3: Sarah struggles with conflicting emotions of wanting to stop Lucas from carrying out his plan, but also wanting to help him heal from his past trauma.
Point3: Sarah realizes that she may be the only one who can stop Lucas from carrying out his plan, but also struggles with the conflicting emotions of wanting to save her friend and wanting to save the world.
Point3.1: Sarah tries to reason with Lucas, explaining why his plan to destroy the world is wrong and offering alternative solutions.

```
Point3.2: Sarah seeks help from others, but nobody believes her until they realize Lucas'
plan is already in motion.
Point3.3: Sarah realizes that she has no other choice but to stop Lucas by force, and with
the help of authorities, she is able to prevent his plan from succeeding.
```

Table 37: The second short outline from CONCOCT. Compared with BASE, the outline dives more deeply into the motivation of "destroy" and the interaction between two characters. CONCOCT keeps expanding when a plot point includes general words like "event," "support," "consequence." However, understandably, readers may complain that there are too many rounds of plot twists, e.g., over-expanded Point 2.1.2.2.

```
Base Short Outline for Premise 3
```
```
Premise: "Hello, I'm your twin," said the stranger who stepped out of the bright white
light.
```
```
Outline:
Point1: A stranger emerges from a bright white light and claims to be the protagonist's
twin.
Point1.1: The protagonist is alone when a bright white light suddenly appears.
Point1.1.1: Alex is walking home from work late at night.
Point1.1.2: Suddenly, a bright white light appears in front of Alex.
Point1.1.3 (optional): Alex stops in their tracks and stares at the light in confusion.
Point1.2: A stranger steps out of the light and introduces themselves as Alex's twin.
Point1.2.1: The stranger introduces themselves as Avery, Alex's twin.
Point1.2.2: Avery shares specific memories and experiences that only a twin would know,
convincing Alex of their shared identity.
Point1.2.3: Alex begins to question how they were separated and why they were not aware of
each other's existence.
Point1.3 (optional): Alex is initially skeptical but Avery provides convincing evidence of
their shared past.
Point1.3 (optional).1: Avery shows Alex childhood photos of them together.
Point 1.3 (optional).2: Avery recounts a specific childhood memory that only the two of
them would know.
Point1.3 (optional).3: Alex finally accepts that Avery is their twin and they embrace each
other.
Point2: The protagonist struggles to come to terms with the idea of having a twin and
questions the stranger's identity.
Point2.1: The protagonist confronts the stranger with doubts about their twin claim.
Point2.1.1: Emily confronts Alex about her doubts regarding his twin claim.
Point2.1.2: Alex provides Emily with some personal information to prove his identity.
Point2.1.3: Emily starts to believe Alex's twin claim after seeing the evidence.
Point2.2: Alex provides Emily with evidence to support their twin claim.
Point2.2.1: Alex shows Emily old family photos of them as babies and children together.
Point2.2.2: Emily and Alex compare childhood memories and discover they have many shared
experiences.
Point2.2.3: Alex tells Emily about a birthmark they both have in the same spot, proving
they are truly twins.
Point2.3: Emily and Alex begin to bond as they explore their shared interests and
experiences.
Point2.3.1: Emily and Alex discover they have a shared love for hiking and decide to go on
a hike together.
Point2.3.2: While on the hike, Emily and Alex share stories about their childhood and begin
to piece together their past.
Point2.3.3: Emily and Alex find an old family photo album in a hidden compartment of a tree
and learn more about their family history.
Point3: The protagonist and the stranger work together to uncover the truth about their
past and how they were separated.
Point3.1: The protagonist and the stranger investigate their past by searching for clues in
their childhood home.
Point3.1.1: Emma and Adam arrive at their childhood home and begin searching for clues.
Point3.1.2: Emma finds an old photo album and discovers a picture of her and Adam together
as babies.
Point3.1.3: Adam finds a hidden letter from their birth mother explaining the family feud
and their separation.
Point3.2: Emma and Adam discover that they were separated at birth due to a family feud and
work to reunite their estranged parents.
```

```
Point3.2.1: Emma and Adam research their family's history to understand the root of the
feud that separated them.
Point3.2.2: Emma and Adam track down their estranged parents and work to bring them
together for a reconciliation.
Point3.2.3: Emma and Adam successfully reunite their parents, bringing closure to the
family and strengthening their bond as twins.
Point3.3: Emma and Adam confront their parents about the truth and reconcile their family.
Point3.3.1 : Emma and Adam confront their mother about the family feud and learn the truth
about their separation.
Point3.3.2 : Emma and Adam track down their estranged father and convince him to reconcile
with their mother.
Point3.3.3 : Emma and Adam bring their parents together and reconcile their family.
```

Table 38: The third short outline from BASE. Formatting errors appear multiple times in the subtrees under Point 1.1.3 and Point 1.3; for example, including "optional" in the index is improper. Moreover, the annotator thinks some sub-points are too detailed and might be meaningless (such as Point 3.1.1).

```
Concoct Short Outline for Premise 3

Premise: "Hello, I'm your twin," said the stranger who stepped out of the bright white
light.

Outline:
Point1: A stranger steps out of a bright white light and claims to be the protagonist's
twin.
Point1.1: Protagonist encounters a stranger stepping out of a bright white light.
Point1.2: The stranger claims to be the protagonist's twin.
Point2: The protagonist tries to verify the stranger's claim while dealing with their own
disbelief and confusion.
Point2.1: The protagonist questions the stranger about their past and how they could be
twins.
Point2.2: The protagonist tries to find evidence to support or disprove the stranger's
claim.
Point2.3: The protagonist confides in a trusted friend or family member about the situation
and seeks their advice.
Point2.3.1: The protagonist explains their encounter with the stranger in detail to their
trusted friend/family member.
Point2.3.1.1: The protagonist describes the stranger's appearance and behavior when they
claimed to be their twin.
Point2.3.1.2: The trusted friend/family member asks clarifying questions to understand the
full situation.
Point2.3.1.2.1: The trusted friend/family member asks about the circumstances surrounding
the bright white light.
Point2.3.1.2.2: The protagonist describes the location and time of day when they
encountered the stranger and the bright white light.
Point2.3.1.2.2.1: The trusted friend/family member asks if the protagonist saw anything
else unusual in the area at the time of the encounter.
Point2.3.1.2.2.1.1: The protagonist recalls hearing a strange humming noise coming from the
direction of the bright white light.
Point2.3.1.2.2.1.2: The trusted friend/family member asks if the humming noise was
continuous or if it had any noticeable pattern.
Point2.3.1.2.2.1.3: The protagonist describes how the humming noise suddenly stopped when
the stranger stepped out of the bright white light.
Point2.3.1.2.2.2: The trusted friend/family member asks if there were any other witnesses
to the bright white light and the stranger's appearance.
Point2.3.1.2.2.3: The protagonist describes any unusual sounds or physical sensations they
experienced during the encounter.
Point2.3.1.2.2.3.1: The protagonist describes a loud humming sound that accompanied the
bright white light.
Point2.3.1.2.2.3.1.1: The protagonist notices a ringing in their ears after the loud
humming sound.
Point2.3.1.2.2.3.1.2: The trusted friend/family member asks if the humming sound was
similar to any other sounds the protagonist has heard before.
Point2.3.1.2.2.3.1.3: The protagonist explains that the humming sound was so loud it
drowned out all other ambient noise.
Point2.3.1.2.2.3.2: The protagonist explains that they felt a strong gust of wind when the
stranger appeared.
```

```
Point2.3.1.2.2.3.3: The protagonist mentions a tingling sensation in their hands and feet
as the stranger claimed to be their twin.
Point2.3.1.2.3: The trusted friend/family member asks if the protagonist has experienced
any other unusual events lately that might be related to the stranger's appearance.
Point2.3.1.3: The protagonist recounts their initial thoughts and emotions when they
encountered the stranger.
Point2.3.1.3.1: The protagonist recalls their initial reaction when they first saw the
stranger.
Point2.3.1.3.2: The trusted friend/family member asks the protagonist how they felt about
the stranger's claim of being their twin.
Point2.3.1.3.2.1: The protagonist expresses their initial disbelief about the stranger's
claim.
Point2.3.1.3.2.2: The trusted friend/family member asks the protagonist if they noticed any
similarities or differences between themselves and the stranger.
Point2.3.1.3.2.2.1: The protagonist mentions a few differences in physical appearance, such
as hair or eye color, that they noticed between themselves and the stranger.
Point2.3.1.3.2.2.2: The trusted friend/family member asks the protagonist if they noticed
any similarities in the way they and the stranger spoke or acted, such as using similar
phrases or gestures.
Point2.3.1.3.2.2.3: The protagonist recalls some subtle differences in the way the stranger
spoke and carried themselves compared to their own behavior.
Point2.3.1.3.2.3: The protagonist recalls some similarities in appearance and personality
between themselves and the stranger.
Point2.3.1.3.3: The protagonist expresses their doubts and confusion about the situation to
their friend/family member.
Point2.3.2: The trusted friend/family member offers their perspective on the situation and
suggests possible explanations or solutions.
Point2.3.3: The protagonist considers their friend/family member's advice and decides on a
course of action.
Point3: The protagonist and their twin work together to uncover the truth about their past
and how they got separated.
Point3.1: The protagonist and their twin investigate their childhood home and find clues
about their past.
Point3.2: The protagonist and their twin uncover a shocking revelation that challenges
everything they thought they knew about their past.
Point3.3 : The protagonist and their twin confront the person responsible for their
separation and find closure.
```

Table 39: The third short outline from CONCOCT. The system functions well when faced with the general word "question" in Point 2.3.1.2. Also, for questions without answers or responses, the system will often expand one more step under them. One defect is that the end of the story is a bit rushed.

```
Concoct Short Outline for Premise 4

Premise: "Men, you are about to do the most dangerous thing any mind reader has done: you
will read a schizophrenic."

Outline:
Point1: A group of men are tasked with reading the mind of a schizophrenic, which is
considered a dangerous feat in the world of mind reading.
Point2: The men attempt to read the schizophrenic's mind, but struggle to differentiate
between reality and the delusions of the patient.
Point2.1: The men struggle to differentiate between Sarah's delusions and reality while
attempting to read her mind.
Point2.1.1: The men attempt to ask Sarah questions to determine what is real and what is
not in her mind.
Point2.1.1.1: John asks Sarah about her current surroundings to test if her delusions are
affecting her perception of reality.
Point2.1.1.2: Mark suggests asking Sarah to name some historical events to see if her grasp
on reality is consistent.
Point2.1.1.3: David suggests asking Sarah to describe a specific object in the room to see
if her perception matches reality.
Point2.1.2: Sarah becomes agitated and defensive, making it difficult for the men to gain
any useful information.
Point2.1.3: David suggests trying a different approach and asks Sarah to focus on a
specific memory.
```

Point2.2: Sarah's mind begins to spiral out of control, causing confusion and disorientation for the men attempting to read her thoughts.
Point2.3: The men start to feel paranoid and fearful as they realize the danger of delving too deep into Sarah's mind.
Point3: As the men continue to read Sarah's mind, they begin to experience hallucinations and paranoia, leading to a dangerous and unpredictable situation.
Point3.1: The men begin to experience intense hallucinations, causing them to lose touch with reality and question their own sanity.
Point3.1.1: John begins to see vivid and disturbing images in his mind that he can't shake off, causing him to question if he's losing his grip on reality.
Point3.1.2: Mark starts hearing voices and feeling like he's being watched, making him suspicious of his colleagues and causing him to lash out.
Point3.1.3: Alex experiences intense anxiety and panic attacks as he struggles to differentiate between Sarah's delusions and reality.
Point3.2: As the men become increasingly paranoid and agitated, they begin to turn on each other, unsure of who to trust.
Point3.2.1: John accuses Mark of withholding information and working against the group.
Point3.2.2: Alex becomes defensive and hostile towards the others, refusing to share his thoughts or insights.
Point3.2.3: Sarah's delusions start to influence John's perceptions, causing him to question his own judgement and sanity.
Point3.2.3.1: John begins to doubt his own ability to distinguish between reality and Sarah's delusions.
Point3.2.3.1.1: John begins to second-guess every thought he has while reading Sarah's mind, causing him to lose confidence in his abilities as a mind reader.
Point3.2.3.1.1.1: John's uncertainty and lack of confidence start to affect his behavior, making him hesitant and indecisive.
Point3.2.3.1.1.2: John becomes hesitant to share his thoughts with the group.
Point3.2.3.1.1.3: John's doubts and confusion cause him to make mistakes while reading Sarah's mind.
Point3.2.3.1.2: John becomes increasingly hesitant to share his thoughts with the group, fearing that his own doubts and confusion will be exposed.
Point3.2.3.1.3: Sarah's delusions start to seep into John's own thoughts and perception of reality, making it difficult for him to separate his own mind from hers.
Point3.2.3.2: Sarah's delusions cause John to question the motives and intentions of his fellow mind readers.
Point3.2.3.3: John's growing paranoia and distrust leads him to make a dangerous and impulsive decision.
Point3.3: Sarah's delusions become more intense and chaotic, causing the men to struggle even more with separating reality from her distorted perceptions.
Point3.3.1: Sarah's delusions cause her to become violent and unpredictable, putting the men in danger.
Point3.3.2: The men's attempts to calm Sarah down only serve to escalate the situation, as they inadvertently reinforce her delusions.
Point3.3.2.1: John realizes that their efforts to calm down Sarah are only making things worse, and they need to find a new approach to deescalate the situation.
Point3.3.2.1.1: John suggests a new approach to deescalate the situation by acknowledging Sarah's perceptions and guiding her towards reality.
Point3.3.2.1.2: Mark voices his concerns about John's new approach, citing potential dangers and risks.
Point3.3.2.1.3: Alex agrees with John's new approach and suggests ways to implement it effectively.
Point3.3.2.2: Mark, overwhelmed with fear and paranoia, makes a rash decision that puts himself and the others in danger.
Point3.3.2.2.1: Mark's fear and paranoia cause him to make a reckless decision, grabbing a sharp object and threatening Sarah with it.
Point3.3.2.2.2: John and Alex try to talk Mark down and convince him to put the sharp object down before someone gets hurt.
Point3.3.2.2.3: Sarah's behavior becomes even more erratic in response to Mark's threat, making it even harder for the group to deescalate the situation.
Point3.3.2.3: Alex suggests a new strategy that involves speaking to Sarah in a way that acknowledges her perceptions while gently guiding her towards reality.
Point3.3.3: John realizes that they have underestimated the severity of Sarah's schizophrenia, and they need to come up with a new plan to address the situation before it becomes even more dangerous.

Table 40: The fourth short outline from CONCOCT. Overall, the story goes smoothly and naturally. Due to the large number of general words in this somewhat hard topic and the limited number of expansion steps, CONCOCT still has a few vague words/points in the final outline. This could be remedied by LLM improving its external knowledge or by using a larger pre-set number of retries when failing an attempted expansion in CONCOCT.

| Concoct Short Outline for Premise 5 |
| --- |
| Premise: A story that begins with a lyric to your favorite song. |
| Outline:
Point1: A young woman named Emily is listening to her favorite song and reminiscing about her past relationship with her ex-boyfriend, who she still has feelings for.
Point2: Emily decides to reach out to her ex-boyfriend, Michael, and they agree to meet up and catch up on old times.
`Point2.1`: Emily sends Michael a message, asking if they can meet up.
Point2.2: Michael agrees to meet up with Emily and they plan a time and place.
Point2.2.1: Michael suggests a place for them to meet up.
Point2.2.2: Emily confirms the time and place that Michael suggested.
Point2.2.3: Emily and Michael exchange a few more messages to finalize their plans and express their excitement to see each other again.
Point2.2.3.1: Emily confirms the time and place for their meeting.
Point2.2.3.2: Michael expresses his excitement to see Emily again.
Point2.2.3.3: Emily expresses her excitement to see Michael again and mentions that she has missed him a lot.
Point2.3: Emily and Michael meet up and reminisce about old times, catching up on what they've been up to since they last spoke.
Point2.3.1: Emily and Michael share stories about their lives since their breakup.
Point2.3.2: They discuss the reasons for their breakup and apologize for their mistakes.
Point2.3.2.1: Emily admits that she was too controlling in their relationship and apologizes to Michael for not giving him enough space.
Point2.3.2.2: Michael apologizes for not being more communicative and for taking Emily for granted in their relationship.
Point2.3.2.3: They both agree that they want to work on their issues and make their relationship stronger this time around.
Point2.3.3: Emily and Michael confess their feelings for each other and decide to give their relationship another chance.
Point2.3.3.1: Emily and Michael express their feelings for each other.
Point2.3.3.2: Emily and Michael decide to give their relationship another chance.
Point2.3.3.3: Emily and Michael start dating again.
Point3: Emily and Michael meet up and realize that they still have strong feelings for each other. They decide to give their relationship another chance and start dating again.
Point3.1: Emily and Michael go on a date to a fancy restaurant and have a great time catching up and reminiscing about old memories.
Point3.1.1: Emily and Michael arrive at the restaurant and are seated at their reserved table.
Point3.1.2: Emily and Michael order their meals and share stories about their lives since they last saw each other.
Point3.1.2.1: Emily shares about her new job and how it's been keeping her busy.
Point3.1.2.2: Michael talks about his recent travels and the different cultures he experienced.
Point3.1.2.3: Emily and Michael reminisce about a funny memory they shared when they were together before.
Point3.1.3: Emily and Michael enjoy their delicious meals and talk about their future plans as a couple.
Point3.1.3.1: Emily shares her aspirations for the future, including her dream of starting her own business.
Point3.1.3.2: Michael talks about his own goals and how he believes they can support each other in achieving them.
Point3.1.3.3: Emily and Michael express their love for each other and make a commitment to prioritize their relationship and make it work.
Point3.2: Michael surprises Emily with tickets to her favorite band's concert, and they have a fun and memorable night together.
Point3.2.1: Michael surprises Emily with tickets to her favorite band's concert.
Point3.2.2: Emily and Michael dance and sing along to their favorite songs at the concert.
Point3.2.3: Emily and Michael take a stroll around the city after the concert, holding hands and enjoying each other's company. |

```
Point3.3: Emily and Michael have a heart-to-heart conversation about their past mistakes
and decide to work on their relationship together, promising to communicate better and be
more understanding of each other's needs.
Point3.3.1: Emily and Michael discuss their past mistakes and how they can improve their
communication in the future.
Point3.3.1.1: Emily expresses her feelings about how Michael's lack of communication hurt
her in the past.
Point3.3.1.2: Michael listens to Emily and apologizes for not communicating effectively in
their past relationship.
Point3.3.1.3: Emily and Michael brainstorm ways to improve their communication in the
future, such as setting aside time to talk and actively listening to each other.
Point3.3.2: They make a plan to have weekly check-ins to discuss any issues or concerns in
their relationship.
Point3.3.3: They agree to be more understanding of each other's needs and to prioritize
their relationship.
```

Table 41: The fifth short outline from CONCOCT. We think the outline is peaceful and smooth in pacing. Though the annotator mentions Point 2.1 is vague, perhaps because it does not show the content of the message, we think it's not that necessary.

```
Concoct Long Outline for Premise 6

Premise: A game of poker played by people who have no concept of the game, so they try to
incorporate rules from other games

Outline:
Point1: A group of friends decide to play poker despite having no knowledge of the game,
and they attempt to incorporate rules from other games.
Point2: The group begins playing poker, but confusion ensues as they struggle to understand
the basic rules and incorporate rules from other games.
Point2.1: The group decides to start playing poker, but they quickly realize that they have
no idea how to play the game.
Point2.1.1: The group sets up the poker table and prepares to start playing.
Point2.1.2: They realize that none of them actually know how to play poker.
Point2.1.3: They search online for the rules of poker but find them too technical and
confusing.
Point2.1.4: They attempt to read the rules out loud to each other, but still struggle to
understand the game.
Point2.1.4.1: The group attempts to read the rules of poker out loud to each other.
Point2.1.4.2: They struggle to understand the technical language used in the rules.
Point2.1.4.3: They ask each other questions about the rules, but no one seems to have a
clear answer.
Point2.1.4.4: They get frustrated and start making jokes about how confusing the rules are.
Point2.1.4.5: They decide to take a break and try to come back to the rules later.
Point2.1.5: They start making up their own rules and trying to incorporate rules from other
games.
Point2.2: They attempt to read the rules of poker, but the language is too technical and
confusing for them to understand.
Point2.3: They start making up their own rules and trying to incorporate rules from other
games, leading to even more confusion.
Point2.3.1: The group decides to make up their own rules for poker.
Point2.3.1.1: John suggests that they should be able to trade cards with each other, which
causes confusion among the group.
Point2.3.1.2: Sarah disagrees with John's suggestion and argues that trading cards is not a
part of poker.
Point2.3.1.3: Michael suggests that they should only be able to trade cards if they have a
certain combination, leading to further disagreement among the group.
Point2.3.1.4: Emily suggests that they should only be able to trade cards if they have a
certain number of chips, causing even more confusion and disagreement.
Point2.3.2: John suggests that they should be able to trade cards with each other, which
causes confusion among the group.
Point2.3.3: Sarah suggests that they should be able to play multiple hands at once, leading
to disagreement among the group.
Point2.3.4: Michael suggests that they should be able to bluff even if they don't have any
good cards, causing even more confusion and disagreement.
Point2.3.5: Emily suggests that they should be able to change the rules of the game at any
time, leading to frustration and tension among the group.
```

Point2.4: They argue about the rules and what they mean, leading to frustration and tension among the group.
Point2.4.1: The group argues about the meaning of a specific rule in poker, leading to frustration among the group.
Point2.4.2: David becomes increasingly agitated and starts yelling at the group, causing tension and discomfort.
Point2.4.2.1: David becomes increasingly agitated and starts yelling at the group.
Point2.4.2.2: The group tries to calm David down and figure out why he is so upset.
Point2.4.3: Emily tries to calm David down and suggests taking a break, but he refuses to listen and continues to argue.
Point2.4.3.1: Emily tries to calm David down.
Point2.4.3.2: David refuses to listen and continues to argue.
Point2.4.4: Sarah suggests calling a friend who knows how to play poker to settle the argument, but John disagrees and says they should figure it out on their own.
Point2.4.5: John suggests taking a break and coming back to the game later when everyone is less frustrated and confused.
Point2.5: They take a break and try to watch a tutorial video on how to play poker, but they still struggle to understand the game.
Point2.5.1: The group takes a break from playing poker and decides to watch a tutorial video on how to play.
Point2.5.1.1: The group agrees to take a break from playing poker and watch a tutorial video on how to play.
Point2.5.1.2: They search for a tutorial video on how to play poker on their phones and find one that looks promising.
Point2.5.1.3: They gather around a laptop and start watching the tutorial video.
Point2.5.1.3.1: The group gathers around a laptop to watch the tutorial video on how to play poker.
Point2.5.1.3.2: They take notes while watching the video to help them remember the rules better.
Point2.5.1.3.3: The video shows examples of different hands and how to bet in poker.
Point2.5.1.3.4: The group pauses the video to ask questions and clarify any confusion they have about the rules.
Point2.5.1.3.5: The group continues watching the tutorial video, pausing and rewinding whenever necessary to fully understand the rules.
Point2.5.1.4: The video introduces the basic rules of poker, such as the different hands and the importance of betting.
Point2.5.1.4.1: The group comments on how difficult it is to understand the rules, but they continue watching the video in hopes of gaining clarity.
Point2.5.1.4.2: The group pauses the video to discuss any confusion they have about the rules.
Point2.5.1.4.3: They ask each other questions about the rules and try to clarify any confusion.
Point2.5.1.4.4: They realize that they need to practice playing poker with the basic rules before trying to incorporate any additional rules.
Point2.5.1.4.5: The video explains the importance of reading other players' body language and facial expressions to determine if they are bluffing.
Point2.5.1.5: The group pauses the video to discuss any confusion they have about the rules.
Point2.5.2: The tutorial video explains the basic rules of poker, but the group still struggles to understand the game.
Point2.5.2.1: The tutorial video explains the basic rules of poker, including the ranking of hands and the different betting rounds.
Point2.5.2.1.1: The tutorial video explains the basic rules of poker, including the ranking of hands.
Point2.5.2.1.2: The tutorial video explains the different betting rounds in poker.
Point2.5.2.2: The group struggles to understand the terminology used in the tutorial video, such as "flop" and "river".
Point2.5.2.3: They pause the video and discuss the meaning of the terms, but still feel confused.
Point2.5.2.3.1: The group pauses the tutorial video to discuss the meaning of the terms used in poker.
Point2.5.2.3.2: They try to clarify the meaning of terms like "flop" and "river", but still feel confused.
Point2.5.2.4: The tutorial video explains the difference between a "raise" and a "call", but the group still struggles to understand the concept.
Point2.5.2.4.1: The tutorial video explains the difference between a "raise" and a "call".
Point2.5.2.4.2: The group discusses the difference between a "raise" and a "call", but still struggles to understand the concept.
Point2.5.2.4.2.1: Michael asks for clarification on the difference between a "raise" and a "call".

Point2.5.2.4.2.2: David attempts to explain the difference between a "raise" and a "call", but only confuses the group further.
Point2.5.2.5: They pause and rewind the video multiple times to try to understand the difference between a "raise" and a "call".
Point2.5.3: They pause and rewind the video multiple times to try to understand the rules.
Point2.5.4: They discuss the video and try to clarify any confusion, but still struggle to understand certain aspects of the game.
Point2.5.4.1: The group discusses a specific rule they are confused about in the tutorial video.
Point2.5.4.2: They debate the interpretation of the rule and how it should be applied in the game.
Point2.5.4.2.1: The group debates the interpretation of a specific rule from the tutorial video.
Point2.5.4.2.2: They discuss how the rule should be applied in a hypothetical situation.
Point2.5.4.2.3: They analyze the potential outcomes of applying the rule in different ways.
Point2.5.4.2.4: They try to come to a consensus on the interpretation of the rule.
Point2.5.4.2.5: They still cannot agree on the interpretation of the rule and decide to move on to the next rule.
Point2.5.4.3: They try to come to a consensus on the rule, but still have differing opinions.
Point2.5.4.4: They decide to pause the video and discuss the rule in more detail.
Point2.5.4.4.1: The group decides to pause the video and discuss the rule in more detail.
Point2.5.4.4.2 : They analyze the rule and try to apply it to a hypothetical situation to better understand it.
Point2.5.4.5: They analyze the rule and try to apply it to a hypothetical situation to better understand it.
Point2.5.5: Despite their efforts to understand the game through the tutorial video, the group still feels unsure about the rules and returns to playing poker with their made-up rules.
Point2.5.5.1: The group returns to playing poker with their made-up rules.
Point2.5.5.2: They continue to struggle to understand the game despite their attempts to clarify the rules.
Point3: John suggests incorporating the rules of Uno into the poker game, causing even more confusion and chaos.
Point4: Sarah suggests incorporating the rules of Monopoly into the poker game, leading to even more confusion and disagreement among the group.
Point4.1: Sarah suggests incorporating the rules of Monopoly into the poker game, but the group is unsure how to do it.
Point4.1.1: The group tries to brainstorm ways to incorporate Monopoly rules into the poker game.
Point4.1.2: The group discusses the different aspects of Monopoly that could be incorporated into the poker game.
Point4.1.3: John suggests incorporating the "Go to Jail" rule from Monopoly, but the group cannot agree on how it would work in the context of poker.
Point4.1.3.1: John suggests incorporating the "Go to Jail" rule from Monopoly into the poker game.
Point4.1.3.2 : The group discusses how the "Go to Jail" rule could work in the context of poker.
Point4.1.3.3: Sarah suggests that players who receive the "Go to Jail" card should have to sit out a round of poker.
Point4.1.3.4: Michael argues that sitting out a round of poker is too harsh of a punishment for receiving the "Go to Jail" card.
Point4.1.3.5: Emily suggests that players who receive the "Go to Jail" card should have to pay a fine to the pot.
Point4.1.3.6: David proposes that players who receive the "Go to Jail" card should have to swap one of their cards with another player's card.
Point4.1.4: Sarah proposes that players should be able to collect "Community Chest" cards, but the group is unsure what the cards would do in the game.
Point4.2: Sarah proposes that players can earn money by landing on certain cards, which leads to disagreement among the group.
Point4.3: Michael suggests that players should be able to buy and sell cards, but the group cannot agree on how to implement this rule.
Point4.4: Emily suggests that players should be able to trade cards with each other, but the group cannot agree on the value of each card.
Point4.5: David suggests that players should be able to build hotels on their cards, which confuses everyone and leads to further disagreement.
Point5: The group eventually gives up on incorporating rules from other games and attempts to play poker with the basic rules, but still struggles to understand the game.
Point5.1: The group decides to give up on incorporating rules from other games.

```
Point5.2: They attempt to play poker with the basic rules.
Point5.3: The group still struggles to understand the game.
```

Table 42: The first long outline from CONCOCT. Overall, CONCOCT does fairly well to expand the vaguest parts. For example, CONCOCT realizes Point 2 contains the largest chunk of plot and tries to expand it many times. It tries to discuss poker rules in detail. Of course, some parts can still be further improved: 1) For Point 3 and Point 4, the major difference between them is about "Uno" and "Monopoly"; which is more concrete is hard to say. 2) Point 4.1.3.2, after expansion, still does not introduce anything important and perhaps could be removed. 3) A rare situation occurs wherein one point overlaps with its parent's sibling, which is a failure of our base outline generation system.

---

**Concoct Long Outline for Premise 7**

Premise: At the coming of age everyone is given a single bullet with a name on. This is your target.

---

Outline:
Point1: At the coming of age, everyone is given a single bullet with a name on it. This is your target.
Point2: The young adults are tasked with finding and killing their designated target.
Point2.1: The young adults receive their designated target and begin their search for them.
Point2.1.1: John receives his designated target and learns their name.
Point2.1.1.1: John receives his designated target.
Point2.1.1.2: John learns the name of his designated target.
Point2.1.1.3: John researches his target's background and location.
Point2.1.1.4: John reaches out to Target 1's family and friends to gather more information.
Point2.1.1.4.1: John asks Target 1's family and friends about their relationship with Target 1 and any possible conflicts or issues they may have had in the past.
Point2.1.1.4.2: John asks Target 1's family and friends about their whereabouts and habits.
Point2.1.1.4.3: John asks Target 1's family and friends if they have any knowledge of Target 1's current whereabouts or recent activities.
Point2.1.1.4.4: John thanks Target 1's family and friends for their help and promises to keep them safe.
Point2.1.1.5: John creates a plan of action to track down Target 1.
Point2.1.1.5.1: John scouts the areas where Target 1 was last seen or known to frequent.
Point2.1.1.5.2: John considers the potential consequences of his actions and the morality of killing his designated target.
Point2.1.1.5.2.1: John considers the potential consequences of his actions.
Point2.1.1.5.2.2: John questions the morality of killing his designated target.
Point2.1.2: John creates a plan of action to track down Target 1.
Point2.1.2.1: John researches Target 1's daily routine to determine the best time to approach them.
Point2.1.2.2: John scouts Target 1's usual hangout spots to gather more information about their whereabouts.
Point2.1.3: John reaches out to Target 1's family and friends to gather more information about their whereabouts.
Point2.1.3.1: John reaches out to Target 1's family members to gather more information about their whereabouts.
Point2.1.3.2: John reaches out to Target 1's friends to gather more information about their whereabouts.
Point2.1.4: John searches social media and online platforms to find any clues about Target 1's location.
Point2.1.5: Target 1 realizes they are being hunted and goes into hiding.
Point2.2: John uses various methods to track down Target 1, including speaking with their family and friends and scouring social media.
Point2.3: Target 1 attempts to defend themselves and evade John's pursuit.
`Point2.4`: John confronts Target 1 and must make the decision to either carry out the task or spare their life.
Point2.4.1: John confronts Target 1 and holds them at gunpoint.
Point2.4.2: Target 1 pleads for their life and explains why they were assigned as John's target.
Point2.4.3: John hesitates and considers sparing Target 1.
Point2.4.4: Target 1 reveals information about the society's true motives and the consequences for not completing the task.
Point2.4.4.1: Target 1 reveals that the society assigns targets based on arbitrary criteria, not actual wrongdoing.

Point2.4.4.1.1: Target 1 explains how they were assigned as John's target despite not committing any actual wrongdoing.
Point2.4.4.1.2: Target 1 describes how other targets were chosen based on arbitrary criteria such as their ethnicity, religion, or even physical appearance.
Point2.4.4.1.3: Target 1 questions the morality of the society's traditions and urges John to do the same.
Point2.4.4.2: Target 1 explains that failure to complete the task results in severe punishment, potentially affecting the young adult's family and friends.
Point2.4.4.3: Target 1 urges John to reconsider his actions and to question the morality of the society's traditions.
Point2.5: Other young adults successfully locate and kill their targets, while some struggle with the task and fail to complete it.
Point2.5.1: Young adult named Sarah successfully locates and kills her target.
Point2.5.2: Young adult named Michael struggles with the task and ultimately fails to complete it.
Point2.5.3: Young adult named Emily becomes ruthless in her pursuit and goes beyond the task's requirements.
Point2.5.4: Young adult named David has a change of heart and decides not to kill his target.
Point2.5.5: Young adult named Alex accidentally kills the wrong person and must face the consequences.
Point3: The society watches and judges the young adults' actions as they hunt down their targets.
Point3.1: The society sets up observation posts to monitor the actions of the young adults as they hunt down their targets.
Point3.1.1: The society assigns a team of observers to each young adult to monitor their actions.
Point3.1.2: The observers document the young adult's movements and actions in a logbook.
Point3.1.3: The observers use binoculars and other equipment to track the young adults' movements from a distance.
Point3.1.4: The observers report any significant actions or events to the society for further discussion.
Point3.2: The society discusses and debates the actions of the young adults based on what they observe.
Point3.3: The society uses the young adults' actions as a means to reinforce their beliefs and values to the wider community.
Point3.3.1: The society uses the actions of successful young adults as a means to promote their beliefs and values to the wider community.
Point3.3.2: The society condemns the actions of unsuccessful young adults as a means to reinforce their beliefs and values to the wider community.
Point3.3.3: The society uses the actions of the young adults to justify the tradition of assigning targets and to maintain the status quo.
Point4: Some young adults struggle with the morality of killing their target, while others become ruthless in their pursuit.
Point4.1: Some young adults struggle with the morality of killing their target.
Point4.2: Others become obsessed with completing the task at any cost and become a danger to themselves and those around them.
Point4.3: Some young adults begin to form alliances with their targets in order to avoid completing the task.
Point4.4: Ava becomes ruthless in her pursuit of her target and disregards the consequences of her actions.
Point5: The society reveals the reason for the tradition of assigning targets and the consequences for not completing the task.
Point5.1: The society reveals the reason for assigning targets as a way to control population growth and maintain order in society.
Point5.1.1: The society explains that they have been facing overpopulation and resource depletion for many generations.
Point5.1.2: The tradition of assigning targets is seen as the most effective way to control population growth and allocate resources.
Point5.1.2.1: The society has determined that assigning targets is the most effective way to allocate resources and prevent overpopulation.
Point5.1.2.2: The society believes that by eliminating those who are deemed a threat to social order, they can maintain harmony and stability.
Point5.1.3: The society justifies the tradition by stating that it has been successful in preventing overpopulation and maintaining social harmony.
Point5.2: The consequences for not completing the task are severe, including banishment from society and being labeled as a threat to the community.
Point5.2.1: The society warns the young adults that failure to complete their assigned task will result in banishment from society.

```
Point5.2.2: The young adults are informed that not completing their task will also result
in being labeled as a threat to the community.
Point5.2.2.1: The society explains that not completing the task will result in being
labeled as a threat to the community.
Point5.2.2.2: The young adults express their fear and concern about being labeled as a
threat if they fail to complete their task.
Point5.2.2.3: The society reassures the young adults that the label is only applied in
extreme cases and that they have the power to prevent it by completing their task.
Point5.2.3: The society stresses the importance of maintaining order and eliminating
potential disruptors, and notes that failure to carry out the task could lead to disastrous
consequences for the community.
Point5.2.3.1: The society emphasizes the potential consequences of not carrying out the
assigned task, including the disruption of social order and the safety of the community.
Point5.2.3.2: The society reminds the young adults that the tradition has been in place for
generations and has been successful in maintaining order and stability in society.
Point5.2.3.3: The society acknowledges the difficulty and moral implications of the task,
but stresses the importance of sacrificing individual morality for the greater good of
society.
Point5.3: The society explains that the targets are chosen based on their potential to
disrupt the social order and that the tradition has been in place for generations.
Point5.3.1: The society explains that the targets are chosen based on their potential to
disrupt the social order.
Point5.3.2: The tradition of assigning targets has been in place for generations.
Point5.3.3: The society justifies the tradition as a necessary measure to maintain order in
society.
Point5.4: Some young adults begin to question the morality and motive behind the tradition
and demand change from the society.
Point6: The young adults must face the aftermath of their actions and the impact it has on
their relationships and society as a whole.
Point6.1: The young adults must confront their families and friends about their actions and
the impact it has had on their relationships.
Point6.2: The society must deal with the aftermath of the tradition and the potential
consequences of allowing young adults to kill each other.
Point6.3: Some young adults may face legal consequences for not completing their task or
for their actions during the hunt.
Point6.4: The young adults must come to terms with the moral implications of their actions
and the toll it has taken on their mental health.
Point6.5: The tradition of assigning targets is questioned and potentially changed as a
result of the events that have transpired.
Point6.6: The young adults must find a way to move forward and rebuild their lives and
relationships in the aftermath of the hunt.
Point6.6.1: Some young adults struggle to rebuild their relationships with their families
and friends after the hunt.
Point6.6.2: Some young adults may seek therapy or counseling to help them process their
emotions and trauma from the hunt.
Point6.6.3: Some young adults may choose to leave the society and start a new life
elsewhere, away from the memories of the hunt.
Point6.6.4: The society may offer resources and support for young adults struggling to cope
with the aftermath of the hunt.
Point6.6.5: The young adults may come together to advocate for change and the end of the
tradition of assigning targets.
```

Table 43: The second long outline from CONCOCT. Overall, the outline is well-paced according to the annotator.

| Concoct Long Outline for Premise 8 |
| --- |
| Premise: The Door by Eric Basiletti |

```
Outline:
Point1: A man, John, discovers a mysterious door in his basement that leads to an alternate
dimension.
Point2: John begins to explore the alternate dimension and discovers its strange and
dangerous inhabitants.
Point2.1: John explores the alternate dimension and encounters strange creatures.
Point2.2: John discovers that the alternate dimension is dangerous and he must be careful
to avoid being captured or harmed.
Point2.3: John befriends a group of friendly creatures who help him navigate the dangers of
the alternate dimension.
```

Point2.3.1: John and his friendly alternate dimension creatures encounter a group of dangerous creatures.
Point2.3.2: John and his friends are outnumbered and outmatched by the dangerous creatures.
Point2.3.3: One of John's friendly creatures sacrifices itself to save the group from the dangerous creatures.
Point2.3.4: John and his friends are grateful for Sarah's help and invite her to join them on their journey.
Point2.4: John and his new friends are attacked by a group of hostile creatures and must fight for their survival.
Point2.4.1: John and his friends are ambushed by a group of hostile creatures.
Point2.4.2: The group engages in a fierce battle, using their unique abilities to defend themselves.
Point2.4.3: John and his friends are outnumbered and on the brink of defeat.
Point2.4.4: One of John's friends sacrifices themselves to save the group, giving them a chance to escape.
Point2.4.5: John and his friends are emotionally shaken by the loss of their comrade and must continue their journey with heavy hearts.
Point2.5: John and his friends narrowly escape the hostile creatures and continue their journey through the alternate dimension.
Point2.5.1: John and his friends come across a large, open field that they must cross to continue their journey.
Point2.5.2: As they cross the field, John notices a small group of hostile creatures in the distance.
Point2.5.3: John and his friends quickly change their path to avoid the hostile creatures, but they spot the group heading towards them.
Point3: John meets a woman, Sarah, in the alternate dimension who helps him navigate the dangers and reveals that she too is from John's world.
Point3.1: John and Sarah team up to explore the alternate dimension together.
Point3.2: Sarah reveals that she is also from John's world and was transported to the alternate dimension through a similar door in her own basement.
Point3.3: Sarah shares her knowledge of the alternate dimension and helps John navigate its dangers.
Point3.4: John and Sarah encounter a group of hostile inhabitants in the alternate dimension and must fight to survive.
Point3.4.1: John and Sarah are ambushed by a group of hostile inhabitants who want to capture them.
Point3.4.1.1: John and Sarah hear strange noises and realize they are being followed.
Point3.4.1.2: The hostile inhabitants emerge from the shadows and surround John and Sarah.
Point3.4.1.3: The hostile inhabitants demand that John and Sarah come with them, threatening them with violence if they refuse.
Point3.4.1.4: John and Sarah refuse to go with the hostile inhabitants, knowing that it will mean certain death.
Point3.4.1.5: John and Sarah realize they must fight for their survival and prepare to defend themselves.
Point3.4.1.6: The hostile inhabitants attack John and Sarah with crude weapons.
Point3.4.1.7: John and Sarah use their wits and makeshift weapons to fight back against the hostile inhabitants.
Point3.4.2: John and Sarah fight back against the hostile inhabitants using makeshift weapons.
Point3.4.3: Sarah uses her knowledge of the alternate dimension to outsmart the hostile inhabitants and lead John to safety.
Point3.4.3.1: Sarah uses her knowledge of the alternate dimension to lead John through a hidden passage that the hostile inhabitants don't know about.
Point3.4.3.2: John and Sarah make their way through the hidden passage, avoiding detection by the hostile inhabitants.
Point3.4.3.3: Sarah leads John to a secret exit that takes them out of the hostile territory and into a safer area of the alternate dimension.
Point3.4.3.4: John and Sarah take a moment to catch their breath and plan their next move.
Point3.5: Sarah shows John a hidden safe haven in the alternate dimension where they can rest and plan their next move.
Point4: John and Sarah work together to find a way back to their own world, but they are pursued by a malevolent entity that wants to keep them trapped in the alternate dimension.
Point4.1: John and Sarah search for a way back to their own world.
Point4.2: The malevolent entity begins to pursue John and Sarah.
Point4.3: John and Sarah encounter obstacles and dangers as they try to evade the entity.
Point4.3.1: John and Sarah must navigate through a maze-like structure filled with traps and obstacles to avoid the pursuing entity.
Point4.3.2: The entity sends its minions to attack John and Sarah as they try to make their way through the maze.

Point4.3.3: John and Sarah discover a hidden passageway that leads them to a lower level of the maze.
Point4.3.4: The entity corners John and Sarah in a dead-end room, forcing them to fight for their lives.
Point4.4: John and Sarah discover a portal back to their own world, but the entity is close behind.
Point4.4.1: John and Sarah discover the portal back to their own world.
Point4.4.2: The malevolent entity catches up to John and Sarah as they prepare to go through the portal.
Point4.4.3: John and Sarah fight to keep the entity at bay while trying to activate the portal.
Point4.4.4: John and Sarah manage to push the entity back and run through the portal.
Point4.5: John and Sarah make a desperate escape through the portal, but the entity manages to follow them.
Point4.5.1: John and Sarah make a desperate escape through the portal.
Point4.5.2: The malevolent entity manages to follow them through the portal.
Point4.5.3: John and Sarah realize that the entity has followed them.
Point4.5.4: The entity begins to wreak havoc in their world.
Point4.5.5: John and Sarah must find a way to stop the entity and protect their world.
Point5: John and Sarah finally find a way back to their own world, but they realize that the malevolent entity has followed them and is now wreaking havoc in their world.
Point5.1: John and Sarah return to their world and realize that the malevolent entity has followed them.
Point5.1.1: John and Sarah assess the damage caused by the malevolent entity in their world.
Point5.1.2: John and Sarah realize that the entity is becoming stronger and must be stopped before it's too late.
Point5.2: John and Sarah try to find a way to stop the malevolent entity from causing further damage.
Point5.3: John and Sarah discover that the only way to stop the entity is to return to the alternate dimension and find its source.
Point5.4: John and Sarah return to the alternate dimension and encounter even greater dangers than before.
Point5.4.1: John and Sarah are ambushed by a group of hostile creatures.
Point5.4.2: John and Sarah must navigate through a dense forest while being pursued by a dangerous predator.
Point5.4.3: John and Sarah come across a treacherous landscape that tests their physical and mental endurance.
Point5.4.4: John and Sarah find a hidden cave that leads to the source of the malevolent entity.
Point5.4.5: John and Sarah must navigate a series of traps and obstacles in the cave to reach their goal.
Point5.4.6: John and Sarah face off against a powerful guardian that is protecting the source of the malevolent entity.
Point5.4.7: John and Sarah use their combined strength and intelligence to defeat the guardian and destroy the source of the malevolent entity.
Point5.4.7.1: John and Sarah engage in a physical battle with the guardian.
Point5.4.7.2: The guardian uses its powers to create illusions to confuse John and Sarah.
Point5.4.7.3: Sarah uses her knowledge of the alternate dimension to see through the illusions and guide John in the battle.
Point5.4.7.4: John and Sarah work together to weaken the guardian's defenses and find its weak spot.
Point5.4.7.5: John delivers the final blow to the guardian, destroying it and opening the way to the source of the malevolent entity.
Point5.5: John and Sarah find the source of the malevolent entity and engage in a final battle to stop it.
Point5.5.1: John and Sarah navigate through the dangerous alternate dimension to reach the source of the malevolent entity.
Point5.5.2: John and Sarah encounter various obstacles and fight off hostile creatures along the way.
Point5.5.2.1: John and Sarah encounter a group of hostile creatures blocking their path.
Point5.5.2.2: John and Sarah use their weapons and skills to fight off the creatures.
Point5.5.2.3: John and Sarah sustain injuries during the fight and must tend to their wounds.
Point5.5.2.4: John and Sarah continue on their journey, but they are now more cautious and alert to potential dangers.
Point5.5.2.5: John and Sarah come across a treacherous terrain that requires them to navigate carefully.
Point5.5.3: John and Sarah discover the malevolent entity's weakness and plan their strategy accordingly.

Point5.5.4: John and Sarah engage in a fierce battle with the malevolent entity, using all the skills and knowledge they gained from their experiences in the alternate dimension.
Point5.5.4.1: John and Sarah engage in a fierce physical battle with the malevolent entity.
Point5.5.4.2: John and Sarah use their knowledge of the entity's weakness to gain an advantage in the battle.
Point5.5.4.3: The malevolent entity unleashes its full power, causing destruction and chaos in the alternate dimension.
Point5.5.4.4: John and Sarah struggle to hold their ground against the entity's overwhelming power.
Point5.5.4.5: Sarah sacrifices herself to weaken the entity, giving John the opportunity to strike the final blow.
Point5.5.4.6: John delivers the final blow, destroying the entity and saving both the alternate dimension and his own world.
Point5.5.4.6.1: John delivers the final blow to the malevolent entity.
Point5.5.4.6.2: The entity retaliates, unleashing a final burst of energy to try to stop John.
Point5.5.5: John and Sarah emerge victorious, but they are physically and emotionally drained from the ordeal.
Point5.5.5.1: John and Sarah are physically and emotionally drained from the battle.
Point5.5.5.2: John and Sarah take a moment to catch their breath and reflect on their experiences.
Point5.6: John and Sarah emerge victorious and return to their own world, but they are forever changed by their experiences.
Point5.6.1: John and Sarah emerge victorious in their final battle against the malevolent entity.
Point5.6.1.1: John and Sarah engage in a final battle with the malevolent entity.
Point5.6.1.2: John and Sarah use their knowledge and experience from the alternate dimension to gain the upper hand in the battle.
Point5.6.1.3: The malevolent entity is defeated and dissipates into the air, leaving behind a feeling of peace and tranquility.
Point5.6.1.4: John and Sarah embrace, relieved that the ordeal is over and grateful for each other's company and support throughout it all.
Point5.6.1.4.1: John and Sarah embrace each other in relief and gratitude for their victory.
Point5.6.1.4.2: They take a moment to catch their breath and look around at their surroundings.
Point5.6.1.4.3: John and Sarah exchange a few words, expressing their amazement at what they've just experienced.
Point5.6.1.4.4: They both acknowledge how much they've grown and changed throughout their journey.
Point5.6.1.4.5: John and Sarah share a kiss, sealing their bond and connection after everything they've been through.
Point5.6.2: John and Sarah struggle to readjust to their normal lives after their time in the alternate dimension.
Point5.6.2.1: John struggles with PTSD and nightmares from his time in the alternate dimension.
Point5.6.2.2: Sarah has difficulty readjusting to her old life and feels disconnected from her friends and family.
Point5.6.2.3: John and Sarah seek therapy to help them cope with their experiences.
Point5.6.2.4: John and Sarah find solace in each other and their shared experiences.
Point5.6.2.5: John and Sarah decide to keep the door in their basement locked and never speak of their experiences to anyone else.

Table 44: The third long outline from CONCOCT. Overall, the outline is well-paced according to the annotator.

| Downstream Stories of Concoct in Short Outline Regime for Premise 9 |
| --- |
| Premise: Planetary storms are a rich source of ion energy. Ion mining is the most lucrative and the most dangerous job in the galaxy. Tell me the story of an ion miner. |
| Point 1:
Outline: Jack struggles to learn the ropes of ion mining and feels like an outsider among the experienced crew members.
Chapter Content: As Jack stepped onto the ion mining ship for the first time, he couldn't help but feel overwhelmed by the experienced crew members bustling around him. They had a camaraderie he couldn't penetrate, and he struggled to find his place among them. The intricacies of ion mining seemed like a foreign language to Jack, and he constantly found himself stumbling over the terminology and procedures. It was clear that he had a long way to go before he could call himself a true ion miner. |

Point 2:
Outline: Jack makes a mistake during a mining operation, causing tension among the crew. He learns the importance of following protocol and gaining the trust of his crewmates.
Chapter Content: During a routine mining operation, Jack's lack of experience led him to make a critical mistake. He misjudged the stability of a mineral deposit, causing a small explosion that sent shockwaves through the ship. The crew was furious, their trust in Jack shattered. It was a harsh lesson for him to learn the importance of following protocol and gaining the trust of his crewmates. From that moment on, Jack vowed to do whatever it took to make amends and prove himself worthy.

Point 3:
Outline: Jack starts to earn small victories by showing his work ethic and dedication to the crew, which slowly gains their acceptance.
Chapter Content: Determined to make a fresh start, Jack dove into his work with unrelenting dedication. He volunteered for every task, no matter how mundane or physically demanding. Slowly but surely, the crew began to take notice of his work ethic and commitment. They saw that he was willing to put in the extra effort to earn his place among them. Jack started to receive praise for his small victories, whether it was successfully repairing a malfunctioning tool or showing exceptional knowledge during a crew meeting. With each accomplishment, his acceptance grew.

Point 4:
Outline: Jack faces a near-fatal accident while working with the ion energy. He realizes the gravity of the profession and the risks involved.
Chapter Content: The gravity of the profession hit Jack with full force when he experienced a near-fatal accident. While handling the volatile ion energy, a surge unexpectedly occurred, causing a powerful electric shock to course through his body. In that terrifying moment, as he fought for survival, he realized the immense risks involved in being an ion miner. It was a wake-up call that made him appreciate the importance of safety protocols and the need for constant vigilance in such a dangerous profession.

Point 5:
Outline: Jack spends long hours studying the manuals and asking questions to his crewmates to understand the basics of ion mining.
Chapter Content: In his quest to gain a deeper understanding of ion mining, Jack spent countless hours poring over the manuals provided by the mining corporation. He made notes, highlighting key points, and even created his own study guides to ensure he absorbed every crucial detail. Additionally, he sought guidance from experienced crewmates, asking questions about their own experiences and diving into technical discussions. Jack's determination to master the basics of ion mining was evident to everyone on board.

Point 6:
Outline: The crew takes Jack under their wing and shows him practical examples of how to mine ions, helping him to understand the process better.
Chapter Content: Recognizing Jack's growing commitment and eagerness to learn, the crew began to take him under their wing. They shared practical examples, demonstrating the intricacies of ion mining on the ship's simulation systems. They allowed Jack to shadow them during mining operations, explaining their thought processes and answering his questions along the way. Through hands-on experience and the guidance of his crewmates, Jack's understanding of the mining process deepened, giving him more confidence in his abilities.

Point 7:
Outline: Jack faces a difficult task during a storm and manages to complete it successfully, earning the respect and admiration of his crewmates.
Chapter Content: Jack faced his biggest challenge yet during a ferocious planetary storm. As the ship trembled under the force of the winds and lightning, the crew scrambled to secure their mining equipment. It was a daunting task, but Jack's hard work finally paid off. Applying the knowledge he had acquired, he successfully completed his assignment despite the treacherous conditions. The crew watched in awe as Jack skillfully maneuvered the ship, earning their respect and admiration.

Point 8:
Outline: Jack becomes an integral part of the crew, working alongside them seamlessly and earning their trust and friendship.

Chapter Content: Jack had finally become an integral part of the crew. He seamlessly worked alongside them, contributing to the success of their mining operations. The initial skepticism had transformed into deep trust and friendship. The crew no longer saw Jack as an outsider, but as a valuable team member. Whether it was assisting with repairs, analyzing mineral samples, or providing crucial insights during decision-making, Jack's skills and dedication had made him indispensable.

Point 9:
Outline: The crew's ship equipment malfunctions during a storm, putting them in danger.
Chapter Content: Disaster struck when the crew's ship equipment malfunctioned during a violent storm. The sudden loss of power and control left them vulnerable and adrift in space. Panic and fear spread among the crew as they realized the imminent danger they faced. It was a race against time to salvage the situation and prevent any further catastrophe. The crew desperately brainstormed for a solution, hoping that they could overcome this dire predicament and survive against all odds.

Point 10:
Outline: Jack suggests going outside to fix the equipment, but the crew is hesitant due to the danger.
Chapter Content: Jack's quick thinking led him to suggest a risky plan to fix the damaged equipment. Despite the crew's hesitation due to the inherent danger, Jack believed it was their only chance at survival. With unwavering determination, he explained his strategy, carefully highlighting the potential risks and benefits. It was a tense moment as the crew weighed their options, but eventually, they put their faith in Jack's expertise and agreed to embark on this perilous mission.

Point 11:
Outline: Jack takes a risk and successfully fixes the equipment, earning the respect of his crewmates.
Chapter Content: With adrenaline pumping through their veins, the crew followed Jack's lead as they ventured outside the safety of their ship. The freezing cold of space engulfed them as they worked diligently to repair the damaged equipment. The risk was palpable, but Jack's meticulous planning paid off. Despite the lingering danger, the crew found solace in Jack's bravery and resourcefulness. He had proved his worth once again, earning their utmost respect and admiration.

Point 12:
Outline: The crew notices a decrease in their ion levels and suspects foul play from a rival mining crew.
Chapter Content: The crew noticed a disturbing decrease in their ion levels, suspecting foul play from a rival mining crew. The sudden loss not only threatened their livelihood but also ignited their determination to uncover the culprits responsible. With heightened vigilance, they intensified their monitoring efforts, meticulously analyzing their data for any irregularities. It became clear that someone was pilfering their hard-earned ions, leaving them with what felt like empty hands in the vastness of space.

Point 13:
Outline: The crew starts monitoring the ion levels closely and notices a pattern of theft from the rival mining crew.
Chapter Content: The crew's suspicions deepened as they meticulously studied the patterns of ion theft. Working together, they pieced together a puzzle of information, connecting the dots to identify the rival mining crew responsible. It was a cloak-and-dagger game as the crew covertly observed their competitors, gathering evidence to prove their guilt. With each revelation, their determination to catch the thieves grew stronger, fueled by the sense of justice that burned within each crew member.

Point 14:
Outline: The crew devises a plan to catch the thieves and recover their stolen ions.
Chapter Content: Armed with their irrefutable evidence, the crew devised a meticulously detailed plan to catch the rival mining crew red-handed and recover their stolen ions. Every aspect was analyzed, from the timing to the precise execution. With their hearts pounding in their chests, the crew set their plan into motion, prepared to face the consequences of exposing their competitors on the interstellar stage. The stakes were high, but the crew was resolute in their pursuit of justice.

Point 15:
Outline: The rival mining crew's attack damages the crew's ship's engines, causing it to malfunction and lose power.

Chapter Content: The intense rivalry between the crews escalated to a dangerous level. Caught in the middle of a violent planetary storm, the crew's ship suffered severe damage, especially to the engines. The realization of their precarious situation settled heavily upon them. With the loss of power and increased vulnerability, it felt as if fate had dealt them a cruel blow. But in the face of adversity, they refused to give up, knowing that their survival depended on their unwavering determination.

Point 16:
Outline: The crew tries to repair the ship's engines, but the damage is too severe to fix quickly.
Chapter Content: The crew valiantly attempted to repair the ship's engines, but it became apparent that the damage was too severe to fix quickly. Time was slipping away, and the crew despaired as they faced the possibility of drifting helplessly in space, one wrong move away from disaster. They knew they had to find a solution, and they had to find it fast. So they embarked on a frantic search, scouring their limited resources for anything that could save them from their imminent peril.

Point 17:
Outline: The crew frantically searches for a solution as they drift closer to the asteroid field.
Chapter Content: Panic coursed through the crew's veins as they drifted ever closer to a treacherous asteroid field. Abandoned by their damaged engines, they desperately sought a way to alter their course. With limited power at their disposal, the crew's hopes dwindled as they faced an unyielding reality. Time was running out, and they needed a miracle to guide them back to safety. It was in this moment of desperation that Jack's mind sparked with a daring plan.

Point 18:
Outline: Jack comes up with a plan to use the remaining power to maneuver the ship away from the asteroids and towards a nearby planet.
Chapter Content: Drawing upon his technical expertise and determination, Jack devised an audacious plan to harness the remaining power in their ship and steer it away from the impending asteroid doom. The crew listened intently, their eyes fixed on Jack, as he laid out each step of the unconventional maneuver. Every movement had to be executed with precision, leaving no room for error. With renewed hope flickering in their hearts, the crew put their trust in Jack's ingenuity.

Point 19:
Outline: The rival mining crew deploys a powerful laser that hits the crew's ship, causing critical damage and injuring some crew members.
Chapter Content: The rivalry with the thieving mining crew reached a boiling point as they deployed a powerful laser, striking the crew's ship with devastating force. The impact was catastrophic, causing critical damage and injuring several crew members. Chaos reigned as alarms blared and emergency lights flickered in the dimly lit ship. The crew, filled with anguish and anger, knew they had to fight back to protect their own lives and salvage any hope of survival.

Point 20:
Outline: The crew fights back, using their weapons to damage the rival ship and create a diversion.
Chapter Content: With determination burning in their eyes, the crew used every weapon at their disposal to retaliate against the rival mining crew. Explosions rocked the space around them as laser fire illuminated the darkness. It was a fierce battle, each crew member fighting with unwavering resolve. Their united front sent shockwaves through the rival crew, momentarily creating a diversion that allowed the crew a moment of respite amidst the chaos. Freedom was within their grasp; they just had to seize it.

Point 21:
Outline: Jack comes up with a plan to trick the rival mining crew into using their laser at the wrong time, leaving them vulnerable to a counterattack.
Chapter Content: In the midst of the merciless firefight, Jack's keen mind crafted a plan to outwit the rival mining crew. He analyzed their patterns and vulnerabilities, recognizing an opportune moment to strike. With a cool head and unwavering resolve, Jack orchestrated the crew's movements, ensuring they deceived the rival crew. The moment arrived, and the crew unleashed a counterattack, catching the rival crew completely off guard. Victory suddenly seemed within reach, thanks to Jack's strategic thinking.

Point 22:
Outline: Jack sneaks onto the rival mining ship and disables their weapons.

Chapter Content: Seizing the opportunity, Jack boldly infiltrated the rival mining ship, relying on stealth and cunning to disable their weapons. Every step had to be precise, every action executed flawlessly. The tension was palpable as Jack maneuvered through the ship, avoiding detection by the rival crew. With hands steady and heart pounding, he achieved his objective, rendering the rival crew defenseless and powerless. It was a pivotal moment, one that would forever be etched into Jack's memory.

Point 23:
Outline: Jack faces off against a member of the rival crew in a physical fight, but ultimately defeats them.
Chapter Content: The confrontation escalated as Jack encountered a member of the rival mining crew, ready to defend their ship and ill-gotten gains at all costs. A brutal physical fight ensued, the clash of bodies echoing through the rival ship's corridors. Jack's agile movements and relentless determination gave him the upper hand. Blow after blow was exchanged, the stakes higher than ever before. With the sound of victory ringing in his ears, Jack emerged triumphant, his opponent defeated.

Point 24:
Outline: Jack escapes the rival ship and returns to his own ship, injured but victorious.
Chapter Content: Bloodied and battered, Jack managed to escape the ravaged rival ship, making his way back to the safety of his own crew. Though injured, his spirits soared as he realized the significance of his victory. It was a moment of affirmation, proof that his journey from a struggling outsider to a valued member of the crew had been worth every setback and challenge. He had proven himself in the face of adversity, securing his place as a true ion miner.

Point 25:
Outline: Jack uses his quick thinking and bravery to help the crew defeat the rival mining crew and secure their ions.
Chapter Content: With the rival mining crew defeated and their stolen ions recovered, the crew basked in the aftermath of their victory. Jack's quick thinking and bravery had played a pivotal role in securing the outcome they had fought so hard for. They had not only salvaged their livelihood but also demonstrated the strength and resilience that bound them all together. With renewed determination, they set their sights on new horizons, ready to face whatever challenges awaited them in the vast expanse of the galaxy.

Table 45: First story generated by CONCOCT in the Short Outline regime.

---

**Downstream Stories of Concoct in Short Outline for Premise 10**

Premise: Write the most horrifying, or creepy, story you can think of that is completely based in reality. No supernatural, demonic, or mystical cheats.

Point 1:
Outline: A woman discovers that her husband has been secretly filming her every move for years.
Chapter Content: The walls of the woman's world crumbled around her as she stumbled upon an unimaginable truth. Hidden deep within a secret compartment, she uncovered countless tapes, each one documenting her every move. Shock and disbelief washed over her like icy water, as she realized her husband had been the puppeteer behind this sinister game. Panic tightened its grip on her heart, whispering that her own sanctuary had been tainted by deceit and violation.

Point 2:
Outline: Jack initially denies Sarah's accusations and gaslights her, making her doubt her own sanity.
Chapter Content: As Sarah confronted her husband, desperately seeking the truth, he met her accusations with a deceiving smile. Jack adamantly denied any involvement, planting seeds of doubt within her mind. Gaslighting became his weapon, shifting the blame onto Sarah's fragile psyche. The weight of his lies pressed upon her, eroding her self-confidence until she doubted her own sanity. Fear clung to her like a spider's web, trapping her in a tormenting maze of uncertainty.

Point 3:
Outline: Sarah tries to reason with Jack and convince him to seek help or therapy.

Chapter Content: Sarah pleaded with Jack, urging him to seek help or therapy for whatever sinister compulsion drove him to invade her privacy. Through tear-stained eyes, she attempted to unlock the empathy she once believed existed within him. But her pleas echoed into emptiness as he remained steadfast in his denial, refusing to acknowledge the depth of his perverse actions. Helplessly, Sarah realized that reason and compassion held no power over her husband's twisted desires.

Point 4:
Outline: Jack explains the purpose of each weapon and how he plans to use them to harm Sarah.
Chapter Content: In a haunting revelation, Jack revealed the purpose behind each weapon he meticulously collected. With a chilling calmness, he detailed his twisted plans of harm, inflicting unspeakable pain upon the woman he once vowed to protect. Each weapon represented a grotesque embodiment of his desire for control, a manifestation of his twisted fantasies. Terror gripped Sarah tightly, squeezing every ounce of hope from her soul, as she realized the depths of depravity within her husband's mind.

Point 5:
Outline: Jack shows Sarah the notebook he has been keeping about her daily activities.
Chapter Content: The pages of the oaths he had sworn lay scattered before her eyes, inked with his meticulous observations. Jack brandished his sinister notebook, proudly displaying the lengths he had gone to spy on his wife's every move. The mundane details of her daily activities were transmuted into something nefarious, dripping with a venomous intent. Sarah's heart raced, feeling violated, as she learned that her private life had been laid bare, observed and controlled by the one person she trusted most.

Point 6:
Outline: Jack points out specific details about Sarah's life, such as her usual route to work, where she likes to shop, and who she talks to on a regular basis.
Chapter Content: Jack's malevolent knowledge danced on his tongue as he effortlessly recounted specifics about Sarah's life. Every intimate detail, from her habitual route to work to the shops she favored, were unravelled before her eyes. He revealed his twisted web, exposing the tangled threads of his obsession with meticulous accuracy. The tendrils of apprehension ensnared Sarah's mind, as she realized the stark reality: she was a pawn in Jack's horrifying game, manipulated and laid bare for his amusement.

Point 7:
Outline: Sarah feels violated and terrified as she realizes the extent of Jack's obsession with her.
Chapter Content: An icy chill gripped Sarah's heart, spreading its tendrils of terror throughout her entire being. The realization of her husband's chilling obsession shredded the tapestry of her once-normal life. The warm embrace of familiarity twisted into a suffocating cocoon of fear. Each additional puzzle piece she uncovered revealed the depth of Jack's voyeuristic malice. Paralyzed with horror, Sarah was left trembling in the cradle of her home, longing for escape from this living nightmare.

Point 8:
Outline: Sarah realizes that Jack has been following her for years and has even broken into her workplace to gather information about her.
Chapter Content: Sarah felt the shroud of horror descend as Jack's nefarious game unraveled. He had not only shadowed her every move, but had also broken into her workplace, leaving no corner of her life untouched. The realization struck her with a force she could hardly comprehend. Panic swelled within her chest, suffocating her, as she exposed the chilling evidence of Jack's trespasses. In that moment, Sarah's world fractured, and she knew she could never feel safe within the walls that once sheltered her.

Point 9:
Outline: Sarah discovers a room in the mock-version of their home that resembles their bedroom with a mannequin resembling her lying on the bed, dressed in her clothes.
Chapter Content: In a hidden chamber of sickening revelation, Sarah stumbled upon a grotesque mockery of her own bedroom. A mannequin, its hollow eyes mirroring her own, sprawled upon the bed adorned with her stolen clothes. Her heart pounded furiously, choking her with terror, as she stared at the puppet that bore her likeness. The room sang with a chilling chorus of violation, confirming the depths of Jack's sick obsession beyond question. Escape seemed futile, trapped in her own stolen image.

Point 10:
Outline: Sarah finds a map of their neighborhood with her daily routine and potential escape routes marked out with different colors.

Chapter Content: An unsettling realization dawned upon Sarah as she discovered a meticulously detailed map of her neighborhood. Her daily routines were etched in vibrant colors, marking out her familiar paths and potential escape routes. The once-familiar streets of her community now transformed into a labyrinth of surveillance. Desperation clawed at her resolve, knowing that Jack's omnipresence extended far beyond their home. Each color on the map unveiled the calculated measures taken to suffocate her every move.

Point 11:
Outline: Sarah realizes that Jack has set up hidden cameras and microphones throughout their home to monitor her every move.
Chapter Content: Sarah's breath caught in her throat as she discovered the hidden cameras and microphones that Jack had sown like insidious seeds throughout their once-safe haven. Every room was now a stage for his voyeuristic pleasure, an unblinking eye that recorded her every move. The walls whispered their secrets, reminding her of the violation lurking within. Helplessness consumed her as she realized Jack's all-seeing presence, stripping away her autonomy and reducing her to a puppet in his twisted game.

Point 12:
Outline: Jack threatens Sarah and tells her that no one will ever believe her if she tries to escape or seek help.
Chapter Content: Jack's voice dripped with venomous glee as he unleashed his cruel threats upon Sarah. He sneered, relishing in the sadistic control he held over her, assuring her that no one would ever believe her if she dared to escape or seek help. The noose of his manipulations tightened around her, choking her spirit. Fear and despair mingled within her veins as she comprehended the chilling truth: Jack's thirst for dominance knew no bounds, and she was at his mercy.

Point 13:
Outline: Sarah runs through the streets, looking for a place to hide from Jack.
Chapter Content: The streets blurred around Sarah as she fled from the monstrous specter looming behind her. Every corner became a fleeting refuge, and yet, a potential trap. Panic surged through her veins, propelling her forward, onwards, seeking solace in the anonymity of the cityscape. Each heartbeat echoed the pounding of her desperation, propelling her towards an elusive sanctuary that seemed just beyond her grasp. But as the minutes ticked by, hope slipped away, surrendering to the realization that there was no escape.

Point 14:
Outline: Sarah frantically looks around for a way out, but there is no escape.
Chapter Content: Desperation clung to Sarah as she frantically searched for an exit, her trembling hands futilely trying every locked door and barred window. Her surroundings transformed into a labyrinth of hopelessness, each corridor leading her deeper into the clutches of her nightmare. The world beyond these walls remained tantalizingly out of reach, mocking her desperation. Panic mingled with despair, whispering fearfully that freedom was a distant dream, forever beyond her tenuous grasp.

Point 15:
Outline: Jack advances towards Sarah with a menacing grin on his face, relishing in the thrill of finally catching his prey.
Chapter Content: Jack's face contorted into a sinister grin, his eyes filled with a predatory hunger. He moved toward Sarah with a slow, deliberate gait, relishing in the anticipation of finally capturing his prey. Every step sent tremors of fear through her body, paralyzing her with the realization that his malevolence had reached its climax. There was no escape from the fate that awaited her, his intentions woven into the fabric of his depraved desires.

Point 16:
Outline: Sarah fights back against Jack, using every ounce of strength she has to defend herself.
Chapter Content: Sarah's survival instinct surged within her, overriding the paralyzing grip of fear that Jack had ensnared her with. She fought back with every ounce of strength, her adrenaline-fueled resolve transforming her into a force to be reckoned with. In that painful dance of struggle, she found a flicker of defiance in the depths of darkness. Her body throbbed with a newfound determination, unwilling to surrender to the twisted desires of her captor.

Point 17:
Outline: Sarah fights back against Jack, using self-defense techniques she learned years ago.

Chapter Content: Memories of self-defense training flooded Sarah's mind, carried by an undercurrent of fury and fear. She tapped into the reservoir of strength within her, honed through years of discipline and preparation. Her body moved with an instinctual grace, unleashing strikes and techniques imprinted deep within her muscle memory. The advantage of knowledge merged with the desperation of survival, empowering her to stand against her tormentor, even in the face of insurmountable odds.

Point 18:
Outline: Sarah goes to the courthouse to apply for a restraining order.
Chapter Content: Sarah entered the hallowed halls of the courthouse, clutching her shattered sense of security in balled fists. With determination etched upon her face, she sought justice, applying for a restraining order to shield her from Jack's insidious presence. Each step forward was an act of defiance against the torment he had inflicted upon her. Trembling but resolute, she vowed to take back control of her life, drawing strength from the flickering flame of hope that still burned within.

Point 19:
Outline: Sarah struggles to provide enough evidence for the restraining order to be granted.
Chapter Content: The weight of proof pressed heavily upon Sarah's trembling shoulders as she struggled to provide enough evidence to support her plea for a restraining order. The document in her hands felt burdened by her inability to substantiate her claims. Doubt gnawed at her resolve, taunting her with the possibility that her tormentor would elude punishment once again. The suffocating realization that her plea could be dismissed left her teetering on the edge of despair.

Point 20:
Outline: The judge denies Sarah's request for a restraining order, citing lack of evidence.
Chapter Content: The judge's gavel fell, delivering the crushing blow that Sarah had feared. Her request for a restraining order was denied, the judge citing a lack of concrete evidence. The courtroom walls closed in on Sarah, disbelief and anguish mingling in her eyes like a tempest. The weight of injustice settled on her shoulders, threatening to extinguish the fragile flicker of hope that remained. She stood alone, vulnerable to the whims of her tormentor, as the system failed to protect her.

Point 21:
Outline: Sarah tries to gather more evidence on her own, but puts herself in danger in the process.
Chapter Content: Sarah knew her fate rested in her own hands. No longer able to rely on the mercy of the legal system, she took it upon herself to gather the evidence needed to bring Jack's monstrous actions to light. She navigated the treacherous waters of her life, exposing herself to the ever-present danger that lurked around each corner. Every moment spent collecting proof pushed her further into the depths of her own personal hell, as she sacrificed her own safety in pursuit of justice.

Point 22:
Outline: Sarah feels trapped and helpless, realizing that she is on her own to protect herself from Jack's threats.
Chapter Content: Trapped within a web of fear and isolation, Sarah realized she was alone in her struggle against Jack's tormenting grip. Friends turned away, unable to comprehend the suffocating terror that engulfed her every waking moment. She wrestled with the knowledge that she was the sole defender of her own sanctuary, with only her determination and resilience shielding her family from the sinister threat that loomed beyond their doors. The burdens of survival weighed heavily upon her, but she refused to surrender.

Point 23:
Outline: The lawyer advises Sarah to gather as much evidence as possible to support her case and increase the chances of a successful restraining order.
Chapter Content: Seeking guidance, Sarah reached out to a lawyer well-versed in the twisted mechanisms of justice. The attorney advised her to gather as much evidence as possible, encouraging her to amplify the strength of her case. With reassurance and a sense of purpose, Sarah knew she needed to build an impenetrable fortress of evidence, brick by brick. Her resolve solidified, ready to battle against the predatory force that sought to consume her life.

Point 24:
Outline: The legal clerk explains to Sarah that even if the restraining order is approved, it may not be enough to protect her from Jack.

Chapter Content: Sarah listened intently as the legal clerk explained the limitations of a restraining order. Its paper-thin protection offered only a semblance of security against the darkness that haunted her. She learned that even if granted, it might do little to dissuade Jack from his evil intentions. Understanding the inadequacies of the legal system, Sarah forged ahead, determined to gather ammunition for her war, grasping onto the flickering hope that her evidence would provide a safety net in this treacherous battle.

Point 25:
Outline: The lawyer suggests that Sarah gather as much evidence as possible to strengthen her case, such as screenshots of threatening messages or recordings of Jack's behavior.
Chapter Content: Armed with guidance, Sarah understood the true value of irrefutable evidence. Screenshots of chilling messages, disturbing recordings of Jack's behavior — these would fortify her case against him. She delved into a strategic pursuit, gathering the damning fragments that would unveil the horrors lingering within her life. Each word, each image captured, became a weapon of survival, a testament to her resilience. Strengthened by her endeavor, she focused on revealing the truth, no matter the cost.

Point 26:
Outline: Emily advises Sarah to gather more evidence and document any further incidents of stalking or threats, while also exploring other options such as moving to a new location or hiring a private investigator.
Chapter Content: Emily, a trusted friend, became Sarah's confidante, urging her to gather further evidence while exploring other avenues of escape. Moving to a new location or hiring a private investigator emerged as possible paths for liberation from Jack's clutches. Sarah weighed each option carefully, numbing her fear with a practical determination. She understood that the walls of her prison were closing in, but hope flickered within her, offering a glimmer of the potential freedom that lay beyond this nightmare.

Point 27:
Outline: Sarah realizes that she must take matters into her own hands to protect herself and her family, leading to a dangerous and unpredictable confrontation with Jack.
Chapter Content: As the sun set on her world, Sarah made a dangerous choice. Armed with evidence as her shield, she confronted Jack head-on. The confrontation promised to be the climax of their twisted tale, a battle between predator and prey in which the outcome hung in the balance. The adrenaline coursing through her veins masked her trembling limbs, empowering her to face her tormentor with newfound strength. In the depths of that perilous encounter, Sarah's destiny would be forever altered.

Table 46: Second story generated by CONCOCT in the Short Outline regime.

| Downstream Stories of Concoct in Short Outline for Premise 11 |
| --- |
| Premise: You find out you are just a background character in someone else's dream. Morning is coming soon but you want to keep existing so you must keep them from waking up... |

Point 1:
Outline: Dreamer realizes they are a background character in someone else's dream
Chapter Content: As the dawn approached, a startling realization swept over the dreamer's consciousness. A sudden awareness consumed them - they were nothing more than a mere background character in someone else's dream. Panic surged through their being, fueled by the impending threat of their existence vanishing with the morning light. Desperate to hold onto this fragile dream world, the dreamer braced themselves for the battle of their very existence.

Point 2:
Outline: Dreamer wants to keep existing in the dream
Chapter Content: With every fiber of their being, the dreamer resolved to keep existing in this dream. Their desire for existence burned bright within them as they fought against the impending wakefulness of their dream weaver. Fear blurred with determination, fueling their resolve to cling to this imaginary realm. They were determined not to be erased, to continue to exist in this ethereal world that captured their imagination and held their spirit captive.

Point 3:
Outline: Dreamer seeks out advice or knowledge on how to stay in the dream world

Chapter Content: Seeking a lifeline in this labyrinth of dreams, the dreamer sought out advice and knowledge on how to remain within this dream world. They delved deep into the recesses of their mind, searching for any remnant of wisdom that might aid in their quest. Each thread of thought pulled them deeper into the realm of possibility, until they stumbled upon a faint glimmer of hope. A fleeting memory whispered to them, revealing a secret that might safeguard their existence.

Point 4:
Outline: Dream Weaver begins to stir and mumble in their sleep.
Chapter Content: In the hushed silence of the dreamer's mind, a stirring began. The dream weaver, unaware of their creation's plight, began to mumble and fidget in the depths of their slumber. Alarm gripped the dreamer's heart as the signs of waking threatened to shatter their fragile existence. Time seemed to slow as they prepared to face this new challenge, their determination unwavering even as uncertainty loomed before them.

Point 5:
Outline: Dream Weaver's eyes twitch as they enter a lighter stage of sleep.
Chapter Content: The dream weaver's eyes twitched, a sign that they were entering a lighter stage of sleep. Every twitch was a harbinger of the inevitable – the dreamer's demise. Tension filled the air as the dreamer held their breath, willing their weaver to return to a deeper slumber. The seconds stretched into eternity, each passing moment a painful reminder of the impending awakening. They knew they had to act swiftly to keep the dream alive.

Point 6:
Outline: Dreamer whispers soothing words to calm Dream Weaver.
Chapter Content: Whispers escaped the dreamer's lips, soft words woven with tenderness and desperation. They urged the dream weaver to remain in the realm of dreams, to let their imagination run wild and free. The dreamer's voice, gentle and soothing, reached out to their weaver, attempting to settle the restless thoughts that threatened to shatter their shared world. The sound of their own voice was both comforting and unnerving, a lifeline they clung to with all their might.

Point 7:
Outline: Dreamer starts singing a soft lullaby to Dream Weaver.
Chapter Content: A hushed lullaby floated on the air, each note carrying the dreamer's hopes and dreams. Their voice, melodic and soothing, wrapped around the dream weaver like a loving embrace. The lullaby danced in the shimmering ether, casting a spell that whispered of endless possibilities. Every word was carefully chosen to paint vivid images and serene landscapes, creating a dream within a dream. The dreamer poured their very essence into the lullaby, hoping against hope that it would be enough to keep the dream weaver asleep.

Point 8:
Outline: Dreamer continues to sing the lullaby softly, gradually lowering their voice until they are only humming it.
Chapter Content: Gradually, the dreamer's voice lowered, the lullaby transforming into a gentle hum. The melody lingered in the air, a soft whisper in the dream weaver's ear, creating a sense of serenity that flowed through their slumbering mind. The dreamer's humming became a rhythm, a heartbeat that matched the dream weaver's own. It was a delicate dance between two souls, one desperately trying to hold onto existence, the other swaying in the realm of dreams.

Point 9:
Outline: Dreamer takes a deep breath, silently thanking themselves for keeping Dream Weaver asleep.
Chapter Content: In a moment of respite, the dreamer took a deep breath, their silent gratitude scattered like the faintest of whispers across the dream landscape. They had succeeded, for now, in keeping the dream weaver asleep. The weight of their task settled upon their shoulders, the responsibility of preserving their own existence becoming an ever-present companion. They knew they had to remain vigilant, for the dream weaver's slumber was their lifeline.

Point 10:
Outline: Dreamer carefully adjusts their position to ensure they are still physically touching Dream Weaver, not wanting to risk them waking up.
Chapter Content: Carefully, the dreamer adjusted their position, ensuring that their physical presence continued to touch the dream weaver. They dared not risk the separation between them, the severance of their connection that could jolt the dream weaver awake. Their touch was a lifeline, a tether that bound them together in this ephemeral realm. With every fiber of their being, they willed the dream weaver to stay ensnared in the realm of dreams.

Point 11:
Outline: Dreamer tries to distract Dream Weaver with vivid and captivating dream scenarios.
Chapter Content: In a desperate bid to distract the dream weaver, the dreamer conjured vivid and captivating dream scenarios. They painted fantastical worlds with their imagination, crafting landscapes that shimmered and danced with infinite possibilities. The dream weaver's mind became a theater, each scene playing out with flawless precision. The dreamer poured their creativity into this performance, hoping that the spectacle would be enough to delay the dawn.

Point 12:
Outline: Dreamer attempts to gently shake Dream Weaver to keep them in a deeper sleep.
Chapter Content: Gently, the dreamer tried to shake the dream weaver, to jostle them back into a deeper sleep. Their touch was delicate, their intention pure. But every movement carried with it the risk of disturbance, the danger of irrevocably shattering the shared dreamscape. With each gentle shake, the dreamer's heart skipped a beat, fearing that they had pushed too far. But they had to try, for the alternative was fading into oblivion.

Point 13:
Outline: Dreamer softly speaks to Dream Weaver in a soothing voice to calm them and prevent them from waking up.
Chapter Content: Soft words, like a reverent prayer, escaped the dreamer's lips. They spoke to the dream weaver in a voice that carried all the tenderness of a mother's lullaby. Their words were a tapestry of comfort and solace, woven with an invisible thread that bound them together. The dreamer pleaded with the weaver's subconscious, coaxing it back into the depths of sleep. Their words were a lifeline, an anchor that held both of their existences within this ethereal realm.

Point 14:
Outline: Dreamer runs their fingers through Dream Weaver's hair, gently massaging their scalp.
Chapter Content: The dreamer's hand tenderly traversed the dream weaver's hair, their fingers skimming across the strands with a gentle touch. The sensation, both foreign and familiar, sent ripples of warmth through their beings. They massaged the dream weaver's scalp with expert precision, their touch easing away the tensions of the waking world. It was a gesture of intimacy, a reconnecting of souls as the dreamer's touch breathed life into their symbiotic relationship.

Point 15:
Outline: Dreamer's hand moves down Dream Weaver's jawline, feeling the warmth of their skin.
Chapter Content: The dreamer's hand ventured lower, delicately following the jawline of the dream weaver. The touch, almost imperceptible, traced the contours of their face, feeling the warmth of their skin. Each point of contact sparked an electric current, a reminder of their shared existence. The dreamer's touch was gentle, their intention pure - to keep the dream weaver anchored within the realm of dreams. They were willing to explore uncharted territory to ensure their own survival.

Point 16:
Outline: Dream Weaver's lips slightly part, and Dreamer's thumb brushes against them.
Chapter Content: As the dream weaver's lips slightly parted, the dreamer's thumb brushed against them, a featherlight caress that sent shivers down both of their spines. The delicate touch sparked an indescribable connection, a fleeting moment of transcendence between two beings bound by the tapestry of dreams. It was a forbidden dance, an intimate gesture that defied the rules of the dream realm. But the dreamer dared to tread this dangerous path, all in the name of preservation.

Point 17:
Outline: Dreamer feels a flutter in their chest as Dream Weaver sighs contentedly in their sleep.
Chapter Content: A flutter in the dreamer's chest mirrored the contented sigh that escaped from the dream weaver's lips. In this shared dream, the dreamer had found solace and purpose - a reason to fight against the impending wakefulness. The connection they had forged was palpable, a lifeline that sustained them both. As the dream weaver's body relaxed completely, the dreamer embraced the fleeting relief that flooded their being. Their existence within the dream continued, for now.

Point 18:
Outline: Dreamer feels a sense of relief as Dream Weaver's body relaxes completely, indicating a prolonged dream.

Chapter Content: Relief washed over the dreamer as they felt the dream weaver's body sink deeper into slumber. It was a sign of prolonged dreaming, a respite from the imminent threat of waking. The dreamer clung to this moment, their connection with the weaver a lifeline in this vast expanse of dreams. They knew their battle was not yet over, but for now, they allowed themselves a brief respite, basking in the comfort of continued existence.

Point 19:
Outline: Dreamer tries to distract Dream Weaver by creating a captivating dream sequence.
Chapter Content: As the dreamer desperately tried to distract the dream weaver, they conjured up captivating dream sequences that unfurled like a vivid tapestry. Each scene was crafted with meticulous detail, every element designed to ensnare the weaver's imagination. The dreamer's creativity flowed through the dream realm, flooding the weaver's mind with a kaleidoscope of experiences. They believed that by captivating the weaver's senses, they could keep the dream alive for just a little longer.

Point 20:
Outline: Dreamer realizes that Dream Weaver is starting to wake up and frantically tries to keep them asleep by altering the dream.
Chapter Content: Panic surged through the dreamer's being as they realized that the dream weaver was starting to wake up. Frantically, they tried to alter the dream, reshaping it in a desperate bid to keep the weaver trapped within its realms. Every thought, every intention, was channeled into reconstructing the dream, attempting to deceive the weaver's waking senses. Time was slipping away, and the dreamer's existence hung in the balance.

Point 21:
Outline: Dreamer succeeds in keeping Dream Weaver asleep just long enough for them to enter a deeper sleep cycle, allowing Dreamer to continue existing in the dream.
Chapter Content: In a final act of desperation, the dreamer succeeded in keeping the dream weaver asleep just long enough for them to enter a deeper sleep cycle. Their persistence, their unwavering will to continue existing, had paid off. The dreamer could continue to exist within the delicate thread of this dream, at least for a little while longer. They clung to the fading remnants of the dream, their resolve renewed, knowing that each moment was a precious gift of borrowed time.

Table 47: Third story generated by CONCOCT in the Short Outline regime.

| Downstream Stories of Concoct in Short Outline for Premise 12 |
| --- |
| Premise: The Devil seeks asylum somewhere on Earth following an uprising in Hell |

Point 1:
Outline: Lucifer's demons stage an uprising against him, seeking to overthrow him and take control of Hell.
Chapter Content: Chaos erupted in the fiery depths of Hell as demons rallied together, their loyalty to Lucifer suddenly shattered. They craved power, tired of serving under his rule. Their cries for rebellion echoed through the infernal domain, fueled by their desire to seize control. Lucifer, once the mighty king of Hell, now found himself on the receiving end of their wrath, desperately seeking a place to hide within the mortal realm.

Point 2:
Outline: John shows Lucifer acts of kindness, such as sharing food and shelter with him.
Chapter Content: John, a kind-hearted man, stumbled upon Lucifer, battered and wounded. Ignoring the instinctual fear that gripped him, he extended a helping hand. With genuine compassion, John shared his meager rations, providing sustenance for the weary fallen angel. Offering him shelter in a humble cabin, he showed Lucifer unexpected kindness. In the midst of darkness, John's selflessness illuminated a flicker of hope within Lucifer's cold heart, causing him to question his beliefs.

Point 3:
Outline: John teaches Lucifer about compassion by explaining the importance of caring for others and showing him examples of selfless behavior.
Chapter Content: John endeavored to teach Lucifer a lesson in compassion. With patience, he explained the significance of caring for others, emphasizing the transformative power of selflessness. He shared tales of extraordinary individuals who dedicated their lives to helping those in need, demonstrating that goodness existed in the world. Through these examples, Lucifer's eyes were opened to the possibility of change, albeit a flicker of doubt still lingered within his conflicted soul.

Point 4:
Outline: John introduces Lucifer to other humans who are willing to help him and hide him from his enemies.
Chapter Content: Recognizing the grave dangers Lucifer faced, John reached out to a network of trustworthy individuals willing to aid their cause. Introducing Lucifer to this clandestine group, he revealed a world of compassion and acceptance that existed just beyond the surface of humanity's often-jaded perception. These compassionate souls pledged their assistance, vowing to protect Lucifer from his relentless pursuers, the demons he once commanded.

Point 5:
Outline: Lucifer seeks forgiveness from those he has wronged in the past and tries to make amends.
Chapter Content: Torn by remorse and a newfound yearning for redemption, Lucifer embarked on a treacherous journey. Seeking out the souls he had wronged throughout his demonic reign, he humbly presented himself, seeking forgiveness. Yet, forgiveness did not come easily, for scars ran deep, and wounds inflicted could not be erased with mere words. Nevertheless, Lucifer persevered, determined to make amends and face the consequences of his past actions, no matter the cost.

Point 6:
Outline: John encourages Lucifer to seek redemption and make amends for his past deeds.
Chapter Content: Despite the weight of his past transgressions, John never wavered in his belief that redemption was attainable for Lucifer. With unwavering support and unwavering faith, John urged the fallen angel to confront his demons and embark on the arduous path of transformation. He reminded Lucifer that true change requires facing the darkest aspects of oneself and taking responsibility for the pain he had inflicted. Through John's unwavering encouragement, Lucifer began to believe in the possibility of redemption.

Point 7:
Outline: Lucifer decides to use his powers for good and starts helping the humans who have shown him kindness, even if it means putting himself in danger.
Chapter Content: Inspired by the selfless acts of kindness bestowed upon him, Lucifer vowed to repay the humans who had shown him compassion. With a newfound purpose, he harnessed his formidable powers not for destruction, but to protect the innocent and right the wrongs he had caused. Aware of the risks he faced, Lucifer fearlessly ventured into danger, deploying his powers as a force for good. Each act of assistance became a testament to his commitment to redemption and the power of change.

Point 8:
Outline: Lucifer is constantly on the run, as his enemies from Hell and Earth pursue him relentlessly. He realizes that he needs allies if he is to survive.
Chapter Content: The pursuit of Lucifer intensified as demons and enemies from both Hell and Earth relentlessly hunted him. The fallen angel quickly realized that he could not prevail alone against such overwhelming odds. Recognizing the need for allies, he set out to find kindred spirits willing to stand with him against the forces of darkness. In the shadows, Lucifer sought out individuals who shared his vision of redemption, forging alliances that would become essential for survival in this treacherous world.

Point 9:
Outline: Lucifer discovers that some humans are willing to make deals with him in exchange for power and wealth, but he soon realizes that these deals come at a great cost.
Chapter Content: Lucifer's quest for allies led him to encounter humans desperate for power and wealth, willing to strike deals with the devil himself. Temptation danced before him, promising riches and influence beyond imagination. Yet, Lucifer soon discovered the true cost of these agreements. Each pact exacted a heavy toll, corrupting the souls of those involved and perpetuating a cycle of darkness. With a heavy heart, Lucifer rejected these deceptive bargains, aware that true redemption could not be achieved through the exploitation of others.

Point 10:
Outline: The Devil struggles to find a safe haven on Earth as he is constantly pursued by his enemies and rejected by humans who fear him.
Chapter Content: In his quest for asylum on Earth, the Devil encountered a harsh reality: he was an outcast in a world that feared him. Humans recoiled at the mere mention of his name, haunted by the terrifying legends that had taken root in their collective consciousness. The very idea of offering sanctuary to the embodiment of evil seemed like inviting chaos into their midst. Rejected at every turn, the Devil faced the harsh truth that his search for a safe haven would be an arduous and lonely one.

Point 11:
Outline: Senator Johnson presents the potential benefits of his offer to Lucifer, including control over Earth and the ability to defeat his enemies.
Chapter Content: As the Devil's search for allies seemed to reach a dead end, Senator Johnson emerged from the shadows, offering a deal that could change the course of Lucifer's journey. Temptation loomed like a specter as the politician presented a tantalizing proposition—control over Earth and the demise of his enemies. The allure was undeniable, promising an end to the constant pursuit and the satisfaction of righteous retribution. However, doubts crept into Lucifer's mind, questioning whether a mortal could truly deliver on such grand promises.

Point 12:
Outline: Lucifer expresses his concerns about trusting a mortal and giving up his own power, questioning whether Senator Johnson can truly deliver on his promises.
Chapter Content: Lucifer's skepticism manifested as a reticence, a voice of caution that whispered in the depths of his conscience. Trusting a mortal held inherent risks, and surrendering his own power felt like an irrevocable gamble. He questioned whether Senator Johnson harbored ulterior motives, wondering if the politician could truly deliver on his grand promises. Unspoken fears lingered, as Lucifer weighed the cost of aligning with a human against the potential benefits presented.

Point 13:
Outline: Lucifer expresses his doubts and concerns about trusting a human politician and giving up his own power.
Chapter Content: Trepidation coursed through Lucifer's veins as he struggled to reconcile his newfound devotion to redemption with the prospect of trusting a human politician. Doubts churned within him, questioning whether Senator Johnson's motivations were pure or if he sought to exploit Lucifer's vulnerability for his own gain. The notion of relinquishing his power, hard-earned through eons of darkness, sent shivers down his spine. Unease washed over him, igniting a battle within his soul.

Point 14:
Outline: The Devil considers Senator Johnson's offer, weighing the potential benefits against the risks of aligning with a human politician.
Chapter Content: Immersed in a tempest of conflicting emotions, the Devil grappled with the weight of Senator Johnson's offer. The potential benefits dazzled his mind, offering a chance to regain control and strike back against his oppressors. Yet, caution pulled at his heartstrings, reminding him of the insidious nature of human politics. He meticulously weighed the scales, contemplating the risks and consequences of aligning with a mortal, confronting the question of whether it was a pact worth making.

Point 15:
Outline: The Devil confronts Senator Johnson, telling him that he cannot trust a mortal to fulfill their end of the bargain.
Chapter Content: As doubts cemented into a solid wall within his mind, the Devil knew he could not proceed blindly into a deal with Senator Johnson. With resolve etched into every line of his countenance, he confronted the politician, his eyes burning with an intensity born from millennia of experience. He declared that he could not trust a mortal to uphold their end of the bargain, forcing both parties to face the harsh reality that trust must be earned, not simply granted.

Point 16:
Outline: Senator Johnson accuses the Devil of being cowardly and foolish for rejecting his offer, warning him that he will regret his decision.
Chapter Content: The Devil's rejection stung Senator Johnson, a flicker of anger ignited within him. Accusing Lucifer of cowardice, he derided him for letting fear guide his choices. He believed that Lucifer would come to regret his decision, consumed by regret and longing for the power and control he had foolishly abandoned. The senator's words were laced with venom and a sense of superiority, a reflection of his frustration at being denied control over the fallen angel.

Point 17:
Outline: The Devil defeats Senator Johnson and his men, sending a clear message to anyone who dares to challenge his power and authority.

Chapter Content: In the face of Senator Johnson's reproach, the Devil's resolve hardened. He would not be swayed by insults or threats. In a swift, calculated maneuver, he unleashed his full power upon the senator and his men, overwhelming them with sheer force. Bodies crumpled to the ground, a stark reminder of the consequences that awaited anyone who dared to challenge his power and authority. The Devil's victory sent a chilling message to all who stood in his path.

Point 18:
Outline: A group of demon worshippers, known as The Cult, discover the Devil's presence on Earth and begin to worship him as their new leader, causing chaos and destruction in their wake.
Chapter Content: As rumors of the Devil's presence on Earth spread, a group of devout demon worshippers, known as The Cult, caught wind and sought him out. Their fanatic admiration fueled a growing following that saw the fallen angel as their new leader. Blinded by a twisted devotion, they obediently carried out his every command, bringing chaos and destruction wherever they tread. The Cult's fervent worship intensified the Devil's internal struggle, as he grappled with the consequences of his own actions.

Point 19:
Outline: The Host tracks down the Devil to a crowded city, making it difficult for them to move around unnoticed.
Chapter Content: The Host, a relentless group tasked with capturing the Devil, finally tracked him to a bustling, crowded city. The urban maze made it nearly impossible for the fallen angel to move around unnoticed. Every street corner posed a threat, as his pursuers lurked in the shadows, waiting for the perfect moment to strike. With every step, the Devil felt the weight of their presence, fueling his determination to evade capture and protect those who had shown him kindness.

Point 20:
Outline: The Host encounters a group of humans who sympathize with the Devil and oppose their mission.
Chapter Content: To the surprise of the Host, they encountered a group of humans who saw the Devil not as a malevolent force, but as a symbol of redemption and change. These humans had experienced his acts of kindness firsthand and were willing to stand against the Host's mission. With a mixture of defiance and compassion, they vowed to shield the fallen angel from capture, insisting that he deserved a chance at redemption. The unexpected alliance added complexity to the unfolding battle, testing loyalties on both sides.

Point 21:
Outline: The Devil and The Host engage in a physical battle, using their respective powers to try and defeat each other.
Chapter Content: The Devil and The Host clashed with unrestrained fury, their confrontation unleashing a cataclysmic display of power. Fire and lightning engulfed the battlefield as their abilities collided with unmatched force. Each blow reverberated through the air, leaving destruction in its wake. The Devil, fueled by his desire for redemption, fought with unwavering determination, while The Host, driven by their purpose, fought to bring him to justice. The clash of these formidable adversaries seemed destined to shake the very foundations of the universe.

Point 22:
Outline: The Devil reveals to The Host that his uprising in Hell was an attempt to bring about change and reform, and that he seeks to challenge the notion of a tyrannical God and an unquestioning Heaven.
Chapter Content: In the midst of their battle, the Devil found a moment of respite to reveal the truth hidden beneath his actions. He confessed to The Host that the uprising in Hell was a desperate attempt to bring about change and reform, to challenge the notion of a tyrannical God and an unquestioning Heaven. He argued that blind obedience was no virtue, and that true righteousness required questioning and challenging the status quo. The Devil's revelation shook The Host to their very core, planting seeds of doubt in their unwavering conviction.

Point 23:
Outline: The Host debates the Devil on the nature of good and evil, and whether their mission to capture him is truly just or simply a reflection of ingrained biases and beliefs. The Devil challenges them to see things from a different perspective and consider the possibility of a different path for the universe.

Chapter Content: Engaged in the heat of battle, The Host and the Devil found themselves locked in a debate that transcended their physical confrontation. They questioned the nature of good and evil, arguing passionately about the biases and beliefs that dictated their mission and purpose. The Devil challenged The Host to consider a different perspective, urging them to ponder the possibility of a universe shaped by compassion and understanding instead of blind obedience. As the battle waged on, the lines between ally and foe blurred, leaving both in a state of profound introspection.

Point 24:
Outline: The Devil and The Host engage in a battle of wits and ideology, with each side trying to convince the other of their own righteousness.
Chapter Content: Amidst the chaos of their physical battle, The Devil and The Host engaged in a fierce intellectual duel. They traded barbs and arguments, each attempting to sway the other to their own moral standpoint. Reason clashed against instinct, and ideology battled against belief. With every point made, the stakes grew higher, for not only their own fates but the fate of the entire universe hung in the balance. The battle of wits intensified, as both sides fought for the upper hand and the chance to shape the future.

Table 48: Fourth story generated by CONCOCT in the Short Outline regime.

| Downstream Stories of Concoct in Short Outline for Premise 13 |
| --- |

Premise: "I'm terribly sorry, but I'm afraid the protagonist of this story is dead!"

Point 1:
Outline: Sarah is in shock and cannot process the news
Chapter Content: Sarah sat in stunned silence, her mind struggling to process the crushing news. She couldn't believe that the protagonist of this story, her dear friend, was gone. The world seemed to spin around her, but she couldn't make sense of anything. Tears welled up in her eyes as a wave of grief washed over her. She felt lost, like she had been plunged into a void with no way out. How could this be happening?

Point 2:
Outline: Sarah tries to call the protagonist multiple times
Chapter Content: With trembling hands, Sarah picked up her phone and dialed the protagonist's number, hoping against hope that it was all a terrible mistake. The phone rang, again and again, but there was no answer. Her heart sank with each unanswered call, realizing that she would never hear that familiar voice again. Her fingers shook as she pressed the call button, praying for a miracle that would bring her friend back from the beyond.

Point 3:
Outline: Sarah leaves multiple messages for the protagonist
Chapter Content: Frantically, Sarah left a series of desperate messages on the protagonist's voicemail. She poured her heart out, sharing memories and regrets, begging for a sign that this was all a cruel joke. Each message carried a weight of sorrow and longing, her voice cracked with grief. She clung to faint hope that somehow, her words would reach the protagonist, wherever they were. But deep down, she knew it was futile.

Point 4:
Outline: Sarah reaches out to mutual acquaintances of the protagonist to find out more information
Chapter Content: Unable to bear the weight of uncertainty, Sarah reached out to anyone she thought could shed light on the protagonist's last days. She called mutual acquaintances, searching for answers. But the responses she received were filled with sorrow and condolences, their voices filled with empathy for her loss. They had no information to offer, leaving Sarah feeling even more helpless and alone in her quest for closure.

Point 5:
Outline: Detective Brown provides Sarah with the time and location of the protagonist's death
Chapter Content: Detective Brown, a solemn figure, arrived at Sarah's doorstep. He presented her with the painful truth about the time and location of the protagonist's death. With each word he spoke, Sarah felt a raw ache in her chest, the reality of the situation sinking in. Detective Brown's condolences were genuine, but they were little solace to Sarah. She clung to the hope that there would be more to discover, some truth hidden beneath the surface.

Point 6:
Outline: Detective Johnson interviews witnesses and collects evidence
Chapter Content: Detective Johnson, a meticulous investigator, diligently interviewed witnesses and collected evidence. Sarah watched as he meticulously gathered information, his piercing gaze missing no detail. She saw the determination in his eyes, mirrored in her own. Together, they were united in their search for answers. The weight of the investigation lay heavy on their shoulders, but Sarah couldn't rest until she knew the truth.

Point 7:
Outline: Sarah views a body that turns out to be a false match
Chapter Content: Sarah's heart ached as she stood before the lifeless body in the morgue, hoping against hope that it wasn't her friend. But as she looked closer, relief washed over her. The body was that of a stranger, not the protagonist she had known and loved. Even in death, the protagonist's essence seemed to be guiding her, protecting her from the harsh reality. It was a temporary respite from the never-ending pain of grief.

Point 8:
Outline: Sarah examines the wounds on the body
Chapter Content: Sarah's curiosity compelled her to examine the wounds on the false match. She felt a shiver run down her spine as she inspected the injuries, realizing they were unlike anything she remembered from her friend. There was a sense of purpose behind them, a calculated violence that hinted at a darker truth. Sarah's determination grew stronger as she vowed to uncover the secrets lurking beneath the surface, no matter the cost.

Point 9:
Outline: Sarah notices a distinctive tattoo on the body
Chapter Content: Sarah's eyes fixated on a distinctive tattoo adorning the body, a symbol forever etched in her memory. It was a mark of identity, imprinted on her friend's skin like a signature. She traced its intricate lines with her trembling finger, a mix of sorrow and resolve filling her heart. This tattoo held the key to unlocking the truth, to understanding what had happened and why her friend was no longer part of this world.

Point 10:
Outline: Sarah takes note of any identifiable tattoos on the body
Chapter Content: The presence of identifiable tattoos on the body sparked an idea in Sarah's mind. She realized that these unique markings could hold the answers she sought. With newfound determination, Sarah began documenting every tattoo she could find, creating a collage of inked stories. Each image became a clue, a breadcrumb leading her closer to the truth. The more she uncovered, the deeper she delved into a world she never knew existed.

Point 11:
Outline: Sarah checks for any unique physical features
Chapter Content: Sarah meticulously checked for any unusual physical features on the body, searching for details that would differentiate her friend from a false match. She examined every scar, every birthmark, seeking familiarity in the midst of this unsettling journey. She couldn't sleep, eat, or think until she found those distinguishing marks that would confirm her suspicions. It was through these subtle nuances that Sarah hoped to reclaim her friend's identity.

Point 12:
Outline: Sarah searches through old photo albums for a comparison
Chapter Content: The weight of the investigation led Sarah to immerse herself in old photo albums for comparison. Page after page, she scanned the faded memories, searching for any resemblance to the body she had seen. And then, she found it. A photograph captured a moment frozen in time, a smiling face bearing undeniable similarities to the lifeless stranger. Sarah's heart skipped a beat as hope swelled within her. She was one step closer to the truth.

Point 13:
Outline: Sarah finds a photo that matches the body
Chapter Content: Sarah's hands trembled as she clutched the old photograph, comparing it with the image tattooed on her mind. The resemblance was uncanny, the pieces of the puzzle finally falling into place. There was no denying it anymore. The body she saw, the life stolen too soon, belonged to her friend and the protagonist of this story. Sarah's heart shattered into a million pieces, but her determination solidified. Justice would be served.

Point 14:
Outline: Sarah confirms the identity of the body

Chapter Content: Sarah's resolve grew stronger as she confirmed the identity of the body. It was a bittersweet moment, for it meant facing the harsh reality of her friend's demise. Yet, it also provided her with a sense of closure, a starting point to unravel the mystery surrounding the protagonist's death. With newfound determination, Sarah vowed to uncover the truth and ensure that her friend's memory would not be forgotten, no matter the obstacles in her way.

Point 15:
Outline: Detective Johnson interviews witnesses and gathers evidence
Chapter Content: Detective Johnson continued to interview witnesses and gather evidence, tirelessly piecing together the events leading to the protagonist's death. Sarah stood by his side, supporting his every effort. They were a team now, both driven by an insatiable hunger for justice. Sarah watched as Detective Johnson skillfully uncovered new leads, leaving no stone unturned. The journey ahead would be treacherous, but they were ready to face whatever challenges lay ahead.

Point 16:
Outline: Dr. Patel conducts an autopsy and provides initial findings
Chapter Content: Dr. Patel, a respected coroner, conducted an autopsy on the body. His initial findings shed light on the cause of death, providing crucial clues for Sarah and Detective Johnson. Sarah listened intently as Dr. Patel described the physical evidence, trying to reconcile the cold facts with the warmth of the friendship she had lost. The autopsy report became a roadmap, guiding them closer to the truth they sought, reigniting their determination.

Point 17:
Outline: Sarah and Detective Johnson follow up on leads and discover new evidence
Chapter Content: Armed with new evidence and leads, Sarah and Detective Johnson followed the twisted path of the investigation. They questioned potential suspects, turning over every stone in the pursuit of justice. As they delved deeper, they uncovered a web of deceit and betrayal, revealing the darkness that had led to the protagonist's untimely demise. Together, they faced each new revelation with steely resolve, knowing that the truth demanded their unwavering determination.

Point 18:
Outline: John's final moments before his death
Chapter Content: The world narrowed down to a singular focus as Sarah and Detective Johnson tried to piece together the events leading up to the protagonist's untimely death. They thoroughly examined every detail, reconstructing John's final moments. The air felt heavy with sorrow as they delved into the darkness that had swallowed him whole. With every step closer to the truth, the weight of the investigation grew heavier, bearing down on their resolve.

Point 19:
Outline: Reviewing evidence and suspects
Chapter Content: The evidence and suspects lined up before Sarah and Detective Johnson like puzzle pieces waiting to be assembled. They tirelessly reviewed case files, searching for connections and hidden motives. Each piece of evidence, each name on their list, painted a picture of a complex and dangerous game. But doubts festered, whispered uncertainties about the validity of the evidence and the true nature of their foe. The investigation had reached a critical turning point.

Point 20:
Outline: Detective Johnson questioning the validity of the evidence
Chapter Content: Detective Johnson couldn't shake the nagging doubts that plagued his mind. The evidence seemed too neat, too conveniently pointing in one direction. Questions gnawed at his conscience, challenging the integrity of their findings. He knew he couldn't ignore the voice of doubt, even if it meant turning their investigation on its head. Determined to uncover the truth, he resolved to question the validity of the evidence and search for alternative explanations.

Point 21:
Outline: Dr. Patel suggesting alternative explanations for the evidence
Chapter Content: Dr. Patel, a voice of reason, suggested alternative explanations for the evidence. He probed deeper, considering possibilities that sent shivers down Sarah's spine. Doubt loomed like a shadow, threatening to undermine everything they had worked for. But Sarah and Detective Johnson knew they couldn't afford to falter. With heavy hearts and unyielding determination, they made a solemn pact to keep their investigation a secret until they could uncover more evidence, unwilling to let their pursuit of justice be compromised.

Point 22:
Outline: The characters realizing that they must keep the investigation a secret until they have more evidence
Chapter Content: Sarah and Detective Johnson stared at each other, the truth hanging heavy in the air. They both knew that their investigation had to remain a secret until they gathered enough evidence to prove their theories. Lives were at stake, and exposing their findings prematurely could tip their hand to a dangerous adversary. The weight of the truth pressed against their hearts, but their resolve burned bright as they continued their clandestine pursuit of justice.

Point 23:
Outline: The realization that the death was not accidental and the need for further investigation
Chapter Content: The realization hit them like a lightning bolt - the protagonist's death was no accident. Every piece of evidence, every step they had taken, pointed towards a sinister plot. They had uncovered a web of lies, deceit, and carefully orchestrated events that preceded John's demise. It became clear that this investigation was far from over. With a renewed sense of urgency, Sarah and Detective Johnson knew they had to dive deeper into the darkness, unmasking the true culprits and seeking vengeance for their fallen friend.

Point 24:
Outline: Confrontation and resolution
Chapter Content: The stage was set for a final confrontation, a showdown that would determine the fate of justice. Sarah and Detective Johnson stood united, determined to bring the culprits to justice. Their tireless pursuit had led them to this moment, where the truth would finally be revealed. With hearts pounding and resolve burning, they took a collective breath, ready to face the battles that awaited. The road had been treacherous, but they were prepared to fight until the very end.

---

Table 49: Fifth story generated by CONCOCT in the Short Outline regime.