# OpenReview forum: "Improving Pacing in Long-Form Story Planning"
_EMNLP/2023/Conference — EMNLP 2023 Findings_

### Official Review · Reviewer_cJCu · 2023-07-28

**Typos Grammar Style And Presentation Improvements:** The figures on pages 14, 15 don’t con…
**Soundness:** 3

**Excitement:**

3: Ambivalent: It has merits (e.g., it reports state-of-the-art results, the idea is nice), but there are key weaknesses (e.g., it describes incremental work), and it can significantly benefit from another round of revision. However, I won't object to accepting it if my co-reviewers champion it.

**Paper Topic And Main Contributions:**

The authors present CONCOCT (Concrete Outline Control) for improving the pacing of a story outline. They first train a “concreteness evaluator” M from the BookSum dataset, where given two strings of text, M returns which text is more “concrete” (i.e. low-level detailed). Then, when viewing a story as a hierarchical tree outline, CONCOCT uses M to find fine the vaguest node in the tree and expands it into more precise sub-texts through prompting ChatGPT.

**Questions For The Authors:**

1) I encourage the authors to have a conversation about releasing CONCOCT source code and predictions. Since human rater preference is necessary for evaluating on WritingPrompts, any subsequent work on story pacing on WritingPrompts needs these predictions in order to do comparison. Otherwise, it becomes incredibly difficult to reproduce the results demonstrated in this paper due to the need for human evaluation.

2) The design of the concreteness scheduler needs some more detail, since this seems like an integral part of CONCOCT but it’s not clear to me how the authors came to the final schedule shown in Section C.3. For example, what is the value of E here? And how would the schedule compare to a very naive approach (e.g. fixed thresholds at each level of the tree hierarchy?)

**Reasons To Accept:**

This is a focused paper demonstrating a novel approach to story planning, by structuring stories as a hierarchical tree outline and measuring the “concreteness” of each node in this tree (i.e. each line of the story).

**Reasons To Reject:**

The authors claim that CONCOCT can improve a generated story's pacing. However, from what I can tell, there are major methodological problems with this claim:

1) The paper’s metrics only measure story pacing, and does not measure other qualities of the story generated (coherence, human preference, etc.). As such, while CONCOCT may improve the pacing of a story, it’s not clear if these other qualities of the generated story remain the same. For example, if CONCOCT improves story pacing but decreases the story's coherence, the generalizability of CONCOCT comes into question. It's very difficult to thoroughly assess this method without these evaluations.

    a) This is especially true as there have been a lot of prior work on structured methods to story generation evaluated on WritingPrompts (https://arxiv.org/abs/2212.10077, https://arxiv.org/abs/2210.06774, https://arxiv.org/abs/2112.08596) that are not focused on story pacing directly, but still follow a outlined structure. I wonder if these methods already have strong story pacing, especially compared to the paper's baseline of prompted ChatGPT.

2) It doesn’t seem like the authors actually generate a story given the outlines from CONCOCT, and all evaluation is done on the story outline itself. I don’t think this is a complete study — in order to claim that CONCOCT helps with story pacing, shouldn't the actual story be generated? Otherwise, there isn't much value to the story outline, especially since prior results on WritingPrompts actually generate the story. Please let me know if I've missed something in this paper.

**Reproducibility:**

3: Could reproduce the results with some difficulty. The settings of parameters are underspecified or subjectively determined; the training/evaluation data are not widely available.

**Reviewer Confidence:**

3: Pretty sure, but there's a chance I missed something. Although I have a good feel for this area in general, I did not carefully check the paper's details, e.g., the math, experimental design, or novelty.

---

> ### Author Rebuttal · Authors · 2023-08-29
>
> Thank you for your helpful comments! We appreciate that you found our contribution to be focused and novel. Thanks also for the great questions pointing to valuable additional experiments, which we’ve now run with positive results and will add to the main text.
>
> ***Other Metrics of Story Quality***
>
> In our human evaluations, we asked annotators to also label any “other errors” in addition to metrics related to pacing, and found that CONCOCT did not make more “other errors” than the baseline. We’ve now run an additional set of human evaluations following the main metrics from DOC [2], asking annotators to compare outlines from our “Long Outline” regime solely on overall interestingness, coherence, and premise relevance:
>
> **Long Outline**
> |  Model  | Interesting↑ | Coherent↑ | Relevant↑ |
> |:-------:|:------------:|:---------:|:---------:|
> | BASE    | 54.24        | 45.76     | 47.46     |
> | CONCOCT | 45.76        | 54.24     | 52.54     |While CONCOCT was significantly better on pacing (65 to 35), none of the differences above are significant, and CONCOCT’s average across these three metrics is slightly higher than the baseline’s anyway. The results corroborate our previous finding with the “other errors” metric that CONCOCT does not compromise other desirable qualities to improve pacing.
>
> ***1a. Prior Structured Methods for Story Generation***
>
> Thanks for mentioning the prior work on structured methods for story generation; indeed, we already cite a couple of these works (Re3 [1], and its follow-up DOC [2]). The most recent of these systems, DOC, in fact acknowledges problematic pacing in their Sec. 4 in the results discussion: they state that their outline events are “often inconsistent in level of detail: some remain too vague while others seem over-expanded” [2].
>
> In any case, our Base model is actually designed based on DOC’s outline generation structure– generating a hierarchical outline via breadth-first expansion– although we use a newer LLM (ChatGPT) compared to DOC. In fact, our outlines (both Base and CONCOCT) can be used with the DOC codebase *out of the box* for generating final stories, as we do below.
>
> ***Evaluation of Generated Stories***
>
> Thanks for your suggestion! We didn’t originally evaluate on generated stories because unlike end-to-end story generation systems like [1, 2], CONCOCT only focuses on outline generation. The process of turning the outline into a story is not part of our contribution. But we agree it’s good to evaluate the generated stories as well.
> Our outlines can be used to generate stories using DOC [2] out of the box, and we’ve now done so. As the resulting stories are quite long (often >5000 words) even when using outlines from our “Short Outline” regime, we compared similar-length excerpts rather than complete stories, and used GPT-4 rather than human evaluators (acknowledging that GPT-4’s judgments may be imperfect). Although turning outlines into stories introduces some noise, we found that CONCOCT’s story excerpts were still judged to be significantly more consistently-paced while not compromising other qualities:
>
> **Short Story Excerpts**
> | Model | Pacing↑ | Interesting↑ | Coherent↑ | Relevant↑ |
> |:-------:|:------------:|:---------:|:---------:|:---------:|
> |BASE| 42.82 | 46.29 | 49.26 | 50.50 |
> | CONCOCT    	  | 57.18 | 53.71 | 50.74 | 49.50 |
>
>
> ***Question 1: Reproducibility***
>
> We fully agree that it’s important to maximize reproducibility. Upon publication, we intend to open-source all of our code, trained models, data, and annotations from human evaluations; many of these items are already included in the supplementary zip we submitted.
>
>
> ***Question 2: Concreteness scheduler***
>
> Our scheduler design is motivated by our qualitative observation that it is easier (i.e., requires fewer samples on average) for our base LLM, ChatGPT, to generate more concrete expansions of a vague event than an already concrete one. Therefore, rather than a naive approach where we require new expansions to be more concrete by some fixed threshold T, we intuitively prefer to use a higher threshold initially and then decrease the threshold over time. Accordingly, we schedule T to decrease linearly over time using “E” in our final schedule in Sec. C.3, which simply denotes the number of remaining outline expansion steps to be conducted. However, we found that this linear schedule can sometimes set the initial threshold too high, causing our LLM to be unable to find any valid expansions. Hence the final T is the minimum of two terms, one term linearly decreasing over time, and one term based on differences in concreteness of already-generated outline events. There’s certainly more room for exploration on this threshold scheduling; we just picked one schedule that allowed for efficient sampling without being too lenient on accepting all new expansions.
> We will clarify these intuitions and provide full details in Sec. C.3., and also add some more explanation in the main text using the extra space allowed upon revision.
>
> ***Typos/Style Improvements***
>
> Thanks for pointing out the missing labels; we’ll fix them.
>
> ***Citation***
>
> [1] Yang, Kevin, Yuandong Tian, Nanyun Peng and Dan Klein. “Re3: Generating Longer Stories With Recursive Reprompting and Revision.” EMNLP 2022.
>
> [2] Yang, Kevin, Dan Klein, Nanyun Peng and Yuandong Tian. “DOC: Improving Long Story Coherence With Detailed Outline Control.” ACL 2023.

---

### Official Review · Reviewer_1ZA2 · 2023-08-04

**Soundness:** 3

**Excitement:**

3: Ambivalent: It has merits (e.g., it reports state-of-the-art results, the idea is nice), but there are key weaknesses (e.g., it describes incremental work), and it can significantly benefit from another round of revision. However, I won't object to accepting it if my co-reviewers champion it.

**Paper Topic And Main Contributions:**

This paper proposes the CONCOCT system to address the issue of inconsistent pacing in generating long-form stories using Large Language Models (LLM). They employ a concreteness evaluator to control the pacing in the hierarchical summary generation process and adopted a "vaguest-first" approach to achieve more balanced pacing. Additionally, the evaluator was used to filter new summary items, ensuring the inclusion of events with appropriate details. Comparisons with baseline methods show a significant improvement of over 60% in pacing consistency across different summary lengths.

**Reasons To Accept:**

- The problem is interesting and the proposed CONCOCT system alleviates the issue of inconsistent pacing in generating long-form stories or story outlines using LLMs.
- The overall writing is clear, there are sufficient details for readers to understand and reproduce the experimental results.

**Reasons To Reject:**

- The experimental design for comparing the CONCOCT system with the baseline hierarchical summary generator was insufficient and could be considered biased.
- The evaluation metrics are insufficient. The authors claim in line 2 that inconsistent pacing affects the reader experience negatively, thus it is expected to evaluate whether the proposed system successfully improve the likability of the generated story.

**Reproducibility:**

4: Could mostly reproduce the results, but there may be some variation because of sample variance or minor variations in their interpretation of the protocol or method.

**Reviewer Confidence:**

3: Pretty sure, but there's a chance I missed something. Although I have a good feel for this area in general, I did not carefully check the paper's details, e.g., the math, experimental design, or novelty.

---

> ### Author Rebuttal · Authors · 2023-08-29
>
> Thank you for your helpful comments! We’re glad that you found our problem to be interesting.
>
> ***Experimental Design for CONCOCT vs Baseline Outline Generator***
>
> In our human evaluations, we asked annotators to also label any “other errors” in addition to metrics related to pacing, and found that CONCOCT did not make more “other errors” than the baseline. According to your comment that our experimental design could be considered biased, we’ve now run an additional set of human evaluations following the main metrics from [1], asking annotators to compare outlines from our “Long Outline” regime solely on overall interestingness, coherence, and premise relevance:
>
> **Long Outline**
> |  Model  | Interesting↑ | Coherent↑ | Relevant↑ |
> |:-------:|:------------:|:---------:|:---------:|
> | BASE    | 54.24        | 45.76     | 47.46     |
> | CONCOCT | 45.76        | 54.24     | 52.54     |
>
> While CONCOCT was significantly better on pacing (65 to 35), none of the differences above are significant, and CONCOCT’s average across these three metrics is slightly higher than the baseline’s anyway. The results corroborate our previous finding with the “other errors” metric that CONCOCT does not compromise other desirable qualities to improve pacing.
> We hope that this additional experiment addresses your concern about possibly biased experimental design; please let us know if there was something else you were looking for in particular.
>
> [1] Yang, Kevin, Dan Klein, Nanyun Peng and Yuandong Tian. “DOC: Improving Long Story Coherence With Detailed Outline Control.” ACL 2023.
>
> ***Likability of Generated Story***
>
> It is commonly accepted by professional authors that good control of pacing is desirable in writing (e.g., [1, 2]). However, we did not mean to claim that CONCOCT’s uniform-pacing objective is necessarily best for improving story likability, and will clarify. As mentioned in our Discussion, our main goal in this work is to develop initial *tools* that can be used to control pacing in automatic generation– demonstrating their efficacy when applied with the goal of uniform pacing– but the best *objective* for pacing remains an open question. Thanks for pointing out our mention of “inconsistent” pacing in line 2; we’ll amend our wording in the abstract and introduction to e.g., “unnatural” pacing rather than “inconsistent” pacing.
>
> (On a related note, since you mention generated stories– we’ve now run an additional experiment indicating that generated stories using CONCOCT’s outlines are more consistently-paced compared to those generated from the baseline, shown under “2. Evaluation of Generated Stories” in our reply to reviewer cJCu.)
>
>
> [1] King, Stephen. “On Writing: A Memoir of the Craft.” Scribner, 2000.
>
> [2] Flaherty, Francis. “The Elements of Story.” HarperCollins, 2009.

---

### Official Review · Reviewer_7mEN · 2023-08-05

**Soundness:** 3

**Excitement:**

4: Strong: This paper deepens the understanding of some phenomenon or lowers the barriers to an existing research direction.

**Paper Topic And Main Contributions:**

The authors proposed CONCrete Outline ConTrol (CONCOCT) system to solve the problem of inconsistent pacing in conventional LLM-based story outline generation. The authors first trained a concreteness evaluator, then used it in the hierarchical generation process.


**Questions For The Authors:**

Question A: When I see the notation "Compress %", I get the impression that the higher the number, the higher the compression ratio. Is there a more intuitive way to express this?


**Reasons To Accept:**

The authors focus on the issue of pacing, and it can be highly evaluated that they addressed their own set of issues from the specific perspective of what is needed to write (or generate) a story. In the conclusion, the authors also mention cases in which the story writer intentionally changes the pacing, which makes the reviewer feel that this study was conducted from the user's perspective.
The paper's contribution is also clarified by the fact that the target term (concreteness) is well-defined.

**Reasons To Reject:**

In the Limitations part, the authors stated that the performance of the method depends on the performance of LLMs. Since the dependence on LLMs seems to be an issue not only in terms of performance but also in terms of reproducibility, etc., Limitations should be reconsidered.

**Reproducibility:**

4: Could mostly reproduce the results, but there may be some variation because of sample variance or minor variations in their interpretation of the protocol or method.

**Reviewer Confidence:**

3: Pretty sure, but there's a chance I missed something. Although I have a good feel for this area in general, I did not carefully check the paper's details, e.g., the math, experimental design, or novelty.

---

> ### Author Rebuttal · Authors · 2023-08-29
>
> Thank you for your helpful comments! We’re glad that you appreciated our perspective on story writing.
>
> ***LLMs and Reproducibility***
>
> Thanks for highlighting that reproducibility may also be dependent on LLMs. When we mentioned that CONCOCT’s performance might depend on LLM performance in the Limitations, we mainly meant that the *absolute* quality of the resulting outlines will of course depend on the base LLM. Comparing *relative* quality to an unaugmented baseline using the same base LLM, we think that using CONCOCT would still result in more uniformly-paced outlines. We will clarify the discussion in the Limitations accordingly.
>
> In any case, to maximize our work’s reproducibility, we’ve only used LLMs that are either open-source or accessible through a public API. We will also open-source all code and other artifacts upon publication.
>
> ***Unintuitive Compress % Notation***
>
> We actually completely agree that defining the compression ratio as [token count in summary / token count of raw], i.e. lower number = more compression, is somewhat unintuitive. However, this definition is used in several prior works [1,2,3,4], so we chose to be consistent with them. We’re happy to discard this definition completely and just directly report [token count of raw / token count in summary] as e.g., Raw/Sum instead of Compress% in Table 1.
>
>
> ***Citation***
>
> [1] Goldstein, Jade, Mark Kantrowitz, Vibhu Mittal, and Jaime Carbonell. "Summarizing text documents: Sentence selection and evaluation metrics." ACM SIGIR 1999.
>
> [2] Liu, Yizhu, Qi Jia and Kenny Q. Zhu. “Reference-free Summarization Evaluation via Semantic Correlation and Compression Ratio.” NAACL 2022.
>
> [3] Sclar, Melanie, Peter West, Sachin Kumar, Yulia Tsvetkov and Yejin Choi. “Referee: Reference-Free Sentence Summarization with Sharper Controllability through Symbolic Knowledge Distillation.” EMNLP 2022.
>
> [4] Vodolazova, Tatiana and Elena Lloret. “Towards Adaptive Text Summarization: How Does Compression Rate Affect Summary Readability of L2 Texts?” RANLP 2019.

---

### Meta-Review · Area_Chair_4QUc · 2023-09-20

**Recommendation:** 4

**Metareview:**

The paper presents a concentrated contribution in terms of creating a procedure to improve pacing in story generation. All reviews (after rebuttals) agree that the work is sound in terms of experiments that support its claims and a potentially novel methodology that can be built on.

---

### Decision · Program_Chairs · 2023-10-07

**Decision:**

Accept-Findings

**Comment:**

The paper presents a concentrated contribution in terms of creating a procedure to improve pacing in story generation. All reviews (after rebuttals) agree that the work is sound in terms of experiments that support its claims and a potentially novel methodology that can be built on.